# CORRECT-N-CONTRAST: A CONTRASTIVE APPROACH FOR IMPROVING ROBUSTNESS TO SPURIOUS CORRELATIONS

## ABSTRACT

Spurious correlations pose a fundamental challenge for building robust machine learning models. For example, models trained with empirical risk minimization (ERM) may depend on correlations between class labels and spurious features to classify data, even if these relations only hold for certain data groups. This can result in poor performance on other groups that do not exhibit such relations. When group information is available during training, Sagawa et al. (2019) have shown how to improve worst-group performance by optimizing the worst-group loss (GDRO). However, when group information is unavailable, improving worst-group performance is more challenging. For this latter setting, we propose Correct-N-Contrast (CNC), a contrastive learning method to train models more robust to spurious correlations. Our motivating observation is that worst-group performance is related to a representation alignment loss, which measures the distance in feature space between different groups within each class. We prove that the gap between worst-group and average loss for each class is upper bounded by the alignment loss for that class. Thus, CNC aims to improve representation alignment via contrastive learning. First, CNC uses an ERM model to infer the group information. Second, with a careful sampling scheme, CNC trains a contrastive model to encourage similar representations for groups in the same class. We show that CNC significantly improves worst-group accuracy over existing state-of-the-art methods on popular benchmarks, e.g., achieving 7.7% absolute lift in worst-group accuracy on the CelebA data set, and performs almost as well as GDRO trained with group labels. CNC also learns better-aligned representations between different groups in each class, reducing the alignment loss substantially compared to prior methods.

## 1 INTRODUCTION

For many tasks, deep neural networks are negatively affected by spurious correlations—dependencies between observed features and class labels that only hold for certain groups of the data. For example, consider classifying images of cows or camels, where 90% of cow images depict grassy backgrounds. A model may learn to predict the "cow" class based on the background, and incorrectly classify cow images with non-grass backgrounds as camels (Ribeiro et al., 2016; Beery et al., 2018; Kaufman et al., 2012). This illustrates a widespread issue where neural networks can achieve low test error on certain groups, yet high error on others (Blodgett et al., 2016; Buolamwini & Gebru, 2018; Hashimoto et al., 2018; Sagawa et al., 2019). Prior works have shown that this problem is increasingly aggravated as the correlations between class labels and spurious features become stronger (Sagawa et al., 2020) and easier to learn (Arpit et al., 2017; Hermann & Lampinen, 2020). Since spurious correlations arise in many settings, we wish to design robust methods that perform well on all groups.

How can we obtain neural networks robust to spurious correlations? If group-defining information (i.e. spurious attributes) is known, a common solution is to minimize the worst-group loss, e.g., with group DRO (GDRO) (Sagawa et al., 2019). However, such information may be expensive to collect, and we may not know the spurious attributes *a priori* in a given data set (Oakden-Rayner et al., 2020). When group information is unavailable, prior works typically take a two-stage approach. They first train an ERM model, and then use this model to infer groups and train a more robust model. For example, Sohoni et al. (2020) find that ERM models still learn group-specific features when trained to predict class labels. After first training an ERM model, they infer groups by clustering the ERM

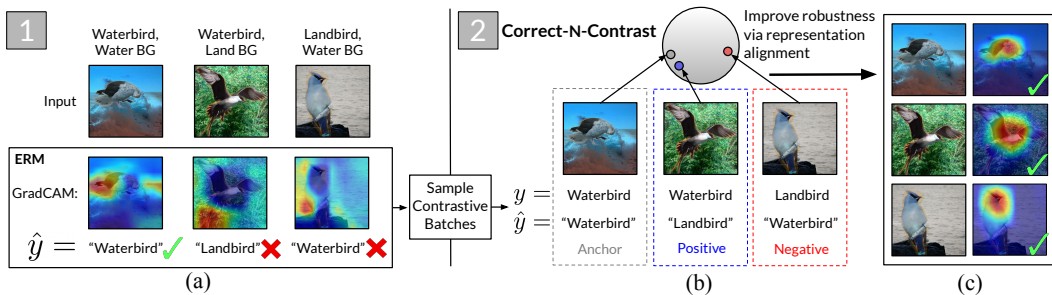

Figure 1: (a) ERM misclassifies samples by spurious background features, visualized with GradCAM (Selvaraju et al., 2017). (b) CNC uses contrastive learning to learn similar representations for same-class samples with different ERM predictions. (c) Resulting models ignore spurious attributes and classify samples correctly.

model's representations, and train a new model with GDRO using these inferred groups. Creager et al. (2021) identify groups under which an initial trained ERM model would maximally violate the invariant risk minimization (IRM) objective (Arjovsky et al., 2019). With these groups they train a new model with GDRO or IRM. Nam et al. (2020); Liu et al. (2021) observe that ERM models often misclassify data points in minority groups, and thus train another model with re-weighted or upsampled points misclassified by an initial ERM model. While these methods promisingly leverage ERM learned biases to significantly improve worst-group error without training group labels, there is still a gap between their robust performance and methods' such as GDRO that use group labels.

In this work, we ask how else we can improve model robustness using a trained ERM model, and aim to close this gap by focusing on improving the learned *representations* of the robust model in the second stage. We support this direction with two key motivations. First, we find that higher worst-group performance consistently correlates with hidden-layer representations exhibiting higher dependence on class labels than spurious attributes. We quantify this correlation using geometric representation *alignment* (Wang & Isola, 2020), which measures the closeness of samples with the same class but different spurious attributes in the model feature space, and mutual information. This relation consistently holds across various data sets, and explains when prior upweighting methods improve worst-group error over ERM (Fig. 4). Second, we theoretically show that a model's representation alignment for a given class can be used to upper bound the gap between its worst-group and average loss for that class. Thus, if we can improve representation alignment for a class, we can reduce the gap between worst-group and average loss for that class.

We thus propose Correct-N-Contrast (CNC), a two-stage procedure using contrastive learning to encourage better representation alignment within each class. In the first stage, we train a regularized ERM model similar to prior work (Liu et al., 2021; Creager et al., 2021), under the premise that ERM predictions help infer group information (i.e., spurious attributes). In the second stage, we wish to improve representation alignment by "pulling together" same-class datapoints and "pushing apart" different-class datapoints, regardless of their individual groups or spurious features. To do so via supervised contrastive learning, we use the heuristic that samples with the same ERM predictions exhibit similar spurious features (and vice versa). With a randomly sampled anchor, we select samples with the same class but different ERM predictions as "positives" we want to pull together, and samples from different classes but the same ERM prediction as *hard* "negatives" we want to push apart. Training a second model with this sampling scheme and supervised contrastive learning encourages this model to ignore spurious correlations that the initial ERM model learned, and improves representation alignment between same-class data points. Thus, CNC corrects for the ERM model's mistakes with contrastive learning in the second model.

We evaluate CNC on four popular and diverse spurious correlation benchmarks. Among methods that similarly do not assume training group labels, CNC substantially improves worst-group accuracy, obtaining up to **7.7%** absolute lift (from 81.1% to **88.8%** on CelebA) over the prior state-of-the-art JTT (Liu et al., 2021), and averaging **3.4%** lift across the four tasks. We also find that CNC nearly closes the gap in worst-group accuracy with robust training methods that assume training group labels, only falling short of GDRO's worst-group accuracy by 0.8% absolute. Finally, we validate that CNC indeed reduces the alignment loss compared to prior methods. This corresponds to an up to 71.1% smaller gap between worst-group versus average accuracy for data points in the same class.

**Contributions**. We summarize our contributions as follows:

1. We empirically show that a model's worst-group performance correlates with the model's alignment loss between different groups within a class, and analyze this connection theoretically.

2. We propose CnC, a two-stage contrastive approach to improve representation alignment and thereby learn representations robust to spurious correlations.

3. We validate that CnC significantly improves worst-group accuracy over existing methods on various benchmarks, and learns better-aligned representations less reliant on spurious features.

## 2 PRELIMINARIES

**Problem setup.** We present our setting and the loss objectives following Sagawa et al. (2019). Let $X = \{x_1, \ldots, x_n\}$ and $Y = \{y_1, \ldots, y_n\}$ be a training data set of size $n$. Each data point has an observed feature vector $x_i \in \mathcal{X}$, label $y_i \in \mathcal{Y}$, and *unobserved* spurious attribute $a_i \in \mathcal{A}$. The set of groups $\mathcal{G}$ is defined as the set of all combinations of class label and spurious attribute pairs, i.e. $\mathcal{G} = \mathcal{Y} \times \mathcal{A}$. Let $C = |\mathcal{Y}|$ be the number of classes and $K = |\mathcal{G}|$ be the number of groups. Following the classical supervised learning setting, we assume that each example $(x_i, y_i, a_i)$ is drawn from an unknown joint distribution $P$. We assume that at least one sample from each group is observed in the training data. Let $P_g$ be the distribution conditioning on $(y, a) = g$, for any $g \in \mathcal{G}$.

Given a model $f_\theta : \mathcal{X} \mapsto \mathbb{R}^C$ and a convex loss $\ell : \mathcal{X} \times \mathcal{Y} \mapsto \mathbb{R}$, let the worst-group loss be:

$$\mathcal{L}_{\text{wg}}(f_\theta) := \max_{g \in \mathcal{G}} \; \mathbb{E}_{(x,y,a) \sim P_g}[\ell(f_\theta(x), y)]. \tag{1}$$

ERM minimizes the training loss as a surrogate for the expected population loss $\mathcal{L}_{\text{avg}}$:

$$\mathcal{L}_{\text{avg}}(f_\theta) := \mathbb{E}_{(x,y,a) \sim P}[\ell(f_\theta(x), y)] \tag{2}$$

While ERM is the standard way to train neural nets, spurious correlations often cause ERM to obtain high error on minority groups even when average error is low. Group DRO, which minimizes the empirical version of (1), is recognized as a strong baseline for improving worst-group error when the group labels $\{a_1, \ldots, a_n\}$ are available during training (Sagawa et al., 2019). In contrast, we focus on the more challenging setting in which the group labels are *not available* during training.

**Contrastive learning.** We briefly describe contrastive learning (Chen et al., 2020), a central component of our approach. Let $f_\theta$ be a neural network model with parameters $\theta$. Let the encoder $f_{\text{enc}} : \mathcal{X} \mapsto \mathbb{R}^d$ be the feature representation layers of $f_\theta$. Let $f_{\text{cls}} : \mathbb{R}^d \mapsto \mathbb{R}^C$ be the classification layer of $f_\theta$, which maps encoder representations to one-hot label vectors. We learn $f_{\text{enc}}$ with the *supervised contrastive loss* $\mathcal{L}_{\text{con}}^{\text{sup}}$ proposed in Khosla et al. (2020). For each anchor $x$, we sample $M$ positives $\{x_i^+\}_{i=1}^M$ and $N$ negatives $\{x_i^-\}_{i=1}^N$. Let $y, \{y_i^+\}_{i=1}^M, \{y_i^-\}_{i=1}^N$ be the labels and $z, \{z_i^+\}_{i=1}^M, \{z_i^-\}_{i=1}^N$ be the normalized outputs of $f_{\text{enc}}(x)$ for the anchor, positives, and negatives respectively. With input $x$ mapped to $z$, the training objective for the encoder is to minimize:

$$\mathcal{L}_{\text{con}}^{\text{sup}}(x; f_{\text{enc}}) = \mathbb{E}_{x, \{x_i^+\}_{i=1}^M, \{x_i^-\}_{j=1}^N} \left[ -\log \frac{\exp(z^\top z_i^+ / \tau)}{\sum_{m=1}^M \exp(z^\top z_m^+ / \tau) + \sum_{n=1}^N \exp(z^\top z_n^- / \tau)} \right] \tag{3}$$

where $\tau > 0$ is a scalar temperature hyperparameter. Minimizing Eq. 3 leads to $z$ being closer to $z^+$ than $z^-$ in feature space. See Sec. 6 for further references related to contrastive learning.

## 3 MOTIVATIONS FOR REPRESENTATION ALIGNMENT

To motivate our method, we present our core observation that a model's worst-group accuracy correlates with how well its learned representations depends on the class labels, but not the spurious attributes. First, we empirically observe that ERM learns spurious correlations by inspecting their hidden layer representations on several spuriously correlated data sets. We find that ERM's worst-group performance is inversely related to a cross-group alignment loss (cf. Eq. (4) below) and mutual information metrics. Second, we theoretically prove that this alignment loss serves as an upper bound on the gap between the average-group loss and the worst-group loss (cf. Theorem 3.1).

### 3.1 RELATING WORST-GROUP PERFORMANCE TO REPRESENTATION ALIGNMENT

We first show that when neural networks are trained with standard ERM on spuriously correlated data, their hidden layer representations exhibit high dependence on the spurious attribute. We quantify this behavior using representation alignment (cf. Eq. (4) below) and mutual information metrics. We observe that these metrics explain trends in ERM's worst-group accuracy on various spuriously correlated data sets. This relationship is also consistent and applies to upsampling methods (JTT) that mitigate the impact of spurious features (Liu et al., 2021).

We model spurious correlations with CMNIST*, a colored MNIST data set inspired by Arjovsky et al. (2019). There are 5 digit classes and 5 colors. We color a fraction $p_{corr}$ of the training samples with a color $a$ associated with each class $y$, and color the test samples uniform-randomly. To analyze learned representations, we train a LeNet-5 CNN (LeCun et al., 1989) with ERM to predict digit classes, and inspect the outputs of the last hidden layer $z = f_{enc}(x)$. As shown in Fig. 2, with low $p_{corr}$, models learn representations with high dependence on the actual digit classes. However, with high $p_{corr}$ we learn $z$ highly dependent on $a$, despite only training to predict $y$.

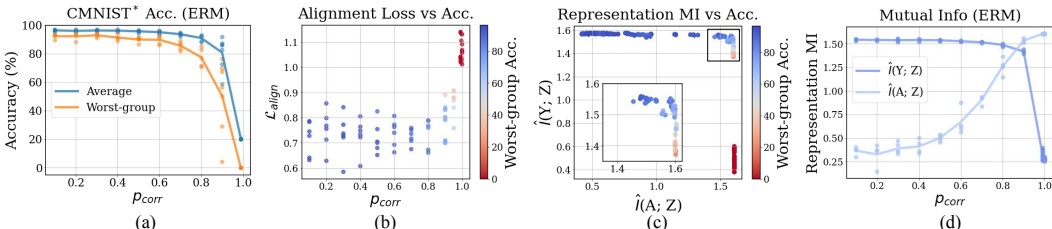

Figure 2: UMAP visualization of learned CMNIST* representations.

**Representation metrics**. To quantify this behavior, we use two metrics designed to capture how well the learned representations exhibit dependence on the class label vs. the spurious attributes. First, we compute an *alignment loss* $\hat{\mathcal{L}}_{align}(f_{enc}; g, g')$ between two groups $g = (y, a)$ and $g' = (y, a')$ where $a \neq a'$. This measures how well $f_{enc}$ maps samples with the same class, but different spurious attributes, to nearby vectors via Euclidean distance. Letting $G$ and $G'$ be the subsets of training data in groups $g$ and $g'$ respectively, and $x$ and $x'$ be any two samples in $G$ and $G'$, we define:

$$\hat{\mathcal{L}}_{align}(f_{enc}; g, g') := \frac{1}{|G|}\frac{1}{|G'|} \sum_{(x,y,a)\in G} \sum_{(x',y,a')\in G'} \|f_{enc}(x) - f_{enc}(x')\|_2. \qquad (4)$$

Thus, lower $\hat{\mathcal{L}}_{align}$ means better alignment. We also quantify representation dependence by estimating the mutual information (MI) of a model's learned representations with the class label, i.e. $\hat{I}(Y; Z)$ and the spurious attributes $\hat{I}(A; Z)$. We defer computational details to Appendix E.

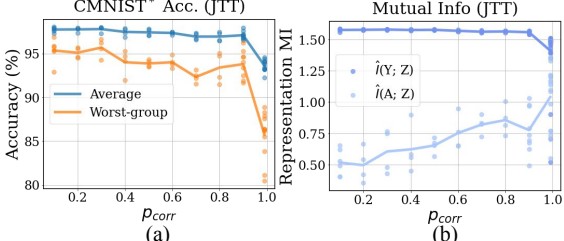

Figure 3: Accuracy and representation metrics from ERM models trained on increasingly spuriously correlated Colored MNIST. High worst-group accuracy corresponds to both $\hat{I}(Y; Z) > \hat{I}(A; Z)$ and small alignment loss.

**Results for ERM**. In Fig. 3 we show a strong association between worst-group error and both alignment and mutual information metrics. As $p_{corr}$ increases, ERM models not only drop in worst-group accuracy, but also incur higher alignment loss (Fig. 3ab). Fig. 3c further illustrates this with mutual information. We plot the estimated mutual information and worst-group accuracy for models at each epoch. A substantial drop in worst-group accuracy occurs with high $\hat{I}(A; Z)$ (especially when $\hat{I}(A; Z) > \hat{I}(Y; Z)$, even with high $\hat{I}(Y; Z)$). Fig. 3d also captures this trend with a trade off between high $\hat{I}(Y; Z)$ with $\hat{I}(A; Z)$ as $p_{corr}$ increases (Fig. 3a).

Figure 4: Higher worst-group accuracy with upsampling coincides with keeping $\hat{I}(Y; Z) \gg \hat{I}(A; Z)$.

**Results for JTT.** In Fig. 4, we also show that this relation holds when training with another recent (upsampling) approach, JTT (Liu et al., 2021). With high $p_{corr}$, models now achieve higher worst-group accuracy, and this corresponds to learning representations with high class label and low spurious attribute dependence. We note however that previous approaches do not explicitly optimize for these representation metrics, suggesting a new direction to improve worst-group performance.

## 3.2 RELATING ALIGNMENT LOSS TO WORST-GROUP LOSS

The empirical observations in Fig. 3 suggest that lower alignment loss correlates with lower worst-group error. Next, we show that this connection applies much more generally. We show that the maximum of $\hat{\mathcal{L}}_{align}(f_{enc}; g, g')$, over any two groups $g, g'$ within the same class, can be used to upper bound the gap between the worst-group loss and average loss for that class. We set up several notations before stating the result. For any class label $y \in \mathcal{Y}$, let $\mathcal{G}_y$ be the set of groups with label $y$ in $\mathcal{G}$. Let $\mathcal{L}_{wg}(f_\theta; y)$ be the worst-group loss among groups in $\mathcal{G}_y$:

$$\mathcal{L}_{wg}(f_\theta; y) := \max_{g \in \mathcal{G}_y} \mathbb{E}_{(x, \tilde{y}, a) \sim P_g} \left[ \ell(f_\theta(x), \tilde{y}) \right].$$

Let $\mathcal{L}_{avg}(f_\theta; y)$ be the average loss among groups in $\mathcal{G}_y$:

$$\mathcal{L}_{avg}(f_\theta; y) := \mathbb{E}_{(x, \tilde{y}, a) \sim P: \forall a \in \mathcal{A}} \left[ \ell(f_\theta(x), \tilde{y}) \right].$$

Additionally, we define a class-specific alignment loss $\hat{\mathcal{L}}_{align}(f_{enc}; y)$ among groups in $\mathcal{G}_y$. Recall that $f_\theta$ involves an encoding function $f_{enc}$ and a linear classification layer $f_{cls}$. We define $\hat{\mathcal{L}}_{align}(f_{enc}; y)$ as the largest cross-group alignment loss among groups in $\mathcal{G}_y$:

$$\hat{\mathcal{L}}_{align}(f_\theta; y) := \max_{g \in \mathcal{G}_y, g' \in \mathcal{G}_y: g \neq g'} \hat{\mathcal{L}}_{align}(f_{enc}; g, g'). \tag{5}$$

where $\hat{\mathcal{L}}_{align}(f_{enc}; g, g')$ is the alignment loss between $g$ and $g'$ defined in Eq. (4). Our main result is that $\hat{\mathcal{L}}_{align}(f_\theta; y)$ is an upper bound on the gap between $\mathcal{L}_{wg}(f_\theta; y)$ and $\mathcal{L}_{avg}(f_\theta; y)$ (up to a norm multiplier and a concentration error), for any $y \in \mathcal{Y}$.

**Theorem 3.1** (Alignment loss upper bounds the gap between worst-group and average-group loss)**.** *In the setting described above, let $f_\theta$ be any neural network satisfying that the weight matrix of the linear classification layer $W$ in $f_{cls}$ satisfies that $\|W\|_2 \leq B$, for some constant $B$. Let $n_g$ be the size of any group $g \in \mathcal{G}$ in the training data set. Assume that the loss function $\ell(x, y)$ is $C_1$-Lipschitz in $x$ and bounded from above by $C_2$, for some positive constants $C_1, C_2$. Then, with probability at least $1 - \delta$ over the randomness of the training data set samples, for any class $y \in \mathcal{Y}$, the following holds:*

$$\mathcal{L}_{wg}(f_\theta; y) \leq \mathcal{L}_{avg}(f_\theta; y) + B \cdot C_1 \cdot \hat{\mathcal{L}}_{align}(f_\theta; y) + \max_{g \in \mathcal{G}_y} C_2 \sqrt{\frac{8 \log(|\mathcal{G}_y|/\delta)}{n_g}}. \tag{6}$$

The proof of Theorem 3.1 is deferred to Sec. B. Since we also know that $\mathcal{L}_{avg}(f_\theta; y) \leq \mathcal{L}_{wg}(f_\theta; y)$, the above result implies that in order to reduce the gap between the worst-group loss and the average loss for class $y$, it suffices to reduce the alignment loss $\hat{\mathcal{L}}_{align}(f_\theta; y)$.

**Broader algorithmic implications**. We summarize Section 3 with two takeaways: (1) When trained on spuriously correlated data sets, ERM networks learn data representations highly dependent on spurious attributes. Clusters of these representations (Sohoni et al., 2020) or the ERM model's outputs (Liu et al., 2021; Nam et al., 2020) can thus serve as (noisy) pseudolabels for spurious attributes. (2) Both representation metrics correlate with worst-group error, such that a viable way to improve worst-group performance is to improve representation alignment within each class.

## 4 CORRECT-N-CONTRAST (CNC)

We now present CNC, a two-stage method to improve worst-group performance and robustness to spurious correlations, without requiring training group labels. Similar to prior works (Sohoni et al., 2020; Liu et al., 2021), our first stage trains an ERM model (with proper regularization[1]) on the training set, ultimately to infer group labels based on samples' spurious attributes.

---

[1] As we train on the same data set we infer the groups on, regularization (via high weight decay or early stopping) is purely to prevent the ERM model from memorizing the class labels. This is standard practice also discussed in Sohoni et al. (2020); Liu et al. (2021). We show in Sec. 5.3 that we do not require the ERM model to perfectly learn the spurious attributes for CNC to substantially improve robustness in practice.

---

**Algorithm 1** Correct-N-Contrast (CNC)

---

**Input:** Training data set $(X, Y)$; # positives $M$; # negatives $N$; learning rate $\eta$, # epochs $K$.
    **Stage 1: ERM Training**
1: Train a regularized ERM model $f_{\hat{\theta}}$ on $(X, Y)$; save the predictions $\hat{y}_i := f_{\hat{\theta}}(x_i)$.
    **Stage 2: Supervised contrastive learning**
2: **for each** epoch $1, \ldots, K$ **do**
3:   **for each** anchor $(x, y) \in (X, Y)$ **do**
4:     Let $\hat{y}$ be the predicted (group) label of $x$ from Stage 1's ERM model.
5:     Get $M$ positives $\{(x_m^+, y_m^+)\}$ where $y_m^+ = y$ but $\hat{y}_m^+ \neq \hat{y}$, for $m = 1, \ldots, M$.
6:     Get $N$ negatives $\{(x_q^-, y_q^-)\}$ where $y_q^- \neq y$ but $\hat{y}_q^- = \hat{y}$, for $q = 1, \ldots, N$.
7:     Update $f_\theta$ by $\theta \leftarrow \theta - \eta \cdot \nabla \hat{\mathcal{L}}(f_\theta; x, y)$ (cf. Eq. (7)) with anchor, $M$ positives, and $N$ negatives.
    **return** final model $f_\theta$ from Stage 2, and throw away the ERM model from Stage 1.

---

The key difference is our second stage: we aim to train a more robust model by learning representations such that samples in the same class but different groups are close to each other. We use contrastive learning, as intuitively by treating samples with the same class but different spurious attributes as distinct "views" of the same class, we train the second stage model to "pull together" these samples' representations and ignore the different spurious features. This is also inspired by Wang & Isola (2020); Robinson et al. (2021), who show that minimizing the contrastive loss improves representation alignment between distinct "views". Later in Sec. 5.1, we verify that CNC indeed reduces $\hat{\mathcal{L}}_{\text{align}}(f_\theta; y)$ substantially. We include further details on both stages below, and summarize CNC in Algorithm 1.

**Stage 1: ERM training.** We train an initial model $f_{\hat{\theta}}$ on the training data set $\{(x_i, y_i)\}_{i=1}^n$ with ERM and regularization, and save its predictions $\{\hat{y}_i\}_{i=1}^n$ on the training data points. We consider two ways to source predictions: using the ERM model's outputs, and clustering its last hidden-layer representations. Both approaches aim to accomplish the same goal of exploiting the ERM model's learned spurious correlations; further details are in Appendix E.2.

**Stage 2: Contrastive learning (CL).** Next, we train a robust model with supervised contrastive learning using the ERM predictions. While CNC is inspired by recent CL works (Chen et al., 2020; Khosla et al., 2020), we introduce new "contrastive batch" sampling and optimization objectives.

***Contrastive batch sampling.*** As described in Sec. 2, contrastive learning requires sampling anchors, positives, and negatives with the general form $\{x\}, \{x^+\}, \{x^-\}$. Here, we wish to sample points such that by maximizing the similarity between anchors and positives (and keeping anchors and negatives apart), the Stage 2 model "ignores" spurious similarities while learning class-consistent dependencies. With prediction set $\{\hat{y}_i\}_{i=1}^n$, for each batch we randomly sample an anchor $x_i \in X$ (with label $y_i$ and ERM prediction $\hat{y}_i$), $M$ positives with the same class as $y_i$ but a different ERM model prediction than $\hat{y}_i$, and $N$ negatives with different classes as $y_i$ but the same ERM model prediction as $\hat{y}_i$. For more signal per batch, we double pairwise comparisons by switching anchor and positive roles.

***Optimization objective and updating procedure.*** While our core objective is to learn aligned representations via contrastive learning, we also wish to train the full model to classify datapoints correctly. As we have the training class labels, we jointly update both the model's encoder layers $f_{\text{enc}}$ with a standard contrastive loss, and the full model $f_\theta$ with a cross-entropy loss:

$$\hat{\mathcal{L}}(f_\theta; x, y) = \lambda \hat{\mathcal{L}}_{\text{con}}^{\text{sup}}(f_{\text{enc}}; x, y) + (1 - \lambda)\hat{\mathcal{L}}_{\text{cross}}(f_\theta; x, y). \tag{7}$$

In the above, $\hat{\mathcal{L}}_{\text{con}}^{\text{sup}}(f_{\text{enc}}; x, y)$ is the supervised contrastive loss of $x$ along with its positive and negative samples, similar to Eq. (3) (see Eq. (16) in Sec. C.2 for the full equation); $\hat{\mathcal{L}}_{\text{cross}}(f_\theta; x, y)$ is averaged cross-entropy loss over $x$, the $M$ positives, and the $N$ negatives; $\lambda \in [0, 1]$ is a balancing hyperparameter. As a remark, the loss objective (7) uses a single anchor in each batch in our setting.

To calculate the loss, we first forward propagate one batch $(x_i, \{x_m^+\}_{m=1}^M, \{x_q^-\}_{q=1}^N)$ through $f_{\text{enc}}$ and normalize them to obtain representation vectors $(z_i, \{z_m^+\}_{m=1}^M, \{z_q^-\}_{q=1}^N)$. To learn closely aligned $z_i$ and $z^+$ for all $\{z_m^+\}_{m=1}^M$, we update $f_{\text{enc}}$ with the $\hat{\mathcal{L}}_{\text{out}}^{\text{sup}}(x; f_{\text{enc}})$ loss. Finally, we also pass the unnormalized outputs of the encoder $f_{\text{enc}}$ to the classifier layers $f_{\text{cls}}$, and compute a batch-wise cross-entropy loss $\hat{\mathcal{L}}_{\text{cross}}(f_\theta)$ using each batch sample's class labels and $f_\theta$'s outputs. Due to space constraints, we include further implementation details and sampling considerations in Appendix C.

## 5 EXPERIMENTAL RESULTS

We conduct experiments to answer the following questions: (1) Does CNC improve worst-group performance over prior state-of-the-art methods on data sets with spurious correlations? (2) Does CNC actually encourage learning hidden layer representations with greater alignment and class-label-only dependence? How is this impacted by the strength of a spurious correlation in the data? (3) Does CNC require perfectly predicting the spurious attribute to work well in practice? Our results for each question follows in the next three subsections (5.1, 5.2, and 5.3). Due to space constraints, we defer ablations on CNC's design choices, including the representation-learning objective and sampling procedure, to Appendix A. Additional comparison to alignment methods proposed for domain adaptation but adjusted for our setting are in Appendix A.2. Below, we briefly describe the benchmark data sets used in this section. We run CMNIST* with $p_{corr} = 0.995$. Further details on data sets, models, and experimental hyperparameters are deferred to Appendix E.

**Waterbirds** (Sagawa et al., 2019): We classify $\mathcal{Y} = \{$waterbird, landbird$\}$, where $95\%$ of images have the same bird type and background $\mathcal{A} = \{$water background, land background$\}$.

**CelebA** (Liu et al., 2015): We classify celebrities' hair color $\mathcal{Y} = \{$blond, not blond$\}$ with $\mathcal{A} = \{$male, female$\}$. Only $6\%$ of blond celebrities in the data set are male.

**CivilComments-WILDS** (Borkan et al., 2019; Koh et al., 2021): We classify $\mathcal{Y} = \{$toxic, not toxic$\}$ comments. $\mathcal{A}$ denotes whether the comment mentions one of eight demographic identities.

### 5.1 CNC IMPROVES WORST-GROUP PERFORMANCE

To study (1), we evaluate CNC on image classification and NLP data sets with spurious correlations. As baselines, we compare against standard ERM and an oracle GDRO approach that assumes access to the group labels. We also compare against recent methods that tackle spurious correlations without requiring group labels: CVaR DRO (Levy et al., 2020), GEORGE (Sohoni et al., 2020), Learning from Failure (LfF) (Nam et al., 2020), Predictive Group Invariance (PGI) (Ahmed et al., 2021), Environment Inference for Invariant Learning (EIIL) (Creager et al., 2021), Contrastive Input Morphing (CIM) (Taghanaki et al., 2021), and Just Train Twice (JTT) (Liu et al., 2021). We also compare against a CNC version without the Stage 1 ERM model, instead only sampling positives and negatives based on class (denoting this SupCon*). Results are reported in Table 1. CNC achieves highest worst-group accuracy among all methods without training group labels on the CMNIST* Waterbirds and CelebA data sets, while also obtaining near-SoTA worst-group accuracy on CivilComments.

While LfF, GEORGE, PGI, EIIL, and JTT similarly use a trained ERM model to estimate groups, CNC uniquely uses ERM predictions to encourage the robust model to learn desirable representations via contrastive learning. We reason that with this approach, by sampling positives and negatives from the ERM predictions, CNC more directly encourages the robust model to ignore learnable spurious correlations compared to previous invariant learning, input transformation, or upweighting approaches. We include additional evidence of this via GradCAM visualizations in Appendix G.

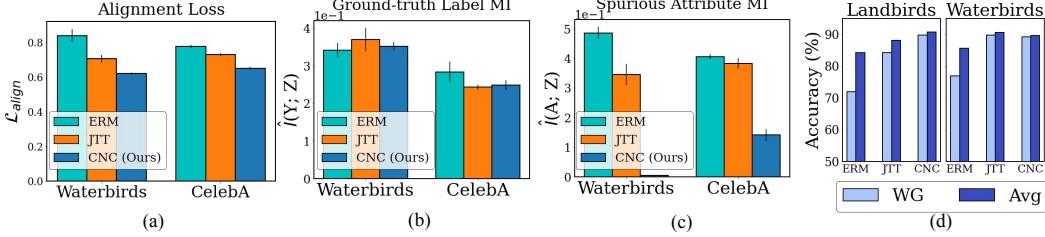

Figure 5: Alignment loss (a) and mutual information (b, c) of models trained with ERM, JTT, and CNC, on Waterbirds and CelebA. CNC most effectively removes dependence on the spurious attribute, and obtains smaller gaps for per-class worst-group vs. average error (d), as supported by Thm. 3.1.

### 5.2 CNC LEARNS REPRESENTATIONS LESS RELIANT ON SPURIOUS FEATURES

To shed light on CNC's worst-group accuracy gains, we investigate if models trained with CNC actually learn representations with higher alignment. Compared to ERM and JTT (the next-best performing method that does not require subgroup labels), CNC learns representations with significantly higher alignment (lower alignment loss) and lower mutual information with spurious attributes (while having comparable mutual information with class labels) (Fig. 5 and Fig. 7).

Table 1: Worst-group and average accuracies, averaged over three seeds (standard deviations in parenthesis). On image data sets, CNC obtains significantly higher worst-group accuracy than comparable methods without group labels, competing with GDRO. CNC also competes with SoTA on CivilComments. Results without standard deviations are reported from the original papers. Further implementation details are in Appendix E.

| Method | CMNIST* | | Waterbirds | | CelebA | | CivilComments-WILDS | |
|---|---|---|---|---|---|---|---|---|
| Accuracy (%) | Worst-group | Avg. | Worst-group | Avg. | Worst-group | Avg. | Worst-group | Avg. |
| ERM | 0.0 (0.0) | 20.1 (0.2) | 72.6 | 97.3 | 47.2 | 95.6 | 57.4 | 92.6 |
| CVaR DRO | 22.1 (5.0) | 61.3 (4.1) | 75.9 | 96.0 | 64.4 | 82.5 | 60.5 | 92.5 |
| LfF | 0.0 (0.0) | 25.0 (0.5) | 78.0 | 91.2 | 77.2 | 85.1 | 58.8 | 92.5 |
| GEORGE | 76.4 (2.3) | 89.5 (0.3) | 83.8 (1.0) | 95.7 (0.5) | 54.9 (1.9) | 94.6 (0.2) | - | - |
| PGI | 73.5 (1.8) | 88.5 (1.4) | 73.8 (0.8) | 84.6 (0.1) | 77.8 (1.8) | 82.0 (0.6) | - | - |
| CIM | 0.0 (0.0) | 36.8 (1.3) | 77.2 | 95.6 | 83.6 | 90.6 | N/A | N/A |
| EIIL | 72.8 (6.8) | 90.7 (0.9) | 78.7 | 96.9 | 81.7 (0.8) | 85.7 (0.1) | 67.0 (2.4) | 90.5 (0.2) |
| JTT | 74.5 (3.1) | 90.2 (0.8) | 86.7 | 93.3 | 81.1 | 88.0 | **69.3** | 91.1 |
| SupCon* | 0.0 (0.0) | 22.4 (1.2) | 71.0 (1.9) | 85.9 (0.8) | 62.2 (1.1) | 90.0 (0.1) | - | - |
| CNC (Ours) | **77.4 (3.0)** | 90.9 (0.6) | **89.7 (0.2)** | 90.8 (0.1) | **88.8 (0.9)** | 89.9 (0.5) | 68.9 (2.1) | 81.7 (0.5) |
| Group DRO | 78.5 (4.5) | 90.6 (0.1) | 91.1 (0.2) | 92.4 (0.2) | 88.9 (1.3) | 93.9 (0.1) | 69.8 (2.4) | 89.0 (0.3) |

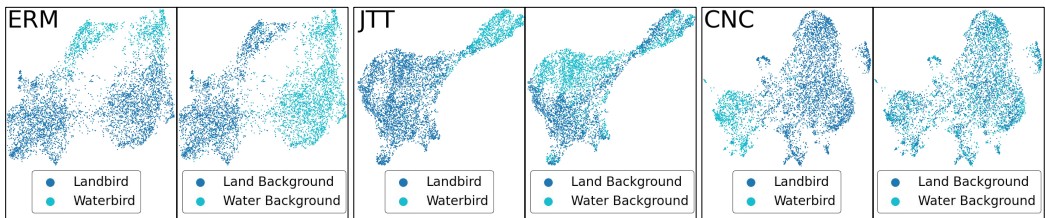

Figure 6: UMAPs of Waterbirds representations, colored by class (left) and spurious attribute (right). ERM depends on both classes $\mathcal{Y}$ and spurious attributes $\mathcal{A}$, though with greater separability for the latter. JTT representations depend more on $\mathcal{Y}$, but also on $\mathcal{A}$. CNC gets closer to fully removing the dependence on $\mathcal{A}$.

We find that CNC representations exhibit the lowest alignment loss consistently for these data sets; this also corresponds to CNC models achieving the highest worst-group accuracy. Furthermore, while all methods result in representations that exhibit high mutual information with the class label (Fig. 5b), only CNC results in representations that drastically reduce mutual information with spurious attributes (Fig. 5c). In Fig. 6, we also illustrate this result on the Waterbirds data set via UMAP visualizations of the learned representations. Notably, all training methods result in representations separable by class label. Yet ERM models exhibit strong separability by spurious attributes, and JTT models interestingly also still depict some learned dependency on the spurious attribute. However, CNC uniquely learns representations that strongly depict class-label-only dependence.

In addition, to study how this relation between representation metrics and worst-group accuracy scales with the strength of the spurious correlation, we compute representation metrics with CNC, ERM, and JTT models trained on increasingly spurious ($\uparrow p_{corr}$) CMNIST* data sets in Fig. 7. We observe that with high spurious correlations, ERM fails to classify digits in the minority classes, while CNC and JTT comparably maintain high worst-group accuracy. CNC also performs better in more spurious settings ($p_{corr} > 0.95$). These improvements over ERM are reflected by drops in alignment loss (averaged over classes); CNC consistently achieves lowest such loss. Fig. 7c shows that CNC's learned representations maintain a more favorable balance of mutual information between the class label and spurious attribute than JTT. While JTT models exhibit slightly higher estimated $I(Y; Z)$ than CNC models, CNC models exhibit much lower dependence on the spurious attribute.

## 5.3 UNDERSTANDING CNC'S SENSITIVITY TO STAGE 1 PREDICTIONS

Finally, we study how sensitive CNC is to how closely the Stage 1 ERM model actually predicts the spurious attribute. As JTT also relies on an initial ERM model's predictions, we compare CNC to JTT in this regard. We find that CNC is more robust to noisy ERM predictions than JTT, and that CNC does not require perfectly inferred groups to perform well.

We first conduct an ablation on CNC and JTT's worst-group and average performance in Fig. 7d with the following synthetic experiment. On CMNIST*, we start with the true spurious attribute labels as the Stage 1 "predictions". We then gradually degrade their quality as follows: for each point, with

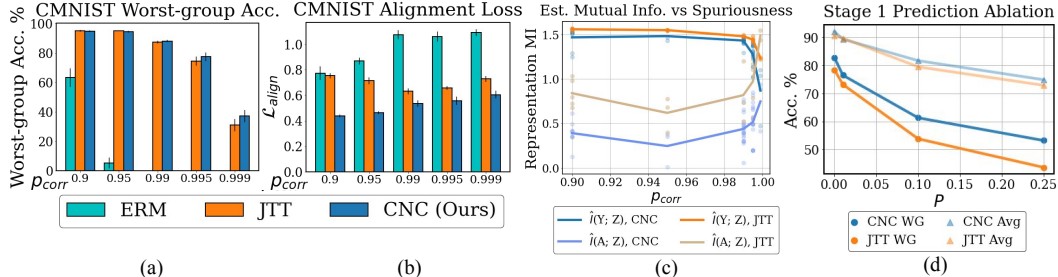

Figure 7: Alignment loss and mutual information representation metrics with worst-group accuracy on increasingly spurious CMNIST*. CNC highest worst-group accuracy (a) coincides with learning representations with better alignment (b) and ratio of mutual information dependence on the labels vs the spurious attribute (c). probability $p$ we change its assigned spurious attribute label to a different label chosen uniformly at random. Both methods' performance degrades as $p$ increases and the Stage 1 "predictions" degrade. However, CNC consistently achieves higher worst-group accuracy and smaller worst-group versus average accuracy gaps. We also observe on other data sets that CNC does not require perfectly spurious ERM predictions to work well. For the Waterbirds and CelebA results in Table 1, the Stage 1 ERM model predictions align with the spurious attribute value $94.7\%$ and $59.3\%$ of the time respectively. While the ERM model is far from perfect at recognizing the spurious attributes, CNC still substantially reduces the worst-group vs. average accuracy gap.

## 6 RELATED WORK

We build on prior work in group robustness and contrastive learning. Further discussion is in App. D.

**Robustness to group shift**. A variety of approaches aim to improve performance on minority data groups. If group labels are known, many works minimize a rebalanced error similar in motivation to correcting class imbalance (He & Garcia, 2009; Cui et al., 2019) or importance weighting (Shimodaira, 2000; Byrd & Lipton, 2019). More recently, Sagawa et al. (2019) minimize worst-group loss during training. Goel et al. (2020) achieve further lift by synthetically generating additional minority group points. Cao et al. (2019) regularize updates on minority groups to improve their generalization.

Another line of work aims to improve group robustness *without* assuming group labels for the training data. The most similar methods to CNC first train an initial ERM model with class labels as a way to infer groups, and then use these groups to train a second model with better worst-group performance. GEORGE (Sohoni et al., 2020) clusters ERM representations, and runs GDRO with these clusters as inferred groups. EIIL (Creager et al., 2021) and PGI (Ahmed et al., 2021) infer groups that maximally violate an invariance objective for the ERM model. With these groups EIIL uses either GDRO or Invariant Risk Minimization (Arjovsky et al., 2019) to train a second robust model, while PGI minimizes the KL divergence of the softmaxed logits for samples in the same class but different groups. LfF (Nam et al., 2020) use a generalized cross-entropy loss to encourage misclassifying minority groups, concurrently training a second model with these datapoints upweighted. JTT (Liu et al., 2021) trains via ERM for a few epochs, before training a second ERM model with incorrect datapoints upsampled. For image data sets, CIM (Taghanaki et al., 2021) trains a transformation network to remove potentially spurious attributes from input features.

**Contrastive learning (CL)**. CL works by predicting whether two inputs are "similar" or "dissimilar" (Le-Khac et al., 2020). This involves specifying batches of *anchor* and *positive* datapoints similar to each other (as different "views" of the same source or input), and *negatives* depicting dissimilar points. An encoder is trained to simultaneously maximize the similarity between the feature representations of anchors and positives, and minimize similarity between anchor and negative representations. In unsupervised CL, "negatives" are often sampled uniformly (Bachman et al., 2019), while "positives" are different views of the same object, e.g. via data augmentation (Chen et al., 2020). In supervised CL, negatives are different-class points and positives are same-class points (Khosla et al., 2020). In CNC, we instead treat same-class points with different ERM predictions as positives, and different-class points with the same ERM prediction as negatives. This naturally provides "hard negative mining," a challenge for standard CL (Robinson et al., 2021; Wu et al., 2021; Chuang et al., 2020).

## 7 CONCLUSION

We present CNC, a two-stage CL approach to learn representations robust to spurious correlations. We theoretically analyze the connection between alignment and worst-group vs. average-group losses, and show that CNC achieves SOTA or near-SOTA worst-group accuracy across several benchmarks.

## ETHICS STATEMENT

We hope that our work is another step towards the important goal of making machine learning models more fair and robust. However, while our work successfully improves worst-group accuracy, this is not necessarily an end-all be-all metric - other fairness-based metrics may be more suitable in certain settings. Also, misuse of metrics could lead to potential harm. To avoid these pitfalls, it is important for practitioners to understand the limitations and tradeoffs of different metrics, including when applying methods such as ours.

## REPRODUCIBILITY STATEMENT

We have submitted our code as part of the supplementary materials. The datasets we use are publicly available (with the exception of CMNIST* which is a modification of the standard MNIST dataset (LeCun et al., 2010); our code to generate this modified dataset is also included). In addition to the details provided in Section 5, further implementation, dataset, and experimental details can be found in Appendix E. For the theory, we include complete proofs of all claims in Appendix B.

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

# A  ADDITIONAL BENCHMARK COMPARISONS AND ABLATIONS

In this section, we include further experiments comparing CNC against additional related methods. We also include additional ablations to study the importance of CNC's presented design choices.

## A.1  COMPARISON TO MINIMIZING THE ALIGNMENT LOSS DIRECTLY

In Sec. 5.1 and Sec. 5.2, we empirically showed that CNC's contrastive loss and hard positive and negative sampling lead to improved worst-group accuracy and greater representation alignment. We now study how CNC performs if instead of the contrastive loss, we train the Stage 2 model to minimize $\mathcal{L}_{\text{align}}$ directly. With this objective, we aim to minimize the Euclidean distance between samples in different inferred groups but the same class. We keep all other components of CNC consistent, and apply $\mathcal{L}_{\text{align}}$ to the anchor and positive samples in each contrastive batch. We report results on CMNIST*, Waterbirds, and CelebA in Table A.1.

Table A.1: Across benchmarks, CNC achieves higher worst-group and average accuracies with the default contrastive loss, compared to using the alignment loss explicitly as a training objective.

| Method | CMNIST* | | Waterbirds | | CelebA | |
|---|---|---|---|---|---|---|
| Accuracy (%) | Worst-group | Average | Worst-group | Average | Worst-group | Average |
| CNC ($\mathcal{L}_{\text{align}}$) | 67.3 (3.1) | 83.7 (1.1) | 83.3 (0.8) | 89.4 (0.2) | 84.6 (0.5) | 88.0 (0.3) |
| CNC (default contrastive) | **77.4 (3.0)** | **90.9 (0.6)** | **89.7 (0.2)** | **90.8 (0.1)** | **88.8 (0.9)** | **89.9 (0.5)** |

We find that CNC with the default contrastive loss outperforms CNC with the alignment loss. We reason that an advantage of the contrastive loss (and specifically the "hard" positive and negative samples), is that it encourages aligning samples with the same class label but different spurious features, *and* pushes apart hard negative samples with different class labels but similar spurious features. This provides additional signal for improving separation between the different classes, so the robust model only learns to rely on ground-truth-specific features for discriminating between datapoints. On the other hand, the $\mathcal{L}_{\text{alignment}}$ objective does not incorporate these hard negatives.

## A.2  COMPARISON TO REPRESENTATION ALIGNMENT METHODS FOR DOMAIN GENERALIZATION AND ADAPTATION

While our main results in Table 1 compare against methods designed to tackle the spurious correlations setting presented in Section 5.1, we now study how CNC fares against existing representation alignment methods proposed in the domain generalization (DG) and unsupervised domain adaptation (UDA) literature. At a high level, a popular idea in DG and UDA is to learn similar representations for datapoints with the same class but sampled from different domains, e.g. via adversarial training to prevent another model from classifying representations' source domains correctly (Ganin et al., 2016), or minimizing representation differences via metrics such as *maximum mean discrepancy* (MMD) (Li et al., 2018). While DG and UDA carry distinct problem settings and assumptions from our spurious correlations setting (c.f. Appendix D.4), we aim to understand if existing representation alignment methods can train models robust to spurious correlations, and compare their performance with CNC. We first explain our protocol for evaluating these methods, and then discuss results.

We carry out our evaluation with domain-adversarial neural networks (DANN) Ganin et al. (2016), a seminal UDA method that aims to learn aligned representations across two domains. To do so, DANN jointly trains a model to classify samples from a "source" domain while preventing a separate "domain classifier" module from correctly classifying the domain for datapoints sampled from both domains. For fair comparison, we use the same ResNet-50 backbone as in CNC, and make several adjustments to the typical DANN and UDA procedure:

1. While UDA assumes that the data is organized into "source" and "target" domains, we do not have domain labels. We thus infer domains using the predictions of an initial ERM model as in CNC.

2. The notion of a domain may also be ambiguous with respect to the groups defined in Section 2. For example, domains may be defined by spurious attributes (e.g., for the Waterbirds dataset, we

may consider the "water background" domain and the "land background" domain). Domains may alternatively be defined by whether samples carry dominant spurious correlations or not (e.g., the "majority group" domain and the "minority group" domain). We train and evaluate separate DANN models for both interpretations. We infer the former by the predicted class of the initial ERM model. We infer the latter by whether the initial ERM model is correct or not.

3. Finally, UDA aims to train with a class-labeled "source" domain and an unlabeled "target" domain such that a model performs well on unseen samples from the specified "target" domain (Ganin et al., 2016). However, our benchmarks have class labels for *all* training points, and do not have a notion of "source" and "target" domains (we aim to obtain high worst-group accuracy, which could fall under any domain). We thus assume access to labels for all domains. During training, the goal for our DANN models is to correctly classify samples from both domains, while learning representations such that a jointly trained domain classifier module cannot determine the samples' domains from their representations alone. At test-time, we evaluate the DANN model on the entire test set for each benchmark, and report the worst-group and average accuracies.

Table A.2: CNC achieves higher worst-group and average accuracies on spuriously correlated benchmarks than DANN, a prior representation alignment method designed for domain adaptation

| Method | Waterbirds | | CelebA | |
|---|---|---|---|---|
| Accuracy (%) | Worst-group | Average | Worst-group | Average |
| DANN (domains by spurious attribute) | 37.4 (3.8) | 87.6 (2.2) | 28.1 (3.1) | 94.6 (0.3) |
| DANN (domains by majority vs minority group) | 67.3 (0.8) | 83.6 (0.2) | 47.2 (3.1) | 88.7 (1.8) |
| CNC | **89.7 (0.2)** | **90.8 (0.1)** | **88.8 (0.9)** | **89.9 (0.5)** |

In Table A.2, we report the worst-group and average accuracies of DANN on the Waterbirds and CelebA datasets across three seeds along with the CNC results. Our results suggest that the domain alignment in DANN is not sufficient to improve worst-group accuracy. We hypothesize this is due to adversarial training with the domain classifier aligning representations without regard to different classes within each domain. Due to the propensity of samples exhibiting spurious correlations, DANN models may thus still learn to rely on these correlations.

## A.3  IMPORTANCE OF ERM-GUIDED CONTRASTIVE SAMPLING

In this section we conduct additional ablations on the sampling procedure in CNC. Although CNC relies on an initial trained ERM model's predictions, can we still improve worst-group accuracy without this step and with supervised contrastive learning alone, i.e. by sampling positives uniform randomly from *all* datapoints with the same label as the anchor? In Table 1, we showed that this approach (denoted SupCon*) led to a drop in worst-group accuracy. Taking this question further, while we use the Stage 1 ERM model's predictions to sample "hard" negatives with different ground-truth classes *and* the same ERM predictions as their anchors—such that to reduce the contrastive loss and learn dissimilar representations for anchors and negatives, the Stage 2 contrastive model must thus learn to ignore spurious features that the initial ERM model learns to depend on—how does CNC's performance fare with alternative negative sampling procedures? Keeping the anchor and positive sampling consistent, we perform additional ablations where we either sample negatives only by having different classes as their anchors, or sample negatives only be having the same ERM model prediction as their anchors. We report these results in Table A.3 below.

We find that the default CNC sampling procedure obtains highest worst-group accuracy and highest or near-highest average accuracy compared to alternative strategies across the CMNIST*, Waterbirds, and CelebA datasets. The results suggests that inferring the spurious attributes (e.g. via an initial ERM model) is important for CNC, and that CNC benefits from using these predictions for sampling both negatives and positives. We reason this is because without this sampling, we can actually encourage the Stage 2 model to rely on spurious correlations. For example, if we just ensure that the anchor and negative samples have different classes, then the contrastive model may just rely on the different spurious features of the anchors and negatives to learn dissimilar representations. However, by ensuring that the anchors and negatives have similar spurious features (via the same trained ERM model prediction), the contrastive model is forced to rely on non-spurious features to learn dissimilar

Table A.3: Ablation on positive and negative sampling strategies in CNC. CNC achieves highest worst-group accuracy when using the Stage 1 ERM model's predictions to sample "hard" positives *and* negatives (the default procedure).

| Method | CMNIST* | | Waterbirds | | CelebA | |
|---|---|---|---|---|---|---|
| Accuracy (%) | Worst-group | Average | Worst-group | Average | Worst-group | Average |
| Negatives by different class | 66.4 (5.1) | 86.0 (1.6) | 82.2 (0.8) | 88.9 (0.3) | 79.2 (0.3) | 88.0 (0.1) |
| Negatives by same prediction | 70.0 (5.1) | 87.1 (1.1) | 85.7 (1.3) | 90.3 (0.2) | 81.1 (1.4) | 88.5 (0.3) |
| SupCon* | 0.0 (0.0) | 22.4 (1.2) | 71.0 (1.9) | 85.9 (0.8) | 62.2 (1.1) | **90.0 (0.1)** |
| CNC (default) | **77.4 (3.0)** | **90.9 (0.6)** | **89.7 (0.2)** | **90.8 (0.1)** | **88.8 (0.9)** | 89.9 (0.5) |

representations for the samples. The same logic applies for learning similar representations for anchor and positive samples. We suspect that choosing negatives from all samples with the same ERM prediction as their anchors performs better than the other ablations as it alone does not encourage learning spurious correlations: the model is asked to "pull apart" samples with the same spurious features, and so must ignore spurious similarities to recognize something different between anchors and negatives. However, this ablation does not ensure that anchor-negative pairs consist of different classes (which our full method does), so the model gets less signal to separate samples by class.

## A.4 ADDITIONAL DESIGN CHOICE ABLATIONS

We first summarize CNC's design choices and differences from standard supervised contrastive learning in Appendix A.4.1. We then empirically validate each component in Appendix A.4.2.

### A.4.1 SUMMARY OF CNC DESIGN CHOICES AND PROPERTIES

**No projection network.** As we wish to learn data representations that maximize the alignment between anchor and positive datapoints, we do not compute the contrastive loss with the outputs of an additional nonlinear projection network. This is inspired by the logic justifying a projection head in prior contrastive learning, e.g. SimCLR (Chen et al., 2020), where the head is included because the contrastive loss trains representations to be "invariant to data transformation" and may encourage removing information "such as the color or orientation of objects". In our case, we view inferred datapoints with the same class but different spurious attributes as "transformations" of each other, and we hypothesize that removing these differences can help us improve worst-group performance.

**Two-sided contrastive sampling.** To incorporate additional comparisons between datapoints that only differ in spurious attribute during training, we employ "two-sided" contrastive batch sampling. This lets us equally incorporate instances where the second contrastive model in CNC treats datapoints that the initial ERM model got incorrect and correct as anchors.

**Additional intrinsic hard positive/negative mining.** Because the new model corrects for potentially learned spurious correlations by only comparing and contrasting datapoints that differ in class label or spurious attribute, but not both (as dictated by the initial ERM model's outputs), the contrastive batches naturally carry "hard" positives and negatives. Thus, our approach provides a natural form of hard negative mining (in addition to the intrinsic hard positive / negative mining at the gradient level with InfoNCE-style contrastive losses (Chen et al., 2020; Khosla et al., 2020)) while avoiding class collisions, two nontrivial challenges in standard self-supervised contrastive learning (Robinson et al., 2021; Wu et al., 2021; Chuang et al., 2020).

**Joint training of encoder and classifier layers.** CNC can train any standard classification model architecture; for any given neural network we just apply different optimization objectives to the encoder and classifier layers. We train both the encoder and classifier layers with a cross-entropy loss, and jointly train the encoder layer with a supervised contrastive loss. For the encoder layers, we balance the two objectives with a hyperparameter $\lambda$ (c.f. Eq. 7).

### A.4.2 EMPIRICAL VALIDATION OF CNC COMPONENTS

To validate the additional algorithmic components of CNC, we report how CNC performs on the Waterbirds dataset when modifying the individual design components. We use the same hyperpa-

rameters as in the main results, and report accuracies as the average over three training runs for the following ablations. Table A.4 summarizes that across these design ablations, default CNC as presented consistently outperforms these alternative implementations.

Table A.4: Ablation over CNC algorithmic components on Waterbirds. Default choices achieve highest worst-group and average accuracy.

| Method | CNC (Default) | Projection Head | One-sided Contrasting | Train + Finetune |
|---|---|---|---|---|
| WG Acc. (%) | **89.7** (0.2) | 82.4 (1.8) | 85.2 (3.6) | 84.0 (1.7) |
| Avg. Acc. (%) | **90.8** (0.1) | 88.7 (0.6) | 90.1 (1.6) | 87.7 (1.1) |

**No projection head.** We incorporate a nonlinear projection head as is typical in prior contrastive learning works (Chen et al., 2020), that maps the encoder output to lower-dimensional representations (from $2048$ to $128$ in our case). We then update the encoder layers and the projection head jointly by computing the contrastive loss on the projection head's output, still passing the encoder layer's direct outputs to the classifier to compute the cross-entropy loss. We note that using the projection head decreases worst-group accuracy substantially. We reason that as previously discussed, while using the projection head in prior work can allow the model to retain more information in its actual hidden layers (Chen et al., 2020), in our case to remove dependencies on spurious attributes we actually want to encourage learning invariant representations when we model the differences between anchor and positive datapoints as due to spurious attributes.

**Two-sided contrastive batches.** Instead of "two-sided" contrasting where we allow both sampled anchors and positives to take on the anchor role, for each batch we only compute contrastive updates by comparing original positives and negatives with the original anchor. When keeping everything else the same, we find that just doing these one-sided comparisons also leads to a drop in performance for worst-group accuracy. This suggests that the increased number of comparisons and training setup where we swap the roles of anchors and positives of the two-sided batches introduces greater contrastive learning signal.

**Additional intrinsic hard positive/negative mining.** We discuss this ablation in Section A.3.

**Joint training of encoder and classifier layers.** Instead of training the full model jointly, we first only train the encoder layers with the contrastive loss in CNC, before freezing these layers and finetuning the classifier layers with the cross-entropy loss. With this implementation, we also obtain noticeable drop in performance. While we leave further analysis for the joint cross-entropy and contrastive optimization for future work, one conjecture is that the cross-entropy loss may aid in learning separable representations while also training the full model to keep the average error small. From our theory, the contrastive loss can help bound the gap between worst-group and average error. Thus we try to minimize average error in the same parameter update.

This also follows prior work, where updating the entire model and finetuning all model parameters instead of freezing the encoder layers leads to higher accuracy (Chen et al., 2020). However, we found that with an initial encoder-only training stage, if we did not freeze the trained layers the fine-tuning on a dataset with spurious correlations would "revert" the contrastive training, resulting in a large gap between worst-group and average error similar to ERM.

We also ablate the balancing hyperparameter $\lambda$ of CNC on CMNIST*. In Table A.5 we find that CNC consistently achieves high worst-group accuracy across a wide range of $\lambda \in [0.4, 0.9]$. For reference, the next best methods GEORGE and JTT obtain $76.4\%$ and $74.5\%$ worst-group accuracy.

Table A.5: Ablation over CNC $\lambda$ parameter to balance cross-entropy and contrastive loss components on CMNIST*. CNC obtains high performance across a range of $\lambda$.

| CNC $\lambda$ | 0.2 | 0.4 | 0.6 | 0.8 | 0.9 |
|---|---|---|---|---|---|
| Robust Acc. | 70.4 (2.9) | 74.2 (2.6) | 75.3 (1.7) | 77.4 (2.5) | 75.8 (1.2) |
| Average Acc. | 89.0 (0.1) | 88.0 (0.7) | 88.3 (0.6) | 89.9 (0.4) | 88.4 (0.1) |

# B    OMITTED PROOFS FROM SECTION 3.2

In this section, we prove that within any class, the gap between the worst-group error and the average error can be upper bounded by the alignment loss times the Lipschitz constant, plus another concentration error term.

*Proof of Theorem 3.1.* Consider two arbitrary groups, denoted by $g_1 = (y, a_1)$ and $g_2 = (y, a_2)$, whose class labels are both $y \in \mathcal{Y}$, whose spurious attributes are $a_1 \in \mathcal{A}$ and $a_2 \in \mathcal{A}$ such that $a_1 \neq a_2$. Let $G_1$ and $G_2$ be the subset of training data that belong to groups $g_1$ and $g_2$, respectively. We note that both $G_1$ and $G_2$ are non-empty since we have assumed that (in Section 2) there is at least one sample from each group in the training data set. Let $n_{g_1} = |G_1|$ and $n_{g_2} = |G_2|$ be the size of these two groups, respectively. Recall that $f_{\text{enc}}$ denotes the mapping of the encoder layers of the full neural network model $f_\theta$. Since the classification layer $f_{\text{cls}}$ is a linear layer, we have used $W$ to denote the weight matrix of this layer. Our definition of the cross-group alignment loss in equation (5), denoted as $\hat{\mathcal{L}}_{\text{align}}(f_\theta; y)$, implies that for $g_1$ and $g_2$,

$$\frac{1}{n_{g_1}} \frac{1}{n_{g_2}} \sum_{(x,y,a_1) \in G_1} \sum_{(x',y,a_2) \in G_2} \|f_{\text{enc}}(x) - f_{\text{enc}}(x')\|_2 \leq \hat{\mathcal{L}}_{\text{align}}(f_\theta; y). \tag{8}$$

Next, let $\mathbb{E}_{(x,y,a_1) \sim \mathcal{P}_{g_1}} [\mathcal{L}_{\text{avg}}(W f_{\text{enc}}(x), y)]$ be the average loss conditioning on a data point being sampled from group $g_1$ (and similarly for group $g_2$). Let $\Delta(g_1, g_2)$ be the difference between the population average losses:

$$\Delta(g_1, g_2) = \left| \mathbb{E}_{(x,y,a_1) \sim \mathcal{P}_{g_1}} [\mathcal{L}_{\text{avg}}(W f_{\text{enc}}(x), y] - \mathbb{E}_{(x,y,a_2) \sim \mathcal{P}_{g_2}} [\mathcal{L}_{\text{avg}}(W f_{\text{enc}}(x), y)] \right|.$$

Recall that $\mathcal{G}_y \subseteq \mathcal{G}$ is the set of groups that have class label $y$. Since the loss $\ell(\cdot)$ is bounded above by some fixed constant $C_2$ according to our assumption, and is at least zero, by the Hoeffding's inequality, the following result holds with probability at least $1 - \delta$, for all $|\mathcal{G}_y|$ groups $g \in \mathcal{G}_y$,

$$\left| \mathbb{E}_{(x,y,a) \sim \mathcal{P}_g} [\mathcal{L}_{\text{avg}}(W f_{\text{enc}}(x), y)] - \frac{1}{n_g} \sum_{(x,y) \in (X,Y)} \ell(W f_{\text{enc}}(x), y) \right| \leq C_2 \sqrt{\frac{2 \log (|\mathcal{G}_y| / \delta)}{n_g}}. \tag{9}$$

Thus, with probability at least $1 - \delta$, the following holds for any $g_1$ and $g_2$ in class $y$ (but having different spurious attributes)

$$\Delta(g_1, g_2) \leq \left| \frac{1}{n_{g_1}} \sum_{(x,y,a_1) \in G_1} \mathcal{L}_{\text{avg}}(W f_{\text{enc}}(x), y) - \frac{1}{n_{g_2}} \sum_{(x',y,a_2) \in G_2} \mathcal{L}_{\text{avg}}(W f_{\text{enc}}(x'), y) \right| \tag{10}$$

$$+ C_2 \left( \sqrt{\frac{2 \log(|\mathcal{G}_y| / \delta)}{n_{g_1}}} + \sqrt{\frac{2 \log(|\mathcal{G}_y| / \delta)}{n_{g_2}}} \right).$$

Next, we focus on the RHS of equation (10). First, equation (10) is also equal to the following:

$$\left| \frac{1}{n_{g_1}} \frac{1}{n_{g_2}} \sum_{(x,y,a_1) \in G_1} \sum_{(x',y,a_2) \in G_2} \ell(W f_{\text{enc}}(x), y)) - \frac{1}{n_{g_1}} \frac{1}{n_{g_2}} \sum_{(x,y,a_1) \in G_1} \sum_{(x',y,a_2) \in G_2} \ell(W f_{\text{enc}}(x'), y)) \right|.$$

Since we have also assumed that the loss function $\ell(x, y)$ is $C_1$-Lipschitz in $x$[2], the above is at most:

$$\left| \frac{1}{n_{g_1} n_{g_2}} \sum_{(x,y,a_1) \in G_1} \sum_{(x',y,a_2) \in G_2} |\ell(W f_{\text{enc}}(x), y) - \ell(W f_{\text{enc}}(x'), y)| \right|$$

$$\leq \frac{1}{n_{g_1} n_{g_2}} \sum_{(x,y,a_1) \in G_1} \sum_{(x',y,a_2) \in G_2} C_1 \cdot \|W f_{\text{enc}}(x) - W f_{\text{enc}}(x')\|_2 \quad \text{(since } y \text{ is the same for } x, x')$$

$$\leq \frac{B}{n_{g_1} n_{g_2}} \sum_{(x,y,a_1) \in G_1} \sum_{(x',y,a_2) \in G_2} C_1 \cdot \|f_{\text{enc}}(x) - f_{\text{enc}}(x')\|_2 \quad \text{(because } \|W\|_2 \leq B \text{ as assumed)}$$

$$\leq B \cdot C_1 \cdot \hat{\mathcal{L}}_{\text{align}}(f_\theta; y). \quad \text{(because of equation (8))}$$

---

[2] In other words, we assume that $|\ell(z, y) - \ell(z', y)| \leq C_1 \cdot \|z - z'\|_2$, for any $z, z'$ and $y$.

Thus, we have shown that for any $g_1$ and $g_2$ within class $y$,

$$\Delta(g_1, g_2) \leq B \cdot \hat{\mathcal{L}}_{\text{align}}(f_\theta; y) + \left( \sqrt{\frac{2\log(|\mathcal{G}_y|/\delta)}{n_{g_1}}} + \sqrt{\frac{2\log(|\mathcal{G}_y|/\delta)}{n_{g_2}}} \right)$$

$$\leq B \cdot C_1 \cdot \hat{\mathcal{L}}_{\text{align}}(f_\theta; y) + \max_{g \in \mathcal{G}_y} C_2 \cdot \sqrt{\frac{8\log(|\mathcal{G}_y|/\delta)}{n_g}}. \tag{11}$$

Finally, we use the above result to bound the gap between the worst-group loss and the average loss. For every group $g \in \mathcal{G}$, let $p_g$ denote the prior probability of observing a sample from $\mathcal{P}$ in this group. Let $q_y = \sum_{g' \in \mathcal{G}_y} p_{g'}$. Let $h(g)$ be a short hand notation for

$$h(g) = \mathbb{E}_{(x,y,a) \sim \mathcal{P}_g} [\mathcal{L}_{\text{avg}}(W f_{\text{enc}}(x), y)].$$

The average loss among the groups with class label $y$ is $\mathcal{L}_{\text{avg}}(f_\theta; y) = \sum_{g \in \mathcal{G}_y} \frac{p_g}{q_y} h(g)$. The worst-group loss among the groups with class label $y$ is $\mathcal{L}_{\text{wg}}(f_\theta; y) = \max_{g \in \mathcal{G}_y} h(g)$. Let $g^\star$ be a group that incurs the highest loss among groups in $\mathcal{G}_y$. We have $\mathcal{L}_{\text{wg}}(f_\theta; y) - \mathcal{L}_{\text{avg}}(f_\theta; y)$ is equal to

$$h(g^\star) - \sum_{g \in \mathcal{G}_y} \frac{p_g}{q_y} h(g) = \sum_{g \in \mathcal{G}_y} \frac{p_g}{q_y} (h(g^\star) - h(g)) \tag{12}$$

$$\leq \sum_{g \in \mathcal{G}_y} \frac{p_g}{q_y} \Delta(g^\star, g) \tag{13}$$

$$\leq B \cdot C_1 \cdot \hat{\mathcal{L}}_{\text{align}}(f_\theta; y) + \max_{g \in \mathcal{G}_y} C_2 \cdot \sqrt{\frac{8\log(|\mathcal{G}|/\delta)}{n_g}}. \tag{14}$$

The last step uses equation (11) on $\Delta(g^\star, g)$ and the fact that $q_y = \sum_{g' \in \mathcal{G}_y} p_{g'}$. Thus, we have shown that the gap between the worst-group loss and the average loss among the groups with the same class label is bounded by the above equation. The proof is now complete. $\square$

The astute reader will note that Theorem 3.1 focuses on comparing groups within the same class $y$, for any $y \in \mathcal{Y}$. A natural follow-up question is what happens when comparing across groups with different labels. Let $\mathcal{L}_{\text{wg}}(f_\theta) = \max_{y \in \mathcal{Y}} \mathcal{L}_{\text{wg}}(f_\theta; y)$ be the worst-group loss across all the labels. Recall that $\mathcal{L}_{\text{avg}}(f_\theta)$ is the average loss for the entire population of data. We generalize Theorem 3.1 to this setting in the following result.

**Corollary B.1** (Extension of Theorem 3.1 to compare across different classes). *In the setting of Theorem 3.1, let $q_y = \sum_{g \in \mathcal{G}_y} p_g$ be the prior probability of observing a sample drawn from $\mathcal{P}$ with label $y$, for any $y \in \mathcal{Y}$. We have that with probability at least $1 - \delta$, the following holds:*

$$\mathcal{L}_{wg}(f_\theta) \leq \left( \min_{y \in \mathcal{Y}} q_y \right)^{-1} \mathcal{L}_{avg}(f_\theta) + B \cdot C_1 \cdot \max_{y \in \mathcal{Y}} \hat{\mathcal{L}}_{align}(f_\theta; y) + \max_{g \in \mathcal{G}} C_2 \cdot \sqrt{\frac{8\log(|\mathcal{G}|/\delta)}{n_g}}. \tag{15}$$

*Proof.* We generalize the argument in the previous result to compare across different labels. The worst-group loss across different labels is

$$\max_{y \in \mathcal{Y}} \max_{g \in \mathcal{G}_y} h(g)$$

$$\leq \max_{y \in \mathcal{Y}} \left( \sum_{g \in \mathcal{G}_y} \frac{p_g}{q_y} h(g) + B \cdot C_1 \hat{\mathcal{L}}_{\text{align}}(f_\theta; y) + \max_{g \in \mathcal{G}_y} C_2 \sqrt{\frac{8\log(|\mathcal{G}_y|/\delta)}{n_g}} \right) \quad \text{(because of equation (14))}$$

$$\leq \frac{1}{\min_{y \in \mathcal{Y}} q_y} \sum_{g \in \mathcal{G}_y} p_g h(g) + B \cdot C_1 \max_{y \in \mathcal{Y}} \hat{\mathcal{L}}_{\text{align}}(f_\theta; y) + \max_{g \in \mathcal{G}} C_2 \sqrt{\frac{8\log(|\mathcal{G}|/\delta)}{n_g}}.$$

Since $\sum_{g \in \mathcal{G}} p_g h(g) = \mathcal{L}_{\text{avg}}(f_\theta)$, we thus conclude that

$$\mathcal{L}_{\text{wg}}(f_\theta) \leq \left( \min_{y \in \mathcal{Y}} q_y \right)^{-1} \mathcal{L}_{\text{avg}}(f_\theta) + B \cdot C_1 \max_{y \in \mathcal{Y}} \hat{\mathcal{L}}_{\text{align}}(f_\theta; y) + \max_{g \in \mathcal{G}} C_2 \sqrt{\frac{8\log(|\mathcal{G}|/\delta)}{n_g}}.$$

The proof is now complete. $\square$

**An example showing that Corollary B.1 is tight.** We describe a simple example in which the factor $\left(\min_{y \in \mathcal{Y}} q_y\right)^{-1}$ in equation (15) is tight (asymptotically). Suppose there are $k$ perfectly balanced classes so that $q_y = 1/k$, for every $y \in \mathcal{Y}$. There is one data point from each class, with loss equal to 0 for all except one of them. The worst-group loss is 1 whereas the average loss is $1/k$. Thus, there is a factor of $k$ between the worst-group loss and the average loss. For equation (15), the factor

$$\left(\min_{y \in \mathcal{Y}} q_y\right)^{-1} = k,$$

since $q_y = 1/k$ for every $y \in \mathcal{Y}$ in this example. Thus, this factor matches the (multiplicative) factor between the worst-group loss and the average loss in this example.

## C   CONTRASTIVE ALGORITHM DESIGN DETAILS

In this section, we provide further details on the training setup and contrastive batch sampling, pseudocode, and additional properties related to CNC's implementation.

### C.1   TRAINING SETUP

In Fig. 8, we illustrate the two training stages of Correct-N-Contrast described in Sec. 4. In Stage 1, we first train an ERM model with a cross-entropy loss. For consistency with Stage 2, we depict the output as a composition of the encoder and linear classifier layers. Then in Stage 2, we train a new model with the same architecture using contrastive batches sampled with the Stage 1 ERM model and a supervised contrastive loss (3) (which we compute after the depicted representations are first normalized) to update the encoder layers. Note that unlike prior work in contrastive learning (Chen et al., 2020; Khosla et al., 2020), as we have the class labels of the anchors, positives, and negatives, we also continue forward-passing the unnormalized representations (encoder layer outputs) and compute a cross-entropy loss to update the classifier layers while jointly training the encoder.

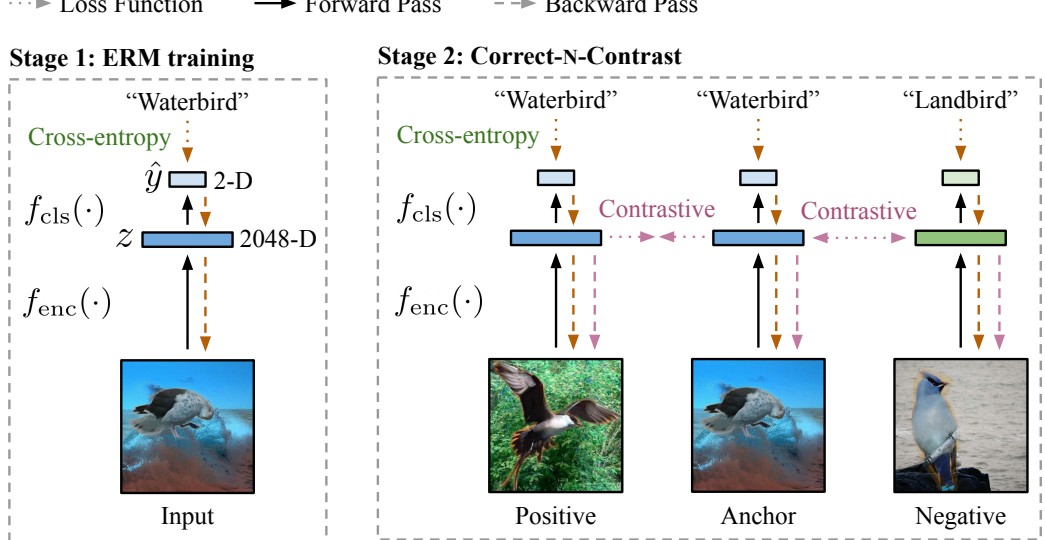

Figure 8: The two stages of Correct-N-Contrast. In Stage 1, we train a model with standard ERM and a cross-entropy loss. Then in Stage 2, we train a new model with the same architecture, but specifically learn spurious-attribute-invariant representations with a contrastive loss (3) and batches of anchors, positives, and negatives sampled with the ERM model's predictions. We also update the full model jointly with a cross-entropy loss on the classifier layer output and the input class labels. Dimensions for ResNet-50 and Waterbirds.

We also note that unlike prior work, we wish to learn invariances between anchors and positives that maximally reduce the presence of features not needed for classification. We thus do not pass

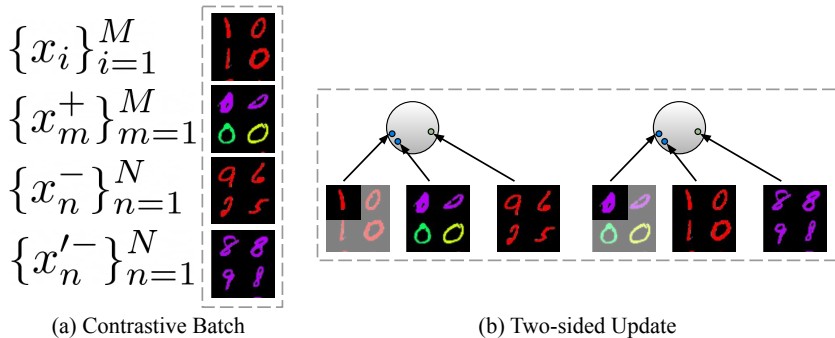

$$\{x_i\}_{i=1}^M$$
$$\{x_m^+\}_{m=1}^M$$
$$\{x_n^-\}_{n=1}^N$$
$$\{x_n'^-\}_{n=1}^N$$

(a) Contrastive Batch         (b) Two-sided Update

Figure 9: Illustration of two-sided contrastive batch sampling with Colored MNIST as an example. From a single batch (a), we can train a contrastive model with two anchor-positive-negative pairings (b). Aside from increasing the number of "hard negatives" for each anchor-positive pair, this intuitively "pushes" together anchors and positives from two different directions for greater class separation.

the representations through an additional *projection network* (Chen et al., 2020). Instead, we use Eq. 3 to compute the supervised contrastive loss directly on the encoder outputs $z = f_{\text{enc}}(x)$. In Appendix A.4.2, we studied ablations with both design choices.

## C.2 TWO-SIDED CONTRASTIVE BATCH IMPLEMENTATION

We provide more details on our default contrastive batch sampling approach described in Sec. 4. To recall, for additional contrastive signal per batch, we can double the pairwise comparisons in a training batch by switching the anchor and positive roles. This is similar to the *NT-Xent* loss in prior contrastive learning work (Chen et al., 2020). We switch the role of the anchor and first positive sampled in a contrastive batch, and sample additional positives and negatives using the same guidelines but adjusting for the "new" anchor. We denote this as "two-sided" sampling in contrast with the "one-sided" comparisons we get with just the original anchor, positives, and negatives.

Implementing this sampling procedure in practice is simple. First, recall our initial setup with trained ERM model $f_{\hat{\theta}}$, its predictions $\{\hat{y}_i\}_{i=1}^n$ on training data $\{(x_i, y_i)\}_{i=1}^n$ (where $\hat{y}_i = f_{\hat{\theta}}(x_i)$), and number of positives and negatives to sample $M$ and $N$. We then sample batches with Algorithm 2.

Because the initial anchors are then datapoints that the ERM model gets correct, under our heuristic we infer $\{x_i\}_{i=1}^M$ as samples from the majority group. Similarly the $M$ positives $\{x_m^+\}_{m=1}^M$ and $N$ negatives $\{x_n^-\}_{n=1}^N$ that it gets incorrect are inferred to belong to minority groups.

For one batch, we then compute the full contrastive loss with

$$\hat{\mathcal{L}}_{\text{con}}^{\text{sup}}(f_{\text{enc}}) = \hat{\mathcal{L}}_{\text{con}}^{\text{sup}}\left(x_1, \{x_m^+\}_{m=1}^M, \{x_n^-\}_{n=1}^N; f_{\text{enc}}\right) + \hat{\mathcal{L}}_{\text{con}}^{\text{sup}}\left(x_1^+, \{x_i\}_{i=1}^M, \{x_n'^-\}_{n=1}^N; f_{\text{enc}}\right) \quad (16)$$

where $\hat{\mathcal{L}}_{\text{con}}^{\text{sup}}\left(x_1, \{x_m^+\}_{m=1}^M, \{x_n^-\}_{n=1}^N; f_{\text{enc}}\right)$ is given by:

$$-\frac{1}{M}\sum_{m=1}^M \log \frac{\exp(z_1^\top z_m^+/\tau)}{\sum_{m=1}^M \exp(z_1^\top z_m^+/\tau) + \sum_{n=1}^N \exp(z_1^\top z_n^+/\tau)} \quad (17)$$

---

**Algorithm 2** Sampling two-sided contrastive batches

**Require:** Number of positives $M$ and number of negatives $N$ to sample for each batch.

1: Initialize set of contrastive batches $B = \{\}$
2: **for each** $x_i \in \{x_i \in X : \hat{y}_i = y_i\}$ **do**
3:      Sample $M - 1$ additional "anchors" to obtain $\{x_i\}_{i=1}^M$ from $\{x_i \in X : \hat{y}_i = y_i\}$
4:      Sample $M$ positives $\{x_m^+\}_{m=1}^M$ from $\{x_m^- \in X : \hat{y}_m^- = \hat{y}_i, \ y_m^- \neq y_i\}$
5:      Sample $N$ negatives $\{x_n^-\}_{n=1}^N$ from $\{x_n^- \in X : \hat{y}_n^- = \hat{y}_i, \ y_n^- \neq y_i\}$
6:      Sample $N$ negatives $\{x_n'^-\}_{n=1}^N$ from $\{x_n'^- \in X : \hat{y}_n'^- = \hat{y}_1^+, y_n'^- \neq y_1^+\}$
7:      Update contrastive batch set: $B \leftarrow B \cup \left(\{x_i\}_{i=1}^M, \{x_m^+\}_{m=1}^M, \{x_n^-\}_{n=1}^N, \{x_n'^-\}_{n=1}^N\right)$

---

and again let $z$ be the normalized output $f_{\text{enc}}(x)$ for corresponding $x$. We compute the cross-entropy component of the full loss for each $x$ in the two-sided batch with its corresponding label $y$.

# D  FURTHER RELATED WORK DISCUSSION

We provide additional discussion of related work and connections to our work below.

## D.1  IMPROVING ROBUSTNESS TO SPURIOUS CORRELATIONS

Our core objective is to improve model robustness to group or subpopulation distribution shifts that arise from the presence of spurious correlations, specifically for classification tasks. Because these learnable correlations hold for some but not all samples in a dataset, standard training with ERM may result in highly variable performance: a model that classifies datapoints based on spurious correlations does well for some subsets or "groups" of the data but not others. To improve model robustness and avoid learning spurious correlations, prior work introduces the goal to maximize worst-group accuracy (Sagawa et al., 2019). Related works broadly fall under two categories:

**Improving robustness with group information.** If information such as spurious attribute labels is provided, one can divide the data into explicit groups as defined in Sec. 2, and then train to directly minimize the worst group-level error among these groups. This is done in group DRO (GDRO) (Sagawa et al., 2019), where the authors propose an online training algorithm that focuses training updates over datapoints from higher-loss groups. Goel et al. (2020) also adopt this approach with their method CycleGAN Augmented Model Patching (CAMEL). However, similar to our motivation, they argue that a stronger modeling goal should be placed on preventing a model from learning group-specific features. Their approach involves first training a CycleCAN (Zhu et al., 2017) to learn the data transformations from datapoints in one group to another that share the same class label. They then apply these transformations as data augmentations to different samples, intuitively generating new versions of the original samples that take on group-specific features. Finally they train a new model with a consistency regularization objective to learn invariant features between transformed samples and their sources. Unlike their consistency loss, we accomplish a similar objective to learn group-invariant features with contrastive learning. Our first training stage is also less expensive. Instead of training a CycleGAN and then using it to augment datapoints, we train a relatively simple standard ERM classification model, sometimes with only a few number of epochs, and use its predictions to identify pairs of datapoints to serve a similar purpose. Finally, unlike both CAMEL and GDRO, we do not require spurious attribute or group labels for each training datapoints. We can then apply CNC in less restrictive settings where such information is not known.

Related to GDRO are methods that aim to optimize a "Pareto-fair" objective, more general than simply the worst-case group performance. Notable examples are the works of Balashankar et al. (2019) and Martinez et al. (2020). However, these approaches similarly do not directly optimize for good representation alignment (unlike our work).

**Improving robustness without training group information.** More similar to our approach are methods that do not assume group information at training time, and only require validation set spurious attribute labels for fine-tuning. As validation sets are typically much smaller in size than training sets, an advantage of CNC and comparable methods is that we can improve the accessibility of robust training methods to a wider set of problems. One popular line of work is distributionally robust optimization (DRO), which trains models to minimize the worst loss within a ball centered around the observed distribution (Ben-Tal et al., 2013; Wiesemann et al., 2014; Duchi & Namkoong, 2019; Levy et al., 2020; Curi et al., 2020; Oren et al., 2019). This includes the CVaR DRO (Levy et al., 2020) method we evaluate against. However, prior work has shown that these approaches may be too pessimistic, optimizing not just for worst-group accuracy but worst possible accuracy within the distribution balls (Sagawa et al., 2019), or too undirected, optimizing for too many subpopulations, e.g. by first upweighting minority points but then upweighting majority points in later stages of training (Liu et al., 2021). Pezeshki et al. (2020) instead suggest that *gradient starvation* (GS), where neural networks only learn to capture statistically dominant features in the data (Combes et al., 2018), is the main culprit behind learning spurious correlations, and introduce a "spectral decoupling" regularizer to alleviate GS. However this does not directly prevent models from learning dependencies on spurious attributes. Similar to CAMEL, Taghanaki et al. (2021) propose Contrastive Input Morphing

(CIM), an image dataset-specific method that aims to learn input feature transformations that remove the effects of spurious or task-irrelevant attributes. They do so without group labels, training a transformation network with a triplet loss to transform input images such that a given transformed image's *structural similarity metric* (based on luminance, contrast, and structure (Wang et al., 2003)) is more similar to a "positive" image from the same class than a "negative" image from a different class. They then train a classifier on top of these representations. Instead of pixel-level similarity metrics, CNC enforces similarity in a neural network's hidden-layer representations, allowing CNC to apply to non-image modalities. Additionally, we sample positives and negatives not just based on class label, but also the learned spurious correlations of an ERM model (via its trained predictions). We hypothesize that our sampling scheme, which intuitively provides "harder" positive and negative examples, allows CNC to more strongly overcome spurious correlations.

Most similar to our approach are methods that first train an initial ERM model with the class labels as a way to identify data points belonging to minority groups, and subsequently train an additional model with greater emphasis on the estimated minority groups. Sohoni et al. (2020) demonstrate that even when only trained on the class labels, neural networks learn feature representations that can be clustered into groups of data exhibiting different spurious attributes. They use the resulting cluster labels as estimated group labels before running GDRO on these estimated groups. Meanwhile, Nam et al. (2020) train a pair of models, where one model minimizes a generalized cross-entropy loss (Zhang & Sabuncu, 2018), such that the datapoints this model classifies incorrectly largely correspond to those in the minority group. They then train the other model on the same data but upweight the minority-group-estimated points. While they interweave training of the biased and robust model, Liu et al. (2021) instead train one model first with a shortened training time (but the standard cross-entropy objective), and show that then upsampling the incorrect data points and training another model with ERM can yield higher worst-group accuracy. Creager et al. (2021) first train an ERM model, and then softly assign the training data into groups under which the initial trained ERM model would maximally violate the invariant risk minimization (IRM) objective. In particular, the IRM objective is maximally satisfied if a model's optimal classifier is the same across groups (Arjovsky et al., 2019), and EIIL groups are inferred such that the initial ERM model's representations exhibit maximum variance within each group. Finally, Nagarajan et al. (2020) provides a theoretical understanding of how ERM picks up spurious features under data set imbalance. They consider a setting involve a single spurious feature that is correlated with the class label and analyze the max-margin classifier in the presence of this spurious feature.

In our work, we demonstrate that the ERM model's predictions can be leveraged to not only estimate groups and train a new model with supervised learning but with different weightings. Instead, we can specifically identify pairs of points that a contrastive model can then learn invariant features between. Our core contribution comes from rethinking the objective with a contrastive loss that more directly reduces the model's ability to learning spurious correlations.

### D.2    CONTRASTIVE LEARNING

Our method also uses contrastive learning, a simple yet powerful framework for both self-supervised (Chen et al., 2020; Oord et al., 2018; Tian et al., 2019; Song & Ermon, 2020; Sermanet et al., 2018; Hassani & Khasahmadi, 2020; Robinson et al., 2021) and supervised (Khosla et al., 2020; Gunel et al., 2021) representation learning. The core idea is to learn data representations that maximize the similarity between a given input "anchor" and distinct different views of the same input ("positives"). Frequently this also involves *contrasting* positives with "negative" data samples without any assumed relation to the anchor (Bachman et al., 2019). Core components then include some way to source multiple views, e.g. with data transformations (Chen et al., 2020), and training objectives similar to noise contrastive estimation (Gutmann & Hyvärinen, 2010; Mnih & Kavukcuoglu, 2013).

An important component of contrastive learning is the method by which appropriate positives and negatives are gathered. For sampling positives, Chen et al. (2020) show that certain data augmentations (e.g. crops and cutouts) may be more beneficial than others (e.g. Gaussian noise and Sobel filtering) when generating anchors and positives for unsupervised contrastive learning. von Kügelgen et al. (2021) theoretically study how data augmentations help contrastive models learn core content attributes which are invariant to different observed "style changes". They propose a latent variable model for self-supervised learning. Tian et al. (2020) further study what makes good views for contrastive learning. They propose an "InfoMin principle", where anchors and positives should

share the least information necessary for the contrastive model to do well on the downstream task. For sampling negatives, Robinson et al. (2021) show that contrastive learning also benefits from using "hard" negatives, which (1) are actually a different class from the anchor (which they approximate in the unsupervised setting) and (2) embed closest to the anchor under the encoder's current data representation. Both of these approaches capture the principle that if positives are always too similar to the anchor and negatives are always too different, then contrastive learning may be inefficient at learning generalizable representations of the underlying classes.

In our work, we incorporate this principle by sampling data points with the same class label but different ERM predictions–presumably because of spurious attribute differences–as anchor and positive views, while sampling negatives from data points with different class labels but the same ERM prediction as the anchor. The anchors and positives are different enough that a trained ERM model predicted them differently, while the anchors and negatives are similar enough that the trained ERM model predicted them the same. Contrasting the above then allows us to exploit both "hard" positive and negative criteria for our downstream classification task. In Appendix A.3, we show that removing this ERM-guided sampling (i.e. only sampling positives and negatives based on class information), as well as trying different negative sampling procedures, leads to substantially lower worst-group accuracy with CNC.

One limitation of our current theoretical analysis regarding the alignment loss (cf. Section 3.2) is that we require knowing the group labels to compute the RHS of equation (6) (in particular, the alignment loss). An interesting question for future work is to provide a better theoretical understanding of the alignment induced by CNC in the context of spurious correlations.

### D.3 LEARNING INVARIANT REPRESENTATIONS

Our work is also similar in motivation to Invariant Risk Minimization (IRM) (Arjovsky et al., 2019), Predictive Group Invariance (PGI) (Ahmed et al., 2021), and other related works in domain-invariant learning (Krueger et al., 2020; Parascandolo et al., 2020; Ahuja et al., 2020; Creager et al., 2021). These methods aim to train models that learn a single invariant representation that is consistently optimal (e.g. with respect to classifying data) across different domains or environments. These environments can be thought of as data groups, and while traditionally methods such as IRM require that environment labels are known, recent approaches such as Environment Inference for Invariant Learning (EIIL) (Creager et al., 2021) and Predictive Group Invariance (PGI) (Ahmed et al., 2021) similarly aim to infer environments with an initial ERM model. In EIIL, they next train a more robust model with an invariant learning objective, similarly selecting models based on the worst-group error on the validation set. However, they train this model using IRM or Group DRO with the inferred environments as group labels. PGI uses EIIL to infer environments, but trains a more robust model by minimizing the KL divergence of the predicted probabilities for samples in the same class, but different groups, using the inferred environments as group labels.

### D.4 REPRESENTATION ALIGNMENT IN DOMAIN GENERALIZATION AND UNSUPERVISED DOMAIN ADAPTATION

Finally, CNC's approach to improve model robustness via a model's learned hidden-layer representations also bears similarity to some prior domain generalization (DG) and unsupervised domain adaptation (UDA) methods. We first discuss similarities and differences between these methods and CNC. We then note fundamental differences between the DG and UDA settings and our spurious correlations setting, which may also explain the poorer performance of the related methods observed in Appendix A.2.

**Related methods for representation alignment.** As introduced in Appendix A.2, to generalize beyond a single domain, such methods try to train model representations for samples that are aligned or invariant across domains. With domain adversarial neural networks (DANN), Ganin et al. (2016) accomplish alignment by adversarially training a model's encoder layers (the "feature extractor") to learn representations such that a separate domain classifier module cannot distinguish samples' domains from the learned representations. To preserve class information, they train a classifier module on top of the feature extractor jointly with a cross-entropy loss. CNC's process for aligning representations is more simple. We do not rely on training separate modules with conflicting objectives to accomplish alignment; instead the supervised contrastive loss with CNC's sampling

procedure encourages learning representations that are both separable across classes and aligned within each class. We thus avoid additional training parameters and optimization issues associated with minimax-based adversarial training (Arjovsky & Bottou, 2017). Instead of relying on a the domain classifier's output, we can train single model to align representations by minimizing the cosine similarity between anchor and positive samples.

Meanwhile, Li et al. (2018) propose *maximum mean discrepancy* MMD adversarial autoencoder (MMD-AAE). To align representations between domains, MMD-AAE (1) trains an autoencoder, (2) uses MMD maximum applied at the bottleneck hidden-layer to match representations across domains, and (3) applies an adversarial discriminator network to match these representations with a Laplace distribution (to encourage more sparse hidden representations). This matching is also not conditioned on the sample classes; an additional classifier head is applied to preserve class-specific information. CNC is also simpler than MMD-AAE, only using the normalized dot product of a single classifier's last hidden-layer representations. Via contrastive learning, CNC also critically also aims to only align representations with the same class but different ERM-inferred groups, while pushing apart samples with different classes but the same ERM-inferred groups. By paying attention to the classes, we directly encourage a model to ignore group-specific information which confused the initial ERM model but that does not discriminate between classes.

Finally, concurrent work applies contrastive learning for alignment in unsupervised domain adaptation (Wang et al., 2021). Similar to CNC, cross-domain contrastive learning (CDCL) trains an encoder to learn aligned representations. However, because the authors tackle UDA, their setup and components are different from ours. They try to align representations for samples with the same class but different known target and source domains. In their setup, domain labels are known. However, *class labels* are unknown for a specific "target" domain. They thus require inferring the *class labels* for target domain samples to set up their anchor, positive, and negative contrastive samples, introducing components that are not relevant for our setting. They infer class labels by first computing class-specific centroids for a "source" domain with class labels, and assign pseudolabels in the "target" domain based on the nearest class-centroid for each target domain sample. CDCL thus tackles a fundamentally different problem. In CNC's spurious correlations setting, we do not have an obvious notion of "target" or "source" domains (instead aiming to improve the robust or worst "group" performance across all unseen test samples), we do not have labels indicating whether samples are from different domains or groups, and we infer the *spurious attribute* values for *all* training samples.

**Fundamental differences in tasks.** All such methods proposed for domain generalization (DG) or unsupervised domain adaptation (UDA) also tackle fundamentally different problem settings from this work. First, both DG and UDA assume knowledge of training data domain labels. However in our setting, we must do the equivalent of inferring these domains. The distribution shift presented in standard DG / UDA benchmarks such as VLCS (Fang et al., 2013) is also distinct from the shift encountered with our spurious correlations datasets. As discussed in Appendix A.2, the notion of a "domain" must also be suitably specified for our setting. For example, in Waterbirds all samples with the same spurious attribute could be the same domain (e.g., the "water background" domain). Alternatively, all samples that exhibit the dominant spurious correlation could be a domain (e.g., the "waterbird, water background; landbird, land background" domain). Using DG or UDA approaches for our setting requires resolving this ambiguity, which is not present in prior DG or UDA works.

**Differences with domain generalization.** In DG, (tackled by MMD-AAE) we have access to multiple "source" domains, know which "group" each sample belongs to (defined by domain-class combinations), and aim to generalize to a specific unseen "target" domain not present during training. Unlike the spurious correlations presented in our evaluated datasets, class distributions within domain are also not skewed. Standard benchmarks such as Office (Saenko et al., 2010) and VLCS (Fang et al., 2013) report much more uniform class distributions for each domain, compared to standard spurious correlations benchmarks such as Waterbirds where $95\%$ of images in each background domain have the same class. Furthermore, in our setting we do not assume group labels or know the domain of any training sample. We also try to improve performance on the worst-group at test-time, which could belong to *any* domain encountered at training time.

**Differences with unsupervised domain adaptation.** In UDA, (tackled by DANN and CDCL) methods also assume knowledge of training sample domains or spurious attributes, whereas CNC and our other comparable methods do not. UDA methods also assume a fundamentally different data setup; the training data is divided into source and target domains, and only the source domain

has labels. The goal is to use the source and target features + the source labels to transfer to the specific target domain. However, in our setting we *do* have class labels for all samples available during training, but *do not* have any natural definition of source and target domains. Applying UDA methods requires additional reintepretation of this setup. Similar to differences in our problem setting with DG, we do not have training data domain labels.

# E  ADDITIONAL EXPERIMENTAL DETAILS

We first further describe our evaluation benchmarks in Appendix E.1. We next provide further details on how we calculate the reported metrics and the experimental hyperparameters of our main results in Appendix E.1. For all methods, following prior work (Liu et al., 2021; Sohoni et al., 2020; Nam et al., 2020; Sagawa et al., 2019; Creager et al., 2021) we report the test set worst-group and average accuracies from models selected through hyperparameter tuning for the best validation set worst-group accuracy. While different methods have different numbers of tunable hyperparameters, we try to keep the number of validation queries as close as possible while tuning for fair comparison.

## E.1  DATASET DETAILS

**Colored MNIST (CMNIST\*).** We evaluate with a version of the Colored MNIST dataset proposed in Arjovsky et al. (2019). The goal is to classify MNIST digits belonging to one of 5 classes $\mathcal{Y} = \{(0, 1), (2, 3), (4, 5), (6, 7), (8, 9)\}$, and treat color as the spurious attribute. In the training data, we color $p_{\text{corr}}$ of each class's datapoints with an associated color $a$, and color the rest randomly. If $p_{\text{corr}}$ is high, trained ERM models fail to classify digits that are not the associated color. We pick $a$ from uniformly interspersed intervals of the `hsv` colormap, e.g. 0 and 1 digits may be spurious correlated with the color red (`#ff0000`), while 8 and 9 digits may be spuriously correlated with purple (`#ff0018`). The full set of colors in class order are $\mathcal{A} = \{$`#ff0000`, `#85ff00`, `#00fff3`, `#6e00ff`, `#ff0018`$\}$ (see Fig. 2). For validation and test data, we color each datapoint randomly with a color $a \in \mathcal{A}$. We use the default test set from MNIST, and allocate 80%-20% of the default MNIST training set to the training and validation sets. For main results, we set $p_{\text{corr}} = 0.995$.

**Waterbirds.** We evaluate with the Waterbirds dataset, which was introduced as a standard spurious correlations benchmark in Sagawa et al. (2019). In this dataset, masked out images of birds from the CUB dataset (Wah et al., 2011) are pasted on backgrounds from the Places dataset (Zhou et al., 2017). Bird images are labeled either as waterbirds or landbirds; background either depicts water or land. From CUB, waterbirds consist of seabirds (ablatross, auklet, cormorant, frigatebird, fulmar, gull, jaeger, kittiwake, pelican, puffin, tern) and waterfowl (gadwell, grebe, mallard, merganser, guillemot, Pacific loon). All other birds are landbirds. From Places, water backgrounds consist of ocean and natural lake classes, while land backgrounds consist of bamboo forest and broadleaf forest classes.

The goal is to classify the foreground bird as $\mathcal{Y} = \{$waterbird, landbird$\}$, where there is spurious background attribute $\mathcal{A} = \{$water background, land background$\}$. We use the default training, validation, and test splits (Sagawa et al., 2019), where in the training data 95% of waterbirds appear with water backgrounds and 95% of landbirds appear with land backgrounds. Trained ERM models then have trouble classifying waterbirds with land backgrounds and landbirds with water backgrounds. For validation and test sets, water and land backgrounds are evenly split among landbirds and waterbirds.

**CelebA.** We evaluate with the CelebA spurious correlations benchmark introduced in Sagawa et al. (2019). The goal is to classify celebrities' hair color $\mathcal{Y} = \{$blond, not blond$\}$, which is spuriously correlated with the celebrity's identified gender $\mathcal{A} = \{$male, female$\}$. We use the same training, validation, test splits as in Sagawa et al. (2019). Only 6% of blond celebrities are male; trained ERM models perform poorly on this group.

**CivilComments-WILDS.** We evaluate with the CivilComments-WILDS dataset from Koh et al. (2021), derived from the Jigsaw dataset from Borkan et al. (2019). Each datapoint is a real online comment curated from the Civil Comments platform, a commenting plugin for independent news sites. For classes, each comment is labeled as either toxic or not toxic. For spurious attributes, each comment is also labeled with the demographic identities $\{$male, female, LGBTQ, Christian, Muslim, other religions, Black, White$\}$ mentioned; multiple identities may be mentioned per comment.

The goal is to classify the comment $\mathcal{Y} = \{\text{toxic, not toxic}\}$. As in Koh et al. (2021), we evaluate with $\mathcal{A} = \{\text{male, female, LGBTQ, Christian, Muslim, other religions, Black, White}\}$. There are then 16 total groups corresponding to (toxic, identity) and (not toxic, identity) for each identity. Groups may overlap; a datapoint falls in a group if it mentions the identity. We use the default data splits (Koh et al., 2021). In Table A.6, we list the percentage of toxic comments for each identity based on the groups. Trained ERM models in particular perform less well on the rarer toxic groups.

Table A.6: Percent of toxic comments for each identity in the CivilComments-WILDS training set.

| Identity | male | female | LGBTQ | Christian | Muslim | other religions | Black | White |
|---|---|---|---|---|---|---|---|---|
| % toxic | 14.9 | 13.7 | 26.9 | 9.1 | 22.4 | 15.3 | 31.4 | 28.0 |

### E.2 IMPLEMENTATION DETAILS

#### E.2.1 REPORTED METRICS

**Main results.** For the CMNIST*, Waterbirds, and CelebA data sets, we run CNC with three different seeds, and report the average worst-group accuracy over these three trials in Table 1. As we use the same baselines and comparable methods as Liu et al. (2021), we referenced their main results for the reported numbers, which did not have standard deviations or error bars reported. For CivilComments-WILDS, due to time and compute constraints we only reported one run. We note that CMNIST* here is extremely challenging, as minority groups together only make up $0.5\%$ of the training set. This severe imbalance explains the very poor worst-group performance of ERM (as well as a couple other methods that fail to sufficiently remediate issue).

**Estimated mutual information.** We give further details for calculating the representation metric introduced in Sec. 3. As a reminder, we report both alignment and estimated mutual information metrics to quantify how dependent a model's learned representations are on the class labels versus the spurious attributes, and compute both metrics on the representations $Z = \{f_{\text{enc}}(x)\}$ over all test set data points $x$. Then to supplement the alignment loss calculation in Sec. 3, we also estimate $I(Y; Z)$ and $I(A; Z)$, the mutual information between the model's data representations and the class labels and spurious attribute labels respectively.

To first estimate mutual information with $Y$, we first approximate $p(y \mid z)$ by fitting a multinomial logistic regression model over all representations $Z$ to classify $y$. With the empirical class label distribution $p(y)$, we compute:

$$\hat{I}(Y; Z) = \frac{1}{|Z|} \sum_{z \in Z} \sum_{y \in Y} p(y \mid z) \log \frac{p(y \mid z)}{p(y)} \tag{18}$$

We do the same but substitute the spurious attributes $a$ for $y$ to compute $\hat{I}(A; Z)$.

#### E.2.2 STAGE 1 ERM TRAINING DETAILS

We describe the model selection criterion, architecture, and training hyperparameters for the initial ERM model in our method. To select this model, recall that we first train an ERM model to predict the class labels, as the model may also learn dependencies on the spurious attributes. Because we then use the model's predictions on the training data to infer samples with different spurious attribute values but the same class label, we prefer an initial ERM model that better learns this spurious dependency, and importantly also does not overfit to the training data. Inspired by the results in prior work (Sohoni et al., 2020; Liu et al., 2021), we then explored using either a standard ERM model, one with high $\ell$-2 regularization (`weight decay = 1`), or one only trained on a few number of epochs. To select among these, because the validation data has both class labels and spurious attributes, we choose the model with the largest gap between worst-group and average accuracy on the validation set. For fair comparison to JTT, we use the same batch size, learning rate, momentum, optimizer, default weight decay, and number of epochs as reported in Liu et al. (2021) to obtain these models. We detail the ERM architecture and hyperparameters for each dataset below:

**Colored MNIST.** We use the LeNet-5 CNN architecture in the `pytorch` image classification tutorial. We train with SGD, few epochs $E = 5$, SGD, learning rate 1e-3, batch size 32, default weight decay 5e-4, and momentum 0.9.

**Waterbirds.** We use the `torchvision` implementation of ResNet-50 with pretrained weights from ImageNet as in Sagawa et al. (2019). Also as in (Sagawa et al., 2019), we train with SGD, default epochs $E = 300$, learning rate 1e-3, batch size 128, and momentum 0.9. However we use high weight decay 1.0.

**CelebA.** We also use the `torchvision` ImageNet-pretrained ResNet-50 and default hyperparameters from Sagawa et al. (2019) but with high weight decay: we train with SGD, default epochs $E = 50$, learning rate 1e-4, batch size 128, momentum 0.9, and high weight decay 0.1.

**CivilComments-WILDS.** We use the HuggingFace (`pytorch-transformers`) implementation of BERT with pretrained weights and number of tokens capped at 300 as in Koh et al. (2021). As in Liu et al. (2021), with other hyperparameters set to their defaults (Koh et al., 2021) we tune between using the AdamW optimizer with learning rate 1e-5 and SGD with learning rate 1e-5, momentum 0.9, and the PyTorch `ReduceLROnPlateau` learning rate scheduler. Based on our criterion, we use SGD, few number of epochs $E = 2$, learning rate 1e-5, batch size 16, default weight decay 1e-2, and momentum 0.9.

### E.2.3   CONTRASTIVE BATCH SAMPLING DETAILS

We provide further details related to collecting predictions from the trained ERM models, and the number of positives and negatives that determine the contrastive batch size.

**ERM model prediction.** To collect trained ERM model predictions on the training data, we explored two approaches: (1) using the actual predictions, i.e. the argmax for each classifier layer output vector, and (2) clustering the representations, or the last hidden-layer outputs, and assigning a cluster-specific label to each data point in one cluster. This latter approach is inspired by Sohoni et al. (2020), and we similarly note that ERM models trained to predict class labels in spuriously correlated data may learn data representations that are clusterable by spurious attribute. As a viable alternative to collecting the "actual" predictions of the trained ERM model on the training data, with $C$ classes, we can then cluster these representations into $C$ clusters, assign the same class label only to each data point in the same cluster, and choose the label-cluster assignment that leads to the highest accuracy on the training data. We also follow their procedure to first apply UMAP dimensionality reduction to 2 UMAP components, before clustering with K-means or GMM (Sohoni et al., 2020). To choose between all approaches, we selected the procedure that lead to highest worst-group accuracy on the validation data after the second-stage of training. While this cluster-based prediction approach was chosen as a computationally efficient heuristic, we found that in practice it either lead to comparable or better final worst-group accuracy on the validation set. To better understand this, as a preliminary result we found that when visualizing the validation set predictions with the Waterbirds dataset, the cluster-based predictions captured the actual spurious attributes better than the classifier layer predictions (Fig. 10). We defer additional discussion to Sohoni et al. (2020) and leave further analysis to future work.

**Number of positives and negatives per batch.** One additional difference between our work and prior contrastive learning methods (Chen et al., 2020; Khosla et al., 2020) is that we specifically construct our contrastive batches by sampling anchors, positives, and negatives first. This is different from the standard procedure of randomly dividing the training data into batches first, and then assigning the anchor, positive, and negative roles to each datapoint in a given batch. As a result, we introduce the number of positives $M$ and the number of negatives $N$ as two hyperparameters that primarily influence the size of each contrastive batch (with number of additional anchors and negatives also following $M$ and $N$ with two-sided batches). To maximize the number of positive and negative comparisons, as a default we set $M$ and $N$ to be the maximum number of positives and negatives that fit the sampling criteria specified under Algorithm 2 that also can fit in memory. In Appendix E.2.4, for each dataset we detail the ERM prediction method and number of positives and negatives sampled in each batch.

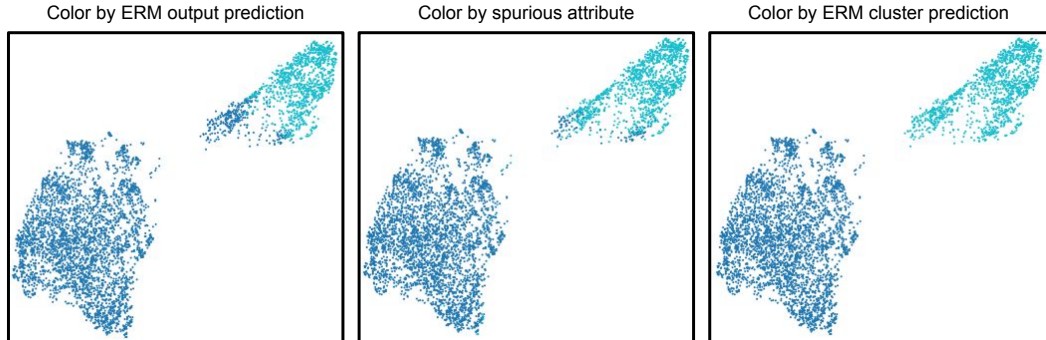

Figure 10: UMAP visualization of ERM data representations for the Waterbirds training data. We visualize the last hidden layer outputs for a trained ERM ResNet-50 model given training samples from Waterbirds, coloring by either the ERM model's "standard" predictions, the actual spurious attribute values (included here just for analysis), and predictions computed by clustering the representations as described above. Clustering-based predictions more closely align with the actual spurious attributes than the ERM model outputs.

### E.2.4 STAGE 2 CONTRASTIVE MODEL TRAINING DETAILS

In this section we describe the model architectures and training hyperparameters used for training the second model of our procedure, corresponding the reported worst-group and average test set results in Table 1. In this second stage, we train a new model with the same architecture as the initial ERM model, but now with a contrastive loss and batches sampled based on the initial ERM model's predictions. We report test set worst-group and average accuracies from models selected with hyperparameter tuning and early stopping based on the highest validation set worst-group accuracy. For all data sets, we sample contrastive batches using the clustering-based predictions of the initial ERM model. Each batch size specified here is also a direct function of the number of positives and negatives: $2M + 2N$.

**Colored MNIST.** We train a LeNet-5 CNN. For CNC, we use $M = 32$, $N = 32$, batch size 128, temperature $\tau = 0.05$, contrastive weight $\lambda = 0.75$, SGD optimizer, learning rate 1e-3, momentum 0.9, and weight decay 1e-4. We train for 3 epochs, and use gradient accumulation to update model parameters every 32 batches.

**Waterbirds.** We train a ResNet-50 CNN with pretrained ImageNet weights. For CNC, we use $M = 17$, $N = 17$, batch size 68, temperature $\tau = 0.1$, contrastive weight $\lambda = 0.75$, SGD optimizer, learning rate 1e-4, momentum 0.9, weight decay 1e-3. We train for 5 epochs, and use gradient accumulation to update model parameters every 32 batches.

**CelebA.** We train a ResNet-50 CNN with pretrained ImageNet weights. For CNC, we use $M = 64$, $N = 64$, batch size 256, temperature $\tau = 0.05$, contrastive weight $\lambda = 0.75$, SGD optimizer, learning rate 1e-5, momentum 0.9, and weight decay 1e-1. We train for 15 epochs, and use gradient accumulation to update model parameters every 32 batches.

**CivilComments-WILDS.** We train a BERT model with pretrained weights and max number of tokens 300. For CNC, we use $M = 16$, $N = 16$, batch size 64, temperature $\tau = 0.1$, contrastive weight $\lambda = 0.75$, AdamW optimizer, learning rate 1e-4, weight decay 1e-2, and clipped gradient norms. We train for 10 epochs, and use gradient accumulation to update weights every 128 batches.

### E.2.5 COMPARISON METHOD TRAINING DETAILS

As reported in the main results (Table 1) we compare CNC with the ERM and Group DRO baselines, as well as robust training methods that do not require spurious attribute labels for the training data: CVaR DRO (Levy et al., 2020), GEORGE (Levy et al., 2020), Learning from Failure (LfF) (Levy et al., 2020), Predictive Group Invariance (PGI) (Ahmed et al., 2021), Contrastive Input Morphing (CIM) (Taghanaki et al., 2021), Environment Inference for Invariant Learning (EIIL) (Creager et al., 2021), and Just Train Twice (JTT) (Liu et al., 2021). For each dataset, we use the same model architecture for all methods. For the Waterbirds, CelebA, and CivilComments-WILDS data sets, we report the worst-group and average accuracies reported in Liu et al. (2021) for ERM, CVaR DRO,

LfF, and JTT. For GEORGE, we report the accuracies reported in Sohoni et al. (2020). For CIM, we report results from Waterbirds and CelebA from Taghanaki et al. (2021) using the CIM + variational information bottleneck implementation Alemi et al. (2016), which achieves the best worst-group performance in their results. For EIIL, we report results from Waterbirds and CivilComments-WILDS from Creager et al. (2021). For these hyperparameters, we defer to the original papers. For GDRO, we reproduce the results with the same optimal hyperparameters over three seeds. For PGI, we used our own implementation for all results, with details specified below. For Colored MNIST, we run implementations for GEORGE, CIM, EIIL, JTT, and GDRO, using code from their authors respectively. LfF and CVaR DRO are also run with code from the JTT authors. We include training details for our own implementations below:

**Colored MNIST (CMNIST$^*$).** We run all methods for 100 epochs, reporting test set accuracies with early stopping. For JTT, we train with SGD, learning rate 1e-3, momentum 0.9, weight decay 5e-4, batch size 32. We use the same initial ERM model as CNC, with hyperparameters described in Appendix E.2.2. For upsampling we first tried constant factors $\{10, 100, 1000\}$. We also tried a resampling strategy where for all the datapoints with the same initial ERM model prediction, we upsample the incorrect points such that they equal the correct points in frequency, and found this worked the best. With $p_{\text{corr}} = 0.995$, this upsamples each incorrect point by roughly 1100. We also use this approach for the results in Fig. 7. For GDRO we use the same training hyperparameters as JTT, but without the upsampling and instead set group adjustment parameter $C = 0$. For LfF, we use the same hyperparameters as JTT, but instead of upsampling gridsearched the $q$ parameter $\in \{0.1, 0.3, 0.5, 0.7, 0.9\}$, using $q = 0.7$. For CVaR DRO we do the same but use hyperparameter $\alpha = 0.1$. For GEORGE we train with SGD, learning rate 1e-3, momentum 0.9, weight decay 5e-4. For CIM, we use the CIM + VIB implementation. We train with SGD, learning rate 1e-3, weight decay 5e-4, $\beta$ parameter 10, and $\lambda$ parameter 1e-5. For EIIL, for environment inference we use the same initial ERM model as CNC and JTT, and update the soft environment assignment distribution with Adam optimizer, learning rate 1e-3, and 10000 steps. Following Creager et al. (2021)'s own colored MNIST experiment, we train the second model with IRM, using learning rate 1e-2, weight decay 1e-3, penalty weight 100, and penalty annealing parameter 80.

**CelebA.** We also tune EIIL for CelebA. We again use the same initial ERM model as CNC, and update the soft environment assignment distribution with Adam optimizer, learning rate 1e-3, and 10000 steps. We train the second model with GDRO, using SGD, 50 epochs, learning rate 1e-5, batch size 128, weight decay 0.1, and group adjustment parameter 3.

**PGI** To compare against PGI, we tried two implementations. First, we followed the PGI algorithm to first infer environments via the same mechanism as in EIIL [2], and trained a second model with the PGI objective using standard shuffled minibatches (aiming to minimize the KL divergence for samples with the same class but different inferred environment labels per batch). However, despite ample hyperparameter tuning (trying loss weighting component $\lambda \in \{0.1, 0.5, 10, 100\}$, we could not get PGI to work well (on Waterbirds, we obtained $51.0 \pm 4.9\%$ worst-group accuracy and $79.6 \pm 2.6\%$ average accuracy). We hypothesize this is due to the strong spurious correlations in our datasets: while Ahmed et al. (2021) only considers datasets where 20% of the training samples do not exhibit a dominant correlation and fall under minority groups. Our evaluation benchmarks are more difficult due to stronger spurious correlations, e.g., in Waterbirds only $5\%$ of samples do not exhibit the dominant correlation; similarly only $7\%$ of training samples lie in the smallest group in CelebA.

We then tried a more balanced batch variation. Instead of using randomly shuffled minibatches, we used the PGI environment inference labels to sample batches similarly to how CNC uses the stage 1 ERM model predictions to sample batches. We construct batches by specifying the same number of "anchors", "positives", and "negatives" as in CNC, and sample batches where anchors and positives are samples with the same class, but different inferred environments. Anchors and negatives are samples in the same inferred environment, but with different classes. We then trained a second model with the PGI criterion with these modified batches.

In Section E.2.6, we include our sweeps for both the method-specific and general hyperparameters.

**Comparison limitations.** One limitation of our comparison is that because for each dataset we sample new contrastive batches which could repeat certain datapoints, the number of total batches per epoch changes. For example, 50 epochs training the second model in CNC does not necessarily lead to the same total number of training batches as 50 epochs training with ERM, even if they use the

same batch size. However, we note that the numbers we compare against from Liu et al. (2021) are reported with early stopping. In this sense we are comparing the best possible worst-group accuracies obtained by the methods, not the highest worst-group accuracy achieved within a limited number of training batches. We also found that although in general the time to complete one epoch takes much longer with CNC, CNC requires fewer overall training epochs for all but the CivilComments-WILDS dataset to obtain the highest reported accuracy.

### E.2.6 HYPERPARAMETER SWEEPS

To fairly compare with previous methods (Liu et al., 2021; Creager et al., 2021; Taghanaki et al., 2021), we use the same evaluation scheme (selecting models based on worst-group validation error), and sweep over a consistent number of hyperparameters, i.e. number of validation set queries. We set this number for CNC to be a comparable number of queries that is reported in prior works. We break this down into method-specific (e.g. contrastive temperature in CNC, upweighting factor in JTT), and shared (e.g. learning rate) hyperparameter categories.

**Method-specific** For CNC, we tune three method-specific hyperparameters: contrastive loss temperature (Eq. 3), contrastive weight (Eq. 7), and gradient accumulation steps values as in Table A.7.

Table A.7: Method-specific hyperparameters for CNC.

| Hyperparameter | Dataset | Values |
|---|---|---|
| Temperature ($\tau$) | All | $\{0.05, 0.1\}$ |
| Contrastive Weight ($\lambda$) | All | $\{0.5, 0.75\}$ |
| Gradient Accumulation Steps | CMNIST$^*$, Waterbirds, CelebA
CivilComments-WILDS | $\{32, 64\}$
$\{32, 64, 128\}$ |

For JTT, the reported results and our CMNIST$^*$ implementation are tuned over the following hyperparameters in Table A.8.

Table A.8: Method-specific hyperparameters for JTT.

| Hyperparameter | Dataset | Values |
|---|---|---|
| Stage 1 Training Epochs | Waterbirds
CMNIST$^*$, CelebA, CivilComments-WILDS | $\{40, 50, 60\}$
$\{1, 2\}$ |
| Upweighting Factor | CMNIST$^*$
Waterbirds, CelebA
CivilComments-WILDS | $\{10, 100, 1000, 1100\}$
$\{20, 50, 100\}$
$\{4, 5, 6\}$ |

For EIIL, our CMNIST$^*$ and CelebA implementations are tuned over hyperparameters reported in Table A.9. Creager et al. (2021) report that they allow up to 20 evaluations with different hyperparameters for Waterbirds and CivilComments-WILDS. When using GDRO as the second stage model, they also report using the same hyperparameters as the GDRO baseline for Waterbirds. We do the same for our evaluation on CelebA. This amounts to primarily tuning the first stage environment inference learning rate and number of updating steps for CMNIST$^*$ and CelebA, and the penalty annealing iterations and penalty weight for the IRM second stage model for CMNIST$^*$.

Table A.9: Method-specific hyperparameters for EIIL.

| Hyperparameter | Dataset | Values |
|---|---|---|
| Environment Inference Learning Rate | CMNIST$^*$, CelebA | {1e-1, 1e-2, 1e-3} |
| Environment Inference Update Steps | CMNIST$^*$, CelebA | {10000, 20000} |
| IRM Penalty Weight | CMNIST$^*$ | {0.1, 10, 1000, 1e5} |
| IRM Penalty Annealing Iterations | CMNIST$^*$ | {10, 50, 80} |
| GDRO Group Adjustment | CelebA | {0, 2, 3} |

For CIM, our CMNIST* implementation uses CIM + VIB, and is tuned over the $\beta$ VIB parameter Alemi et al. (2016) and contrastive weighting parameter $\lambda$ for CIM in Table A.10. Taghanaki et al. (2021) report tuning over a range of values within [1e-5, 1] for $\lambda$ on the CelebA and Waterbirds data sets.

Table A.10: Method-specific hyperparameters for CIM.

| Hyperparameter | Dataset | Values |
|---|---|---|
| CIM $\lambda$ | CMNIST* | {0.01, 0.05, 0.1} |
| VIB $\beta$ | CMNIST* | {1e-5, 1e-3, 1e-1, 10} |

For PGI, we tune $\lambda$ with the same environment inference parameters as used in EIIL (Table A.9). Fixing these parameters to infer environments, we tuned the $\lambda$ component for training the robust model across $\lambda \in \{0.1, 0.5, 10, 100\}$.

**Shared** For all data sets, we use the same optimizer and momentum (if applicable) as reported in the JTT paper. Table A.11 contains the data-specific shared hyperparameter values tried.

Table A.11: Shared hyperparameters

| | CMNIST* | Waterbirds | CelebA | CivilComments-WILDS |
|---|---|---|---|---|
| Learning Rate | {1e-4, 1e-3, 1e-2} | {1e-4, 1e-3} | {1e-5, 1e-4} | {1e-5, 1e-4} |
| Weight Decay | {1e-4, 5e-4} | {1e-4, 1e-3} | {1e-2, 1e-1} | {1e-2} |

### E.3 CNC COMPUTE RESOURCES AND TRAINING TIME

All experiments for CMNIST*, Waterbirds, and CelebA were run on a machine with 14 CPU cores and a single NVIDIA Tesla P100 GPU. Experiments for CivilComments-WILDS were run on an Amazon EC2 instance with eight CPUs and one NVIDIA Tesla V100 GPU.

Regarding runtime, one limitation with the current implementation of CNC is its comparatively longer training time compared to methods such as standard ERM. This is both a result of training an initial ERM model in the first stage, and training another model with contrastive learning in the second stage. In Table A.12 we report both how long it takes to train the initial ERM model and long it takes to complete one contrastive training epoch on each dataset. We observe that while in some cases training the initial ERM model is negligible, especially if we employ training with only a few epochs to prevent memorization (for Colored MNIST it takes roughly two minutes to obtain a sufficient initial ERM model), it takes roughly 1.5 and 3 hours to train the high regularization initial models used for Waterbirds and CelebA. While these hurdles are shared by all methods that train an initial ERM model, we find that the second stage of CNC occupies the bulk of training time. Prior work has shown that contrastive learning typically requires longer training times and converges more slowly than supervised learning (Chen et al., 2020). We also observe this in our work.

We note however that because we sample batches based on the ERM model's predictions, the contrastive training duration is limited by how many datapoints the initial ERM model predicts incorrectly. In moderately sized data sets with very few datapoints in minority groups, (e.g. Waterbirds, which has roughly 4794 training points and only 56 datapoints in its smallest group), the total time it takes to train CNC is on par with ERM. Additionally, other methods such as additional hard negative mining (Robinson et al., 2021) have been shown to improve the efficiency of contrastive learning, and we can incorporate these components to speed up training time as well.

Table A.12: CNC Average total training time for first and second stages of CNC

| Dataset | CMNIST* | Waterbirds | CelebA | CivilComments-WILDS |
|---|---|---|---|---|
| Stage 1 ERM train time | 2 min. | 1.5 hrs | 3 hrs | 3.1 hrs |
| Stage 2 CNC train time | 1.2 hrs | 1.8 hrs | 32.2 hrs | 37.6 hrs |

# F VISUALIZATION OF LEARNED DATA REPRESENTATIONS

As in Fig. 6, we visualize and compare the learned representations of test set samples from models trained with ERM, JTT, and CNC in Fig. 11. Compared to ERM models, both JTT and CNC models learn representations that better depict dependencies on the class labels. However, especially with the Waterbirds and CelebA data sets, CNC model representations more clearly depict dependencies only on the class label, as opposed to JTT models which also show some organization by the spurious attribute still.

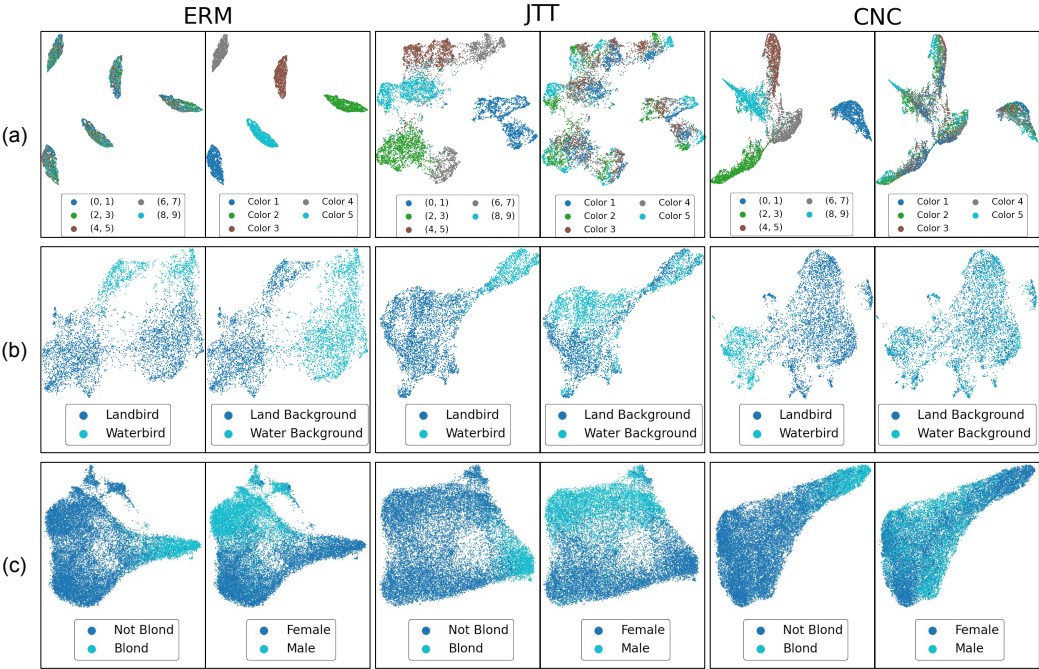

Figure 11: UMAP visualizations of learned representations for Colored MNIST (a), Waterbirds (b), and CelebA (c). We color data points based on the class label (left) and spurious attribute (right). Most consistently across data sets, CNC representations exhibit dependence and separability by the class label but not the spurious attribute, suggesting that they best learn features which only help classify class labels.

# G ADDITIONAL GRADCAM VISUALIZATIONS

On the next two pages, we include additional GradCAM visualizations depicting saliency maps for samples from each group in the Waterbirds and CelebA data sets. Warmer colors denote higher saliency, suggesting that the model considered these pixels more important in making the final classification as measured by gradient activations. For both data sets, we compare maps from models trained with ERM, the next most competitive method for worst-group accuracy JTT, and CNC. CNC models most consistently measure highest saliency with pixels directly associated with class labels and not spurious attributes.

## G.1 WATERBIRDS

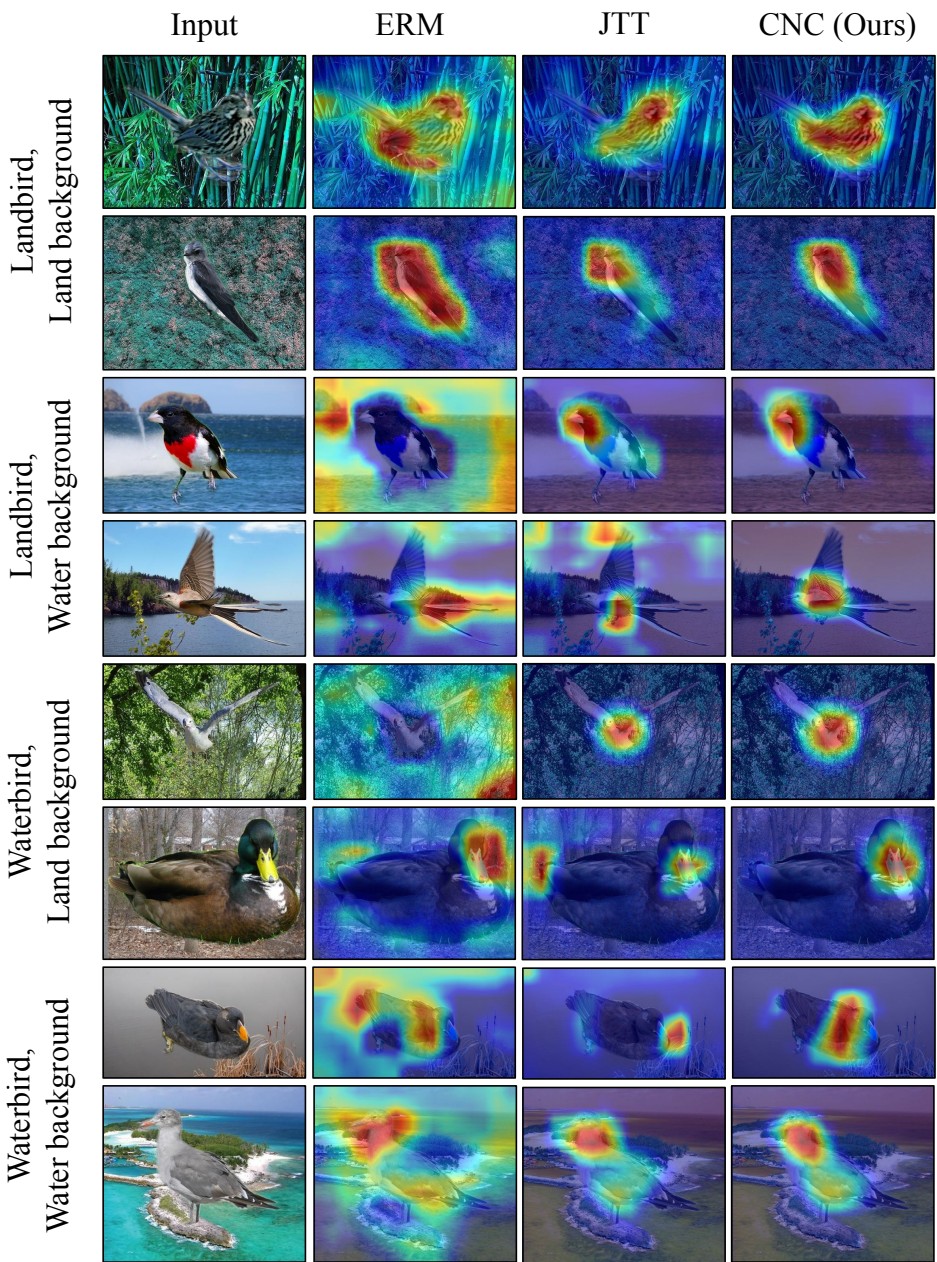

Figure 12: Additional GradCAM visualizations for the Waterbirds dataset. We use GradCAM to visualize the "salient" observed features used to classify images by bird type for models trained with ERM, JTT, and CNC. ERM models output higher salience for spurious background attribute pixels, sometimes almost exclusively. JTT and CNC models correct for this, with CNC better exclusively focusing on bird pixels.

## G.2 CelebA

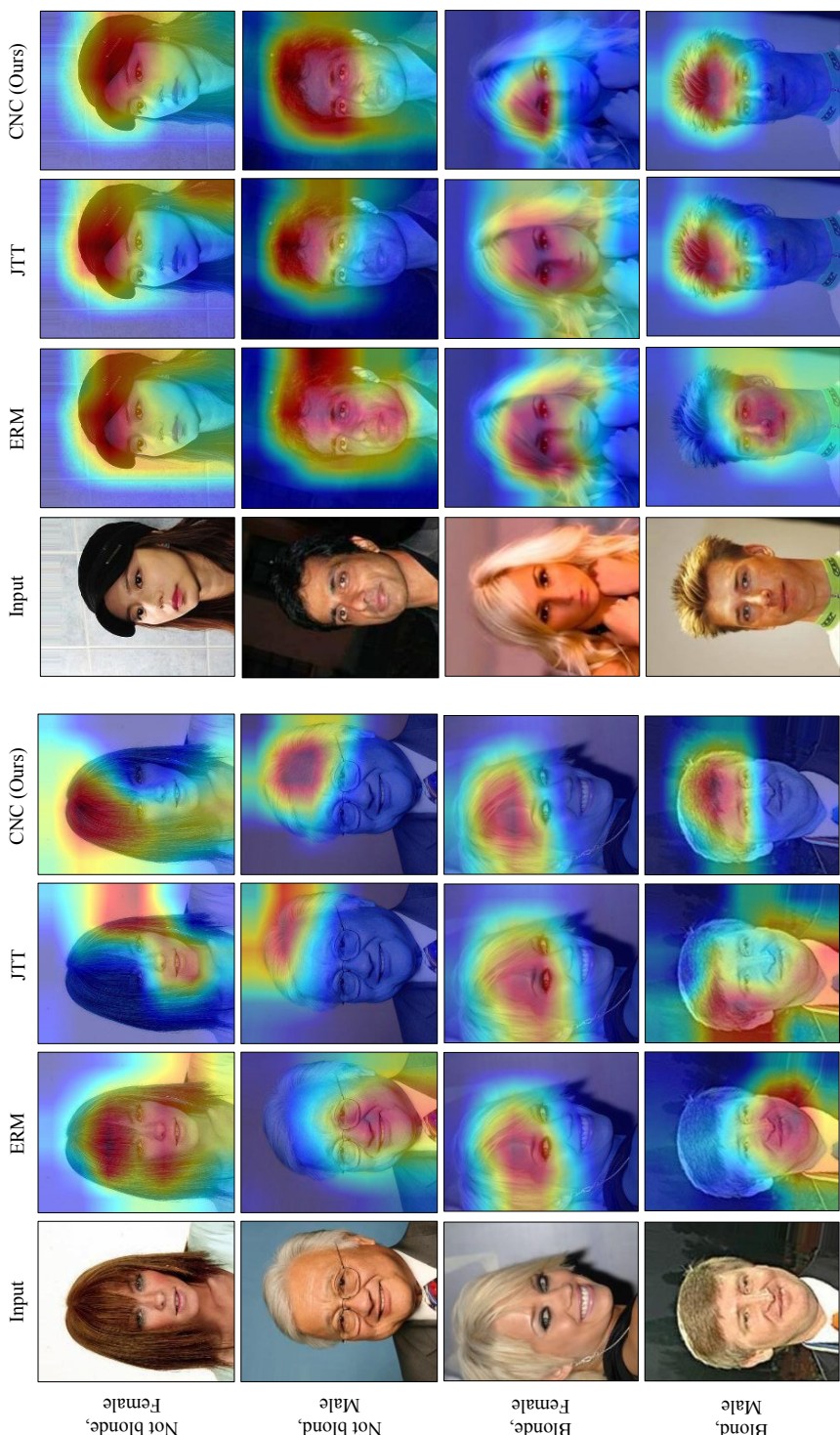

Figure 13: Additional GradCAM visualizations for the CelebA dataset. For models trained with ERM, JTT, and CNC, we use GradCAM to visualize the "salient" observed features used to classify whether a celebrity has blond(e) hair. ERM models interesting also "ignore" the actual hair pixels in favor of other pixels, presumably associated with the spurious gender attribute. In contrast, GradCAMs for JTT and CNC models usually depict higher salience for regions that at least include hair pixels. CNC models most consistently do so.

