# OpenReview forum: "Correct-N-Contrast: a Contrastive Approach for Improving Robustness to Spurious Correlations"
_ICLR.cc/2022/Conference — ICLR 2022 Submitted_

### Official Review · Reviewer_ktqR · 2021-10-25

**Correctness:** 4
**Technical Novelty And Significance:** 4
**Empirical Novelty And Significance:** 4
**Recommendation:** 6
**Confidence:** 4

**Main Review:**


Strengths:
- it's a good idea to explore the connection between learned representations and subgroup performance
- they show fairly clearly that in the explored methods, better separated representations often yield better predictions
- the method seems to perform well experimentally

Weaknesses:
- I'm not sure I totally buy the proof being that intuitively useful - in particular, due to the bound B on the weight matrix. My hesitancy is because the weight matrix and the representations are learned jointly - in fact, we could get equivalent predictions by scaling the weight matrix down and the representations up. Also, the Lipschitz and boundedness constraints on the loss functions do not really apply in any of the settings explored experimentally.
- I get a little lost in Sec 5.2. I don't what understand the role of ERM "predicting the sensitive attribute" - I thought the point was for ERM to predict the label? and how do ERM's predictions of the sensitive attribute play into the CNC algorithm?
- there are some training details buried in the appendix which seem worth discussing - for instance the clustering-based prediction from the first step ERM model seems like an unintuitive step which may be fairly important to the functioning of the method. I would like to see this discussed in the main body, possibly with an ablation study. In my experience, clustering approaches can be quite helpful for these types of problems and I would like to know a bit more about the role it plays, given that it is far from the first thing you would think of doing (which would be just using the standard linear layer)

Other thoughts:
- In Figure 3, the relationship I would really like to see is L_align vs Accuracy: this is the one that makes your point most compellingly
- In Fig 3c, I disagree with the characterization that high worst group accuracy corresponds to a combination of high I(Y,Z) and low I(A, Z). It looks like WG accuracy is mostly (but not fully) invariant to I(A, Z) in this plot, with the level-colour sets extending horizontally (more or less) across the plot
- Some of the notation in the proof in 3.2 is a little sloppy - in particular, y is overloaded in the definitions of L_wg and L_avg, both being used inside the scope of the expectation and outside it


**Summary Of The Paper:**

This paper discusses a two-stage method for improving a model's subgroup robustness, first training an ERM classifier and then performing contrastive learning on the representations. They provide theoretical justification for this procedure and experimental verification of its performance.


**Summary Of The Review:**

The paper can use some cleaning up but the idea is interesting and clearly communicted enough to be of value to the conference.

---

> ### Author Response · Authors · 2021-11-20
> **Updated paper with clarification on theory and role of Stage 1 ERM model (1/2)**
>
> Thank you for your positive feedback and constructive comments. Following your suggestions, we have updated the main paper to include additional details and ablations regarding training details, and have cleaned up our presentation and notation to hopefully address your remaining suggestions.
>
> **“I'm not sure I totally buy the proof being that intuitively useful - in particular, due to the bound B on the weight matrix. My hesitancy is because the weight matrix and the representations are learned jointly - in fact, we could get equivalent predictions by scaling the weight matrix down and the representations up.”**
> Thanks for the insightful remark. While we agree with the reviewer’s intuition that rescaling the weight matrix down and the representation up simultaneously do not change the prediction, we would also like to point out that in the second contrastive step, we normalize the representations to always have unit length. In cases where the representations are not normalized, scaling the weight matrix down and the representations up will not change the weight matrix norm bound times the alignment loss either (in addition to the prediction); thus, our result persists under such rescaling.
>
>
> **“Also, the Lipschitz and boundedness constraints on the loss functions do not really apply in any of the settings explored experimentally.”**
> Thanks for pointing this out. We agree with the reviewer’s suggestion--in the revised paper, we have extended Theorem 3.1 and its proof to cases where the loss function is Lipschitz continuous for any fixed constant and upper-bounded by a fixed constant as well. This is achieved by taking care of the Lipschitz constant (in equation 11) and the boundedness constraint (in equation 9), while keeping all other parts the same. This extension now applies to loss functions such as the cross-entropy loss, as long as the input data comes from a bounded domain, both of which are satisfied in our experimental settings.
>
>
> **Clarification on role of Stage 1 ERM model in Section 5.2 / Section 5.3**
> Apologies for the confusion here. To clarify, you are correct that we train the Stage 1 ERM model to predict the ground-truth class label. However, our intention is to use these predictions as proxies for the spurious attributes of our training datapoints (Sec. 4), such that we sample the specific “hard” positives and negatives used for training the robust Stage 2 contrastive model.
>
> * By training a contrastive model to learn similar representations for points with the same ground-truth class but different spurious attributes, and dissimilar representations for points with different ground-truth classes but the same spurious attribute, we can obtain a model that more strongly relies on class-specific features (Fig. 12, 13) to encode class-specific information (Fig. 5, 6). This model avoids learning spurious correlations, and is more robust in our problem setting.
> * However, while we have ground-truth class labels in our training data, we do not have spurious attribute labels. To obtain anchor-positive and anchor-negative pairs, we must thus infer these spurious labels, which is done via the ERM model predictions.
>   * The logic for this is supported by the ERM behavior described in Sec. 3.1 and prior work [1], where ERM models trained to predict ground-truth classes in spuriously correlated data actually learn spurious correlations. The ERM model therefore makes predictions strongly indicative of spurious features.
>
> Due to this reliance on the stage 1 ERM model predictions, the purpose of Section 5.3 is to then study the effect of how sensitive CNC’s performance is with respect to how well the ERM model predictions actually match with the spurious attributes. Other methods also rely on an initial ERM model’s predictions matching the spurious attributes (e.g., JTT, the next best method on our benchmarks), and we compare CNC with JTT. To reiterate our findings, we find that CNC does not require the Stage 1 ERM model’s predictions to perfectly discriminate samples by their spurious features to work well. CNC also seems to be more robust than JTT when the Stage 1 predictions do not align as well with the true spurious attributes.

---

> > ### Author Response · Authors · 2021-11-20
> > **Clarification on Stage 1 ERM prediction; responses to additional thoughts (2/2)**
> >
> > **Clustering-based vs linear layer-based ERM prediction**
> > We have updated Sec. 4 in the main paper to include details and discussion on these two ways to obtain ERM model predictions. We felt that these approaches were rather similar in motivation and outcome, which is why we delegated this as a training detail. We discuss these similarities and show results for this claim below.
> >
> > Both approaches are used to infer spurious attributes, and exploit the phenomenon that trained ERM models learn spurious correlations with spuriously correlated training data. If this is the case, the ERM’s last hidden-layer representations should encode information about samples’ spurious attributes (which we empirically support in Sec. 3.1, Fig. 3bcd, Fig. 6, Fig. 11), i.e. we’d expect the representations of examples with different spurious attributes to be roughly linearly separable. We can then make predictions indicative of this information using either a linear classification layer for N classes that takes these representations as input, or clustering the hidden-layer representations into N clusters. We thus view the clustering as a rather simple alternative to the standard linear layer-based prediction. We were also inspired by the clustering-based approach to infer groups given an initial ERM model used in Sohoni et al. 2020 [2].
> >
> > Empirically, we found the clustering-based approach to lead to marginally higher worst-group accuracy on the validation sets on the CMNIST and Waterbirds datasets, and so used this approach for all datasets. For example, using the ERM model’s final linear-layer predictions leads to 77.1% worst-group accuracy on CMNIST, compared to 77.4% when using the clustering-based predictions. We will add a full comparison of these two approaches to the final paper.
> >
> >
> > **Responses to other thoughts**
> > ***L_align vs Accuracy in Figure 3:*** In Fig. 3b. we show L_align vs worst-group accuracy, where on average lower alignment loss corresponds to higher worst-group accuracy. We show this relationship because our main theory (Thm 3.1) shows that L_align plays a key role in bounding the worst-group vs average accuracy gap. Fig. 3a and 3b together thus show that as the spurious correlation becomes stronger, ERM models trained on the data exhibit larger worst-group vs average accuracy gaps and higher alignment loss. In our updated draft, we have cleaned up Fig. 3b and 3c to depict the representation metrics for the ERM models at the end of  training (before we reported the metrics after each training checkpoint - the trends still hold).
> >
> > ***Disagreement that high worst-group accuracy corresponds to a combination of high I(Y; Z) and low I(A; Z):*** Thank you for this comment. Our original intended point was that even if there was high I(Y; Z) in the model’s representations, comparably high I(A; Z) led to reduced worst-group performance, motivating our intention to reduce I(A; Z). As a note on the shading, with relatively high I(Y;Z) > 1.4, we go from >80% worst-group accuracy to <40% worst-group accuracy as I(A; Z) increases. The problem is that with highly spuriously correlated data, ERM models naturally learn representations with high I(A; Z) (Fig. 3d).
> >
> > We have edited this characterization to be more precise, and say that models experience a significant drop in worst-group accuracy when I(A; Z) is comparable to or higher than I(Y; Z). We have also cleaned up and updated Fig 3c by only reporting the representation metrics for the final trained ERM models to make this more clear.
> >
> > ***Notation:*** Thanks for the feedback - we have revised these notation issues in the updated draft.
> >
> > [1] Sara Beery, Grant Van Horn, and Pietro Perona. Recognition in terra incognita. In Proceedings of the European Conference on Computer Vision (ECCV), pp. 456–473, 2018.
> >
> > [2] Nimit Sohoni, Jared Dunnmon, Geoffrey Angus, Albert Gu, and Christopher Ré. No subclass left behind: Fine-grained robustness in coarse-grained classification problems. In Advances in Neural Information Processing Systems, volume 33, pp. 19339–19352, 2020.

---

### Official Review · Reviewer_FMKR · 2021-11-01

**Correctness:** 2
**Technical Novelty And Significance:** 2
**Empirical Novelty And Significance:** 3
**Recommendation:** 5
**Confidence:** 4

**Main Review:**

Strengths
* This paper focuses on improving model robustness to group shifts in a practical setting where group information is not available. The proposed method achieves SOTA worst-group performance to be close to GDRO which uses the true group labels.
* Some interesting empirical analysis are presented for relating the representation alignment with worst-group performance.

Weakness
* In general, the observation is not surprising, and the idea of aligning representation for improving model robustness is not novel. There are a lot of work with similar ideas in domain generalization/adaptation literature, e.g., [1], [2]. There’s also a recent work [3] that applies contrastive learning for doing so. A more comprehensive discussion for these related work needs to be included.
* The assumption of Theorem 3.1 is not well explained and motivated. In particular, the assumption that “the loss function l(x; y) is 1-Lipschitz in x and bounded from above by one.” seems to be necessary and simplify the proof a lot, but does not hold for typical losses like cross-entropy for classification and MSE for regression.
* Though the proposed contrastive method leads to improved worst-group performance, it seems to decrease the average-case performance compared to baselines. More crucially, neither part of the two-stage method is justified with sufficient motivation and empirical evidence, as detailed below:
    * Using ERM prediction as the group label is not convincing enough, and it is not clear how it would affect the contrastive part. It could be interesting to more extensively analyze how the label prediction affects the improvement given by the contrastive method, probably using a scientific setup where the label prediction is controlled.
    * For the contrastive part, the current empirical comparison obfuscates the advantage on its own. To decouple it from the effect of wrong group prediction, it is important to compare in the setting where group labels are available, i.e., GDRO vs GDRO + contrastive. Also, there could be a lot of choices of negative selections but only one is used without sufficient explanations, it would be great to include more explanation or compare with some other possible choices as an ablation study.


Additional questions & comments
* In the last paragraph of introduction, it is claimed that “...only falling short of GDRO’s worst-group accuracy by *0.3%* absolute…”. However, in Table 1, the gaps between CNC and GDRO are 1.1%, 1.4%, 0.1%, 0.7%, respectively on each dataset, how is the 0.3% gap calculated?

[1] [Domain-Adversarial Training of Neural Networks](https://arxiv.org/abs/1505.07818)

[2] [Domain Generalization with Adversarial Feature Learning](https://openaccess.thecvf.com/content_cvpr_2018/papers/Li_Domain_Generalization_With_CVPR_2018_paper.pdf)

[3] [Cross-domain Contrastive Learning for Unsupervised Domain Adaptation](https://arxiv.org/pdf/2106.05528.pdf)




**Summary Of The Paper:**

This paper focuses on improving the model robustness to group shifts without prior group information and bridges the gap to those methods with access to group labels (i.e., GDRO). It identifies the relation between the worst-group performance and representation alignment both empirical and theoretically, which motivates a contrastive approach for improving representation alignment and robustness. Empirically, the proposed method demonstrates improved worst-group performance over existing baselines.

**Summary Of The Review:**

Overall, the observation of the paper is not novel and the theoretical analysis is rather weak. Though the proposed method leads to improved worst-group performance, it is not well developed and supported with sufficient empirical evidence. Thus, I recommend a ‘reject’ for the paper.


++++++++ Post-rebuttal ++++++++

I will increase my score to 5, but I still think the paper can be improved a lot before acceptance as detailed [here](https://openreview.net/forum?id=cVak2hs06z&noteId=rfOvGo4VyL9).

---

> ### Author Response · Authors · 2021-11-20
> **Summary of response to Reviewer FMKR (1/6)**
>
> Thanks for the reviewer’s detailed review. While we are grateful for the detailed feedback, we respectfully disagree with some of the reviewer’s comments, including in particular: “Several of the paper’s claims are incorrect or not well-supported.” and “The technical and empirical contributions are only marginally significant or novel.”, which we do not find sufficient justification for in the original review.
>
> We point out that the proposed domain adaptation / domain generalization methods do not apply to our setting, as they **assume domain labels are known** (whereas we need to first determine what defines a domain, and subsequently infer these groups since we do not assume access to the group labels), and seek to accomplish different tasks (i.e., generalizing to a target domain without class labels). We were still interested in how these ideas would perform on our benchmarks, and updated the paper with empirical comparisons with DANN [1] using proper adjustments for the spurious correlations setting in Table A.2 and Section A.2. We observe that our proposed approach CNC outperforms DANN by $22.4$% worst-group accuracy on Waterbirds and by $41.6$% on CelebA. In the revised paper, we have cited the works mentioned by the reviewer and discussed the difference between our setting and these works in Section D.4).
>
> Second, we added additional empirical comparisons to study the contrastive component of CNC. We consider additional negative sampling strategies as suggested, and added discussion and results in our updated draft. To better understand the advantage of the Stage 2 contrastive learning step, we are also running experiments comparing CNC with known group labels versus GDRO.
> - We point out that in our original submission we did “analyze how the [ERM] label prediction affects the improvement” of CNC in Sec. 5.3 and Fig. 7d, where we controlled the spurious attribute label prediction and compared the effect of worsening prediction to model worst-group and average performance. We find that CNC is less sensitive to inaccurate ERM label predictions than JTT.
>
> In our following comments we elaborate on the above, and include responses to additional questions and concerns.

---

> > ### Author Response · Authors · 2021-11-20
> > **Updated paper with discussion on domain generalization / adaptation methods that seek to align representations (2/6)**
> >
> > Thank you for bringing up these related works that seek to align representations to improve domain generalization (DG) / (unsupervised) domain adaptation (UDA). We have updated our related works section to include discussion of the cited works. In our updated draft and our response below, we:
> > 1. Explain why DG and DA methods do not readily apply to our setting
> > 2. Highlight differences between the cited works (DANN [1], MMD-AAE [2], and CDCL [3]) and CNC
> > 3. Explain why our work on aligning representations is a novel contribution to the important problem of combating spurious correlations without requiring spurious attribute labels.
> >
> > **Domain adaptation / generalization vs. spurious correlations**
> > We note that the suggested approaches are not readily in-scope with our paper setting, and contend that CNC is still novel with respect to the existence of these works. First, the cited DG / UDA approaches study a completely different setting (assuming a predefined notion of domains and aiming to generalize beyond a single domain). Importantly, they require knowledge of training samples’ domains, which means they **need group labels to be known** in the spurious correlations setting we study. Improving robustness to spurious correlations when group labels are known is well-studied (e.g., GDRO, as well as other prior works such as [4]). We focus on the more difficult problem of robustness to spurious correlations **without group label knowledge**.
> >
> > Second, the notion of a “domain” must be suitably specified for our setting. For example, in Waterbirds we could treat all samples with the same spurious attribute as belonging to the same domain (e.g., the “water background” domain). We could alternatively treat all samples that exhibit the dominant spurious correlation as a domain (e.g., the “waterbird, water background OR landbird, land background” domain).
> >
> > Finally, even if we first infer the “domains”, the cited DG and UDA approaches are designed for different evaluation settings from our spurious correlations setting.
> >
> > We discuss these differences in detail in our updated further related work section (Sec. D.4). Beyond not assuming group labels, we also try to improve performance on the worst-group at test-time, which could belong to *any* domain. Finally, our setting provides a somewhat different distribution shift challenge. Due to spurious correlations, the class distribution is highly skewed within each domain, e.g. in Waterbirds an ERM model may be inclined to always predict the landbird class for a “land background” domain because 95% of images in that domain have the landbird class. A robust model must overcome such correlations that actively bias the model to make an incorrect prediction based only on the domain. This is not present in canonical DG / UDA datasets, where the class distributions per domain in standard benchmarks such as Office-31 [5], PACS [6], and VLCS [7] are much more uniform.
> >
> > **Methodological differences with CNC**
> > CNC also uses a different alignment procedure than the cited methods. We have updated our related works (Sec. D.4) to discuss these similarities and differences. In particular, we note that CNC’s approach to aligning representations is simpler than MMD-AAE or DANN, avoiding the use of multiple module components and adversarial training.
> >
> > We thank the reviewer for bringing CDCL to our attention, which we think is an interesting concurrent work that aligns with the larger direction of applying contrastive learning (and more specifically a SupCon-style objectives) in new ways to improve model robustness. However, because CDCL tackles UDA, their setup and task is again fundamentally different (we have no notion of “source” and “target” domains, and must determine and infer domain labels / spurious attribute values for all training samples). Despite its recency (arXiv upload on 6/10/2021, within 4 months of ICLR submission deadline), CDCL does not evaluate on any of our existing benchmarks or compare against any of our existing baselines. Finally, CDCL also contains components that are irrelevant to our problem setting (e.g. a central contribution of CDCL is a step to infer *ground-truth class* labels for an unlabeled subset of the training data, whereas in our setting we assume the classes are known but not the groups).
> >
> > **Novelty of representation alignment in our problem setting**
> > We clarify that we do not claim to introduce representation alignment as a method to improve model robustness in general; rather we specifically introduce representation alignment between specific data subsets for the important property of robustness to spurious correlations (as defined in Sagawa et al. ICLR’20). This is a new approach with respect to the established line of work focusing on handling spurious correlations and subpopulation shifts (GDRO, LfF, GEORGE, JTT), which largely aims to upweight minority groups within a cross-entropy-based objective during training.

---

> > > ### Author Response · Authors · 2021-11-20
> > > **Updated paper with empirical comparison with DG / UDA methods, clarifications on Theorem 3.1 (3/6)**
> > >
> > > **Empirical comparison with DG / DA methods with suitable adjustments**
> > > For completeness, we also compare DANN empirically to CNC, measuring worst-group robustness on the real-world Waterbirds and CelebA benchmarks using the same ResNet-50 backbone. These results are in Table A.2 of our updated draft (also reported below). Due to time constraints, we will include the results for an adjusted MMD-AAE [2] for our datasets and evaluation setting in the next revision.
> > >
> > > Because we are not generalizing to a completely unseen domain, nor adapting from a target domain to a source domain, and we do not know which samples belong to which domains, this evaluation requires several adjustments to the methods. Details for how we adapt the DG / DA methods are in Section A.2 of the updated draft. At a high level, we first group training data into inferred domains using the same initial ERM model used in CNC for each dataset. We consider domain splits both based on whether samples share similar spurious attribute features (e.g. water background domain) and whether samples follow the dominant spurious correlation (e.g. same bird-type and background-type domain).
> > > For DANN, we trained the model to be invariant across the inferred splits, but also allow class labels for both domains.
> > > For MMD-AAE, we will train the model to be invariant across these inferred domain splits, and evaluate on the same standard test-set for worst-group accuracy.
> > >
> > > |                                              |  Waterbirds |             |    CelebA   |            |
> > > |----------------------------------------------|:-----------:|:-----------:|:-----------:|:----------:|
> > > |                                              | Worst-group |   Average   | Worst-group |   Average  |
> > > | DANN (domains by spurious attribute)         |  37.4 (3.8) |  87.6 (2.2) |  28.1 (3.1) | 94.6 (0.3) |
> > > | DANN (domains by majority vs minority group) |  67.3 (0.8) |  83.6 (0.2) |  47.2 (3.1) | 88.7 (1.8) |
> > > | CNC                                          | 89.7 (0.2)  | 90.8 (0.1)  | 88.8 (0.9)  | 89.9 (0.5) |
> > >
> > > Our results suggest that prior approaches for domain alignment are not sufficient to improve worst-group accuracy; one reason may be because they try to align representations without regard to different classes within each domain. DANN has multiple components that must be jointly trained together to obtain a robust model, and crucially requires careful balancing between a cross-entropy objective to maintain class-separability and an alignment term to ignore domain-specific or spurious features. On the other hand, we only train a single model in CNC during our robust training step. While we use both a cross-entropy and a contrastive loss, the contrastive component and hard sampling of CNC allows us to learn class-separability and spurious feature invariance at the same time.
> > >
> > > **“The assumption of Theorem 3.1 is not well explained and motivated.”**
> > > Thank you for raising this concern. The reviewer is correct that assuming the loss is 1-Lipschitz in x and bounded from above by one simplifies the proof, although these assumptions are not essential to the analysis. In the revised paper, we have extended the proof of Theorem 3.1 to cases where the loss function is Lipschitz continuous for any fixed constant and upper bounded by a fixed constant as well. This is achieved by taking care of the Lipschitz constant (in equation 11) and the boundedness constraint (in equation 9), while keeping all other parts the same. This extension now applies to loss functions such as the cross-entropy loss and the mean squared loss (that the reviewer suggested), as long as the input data comes from a bounded domain. Our experimental settings satisfy these conditions.

---

> > > > ### Author Response · Authors · 2021-11-20
> > > > **Clarification on CNC main performance and dependence on ERM predictions (4/6)**
> > > >
> > > > **CNC seems to decrease the average-case performance compared to baselines**
> > > > We respectfully disagree with this characterization. All methods decrease average-case performance w.r.t. ERM (including GDRO, our “upper baseline” for worst-group accuracy which assumes labels). This is because there is often a tradeoff between worst-group and average accuracy: the model that attains the best average accuracy is not necessarily the one that attains the best worst-group accuracy, and vice versa. However, in terms of accuracy, CnC’s exceeds that of JTT (the next best SoTA for worst-group accuracy when group labels are unknown) and GDRO for CMNIST, (90.9% CNC vs 90.2% JTT, 90.6% GDRO). We also beat or achieve very close average performance with comparable methods on CelebA (e.g. 89.9% CNC vs 88.0% JTT). Comparing additional methods and across benchmarks, it is not true that we consistently reduce average performance: e.g. we obtain the highest average accuracy on CMNIST, which exhibits very strong spurious correlations, while we obtain the lowest average accuracy on CivilComments, which exhibits relatively weaker spurious correlations. On CelebA, we obtain the median average accuracy. However, for our primary goal to improve worst-group accuracy, we obtain SoTA performance among methods that do not require group labels on 3 out of 4 datasets.
> > > >
> > > > **Using ERM prediction as the group label is not convincing enough, and it is not clear how it would affect the contrastive part. It could be interesting to more extensively analyze how the label prediction affects the improvement given by the contrastive method.**
> > > > Thanks for the suggestion. We would like to emphasize that since group labels are not available during training in our setting, one needs to infer them first. Using the predictions of a (properly-regularized) ERM model to infer group labels is a well-known and simple technique in the context of spuriously-correlated datasets. For example, variants of this idea have been successfully applied in a number of state-of-the-art methods including Sohoni et al. (NeurIPS ‘20), Nam et al. (NeurIPS ‘20) and Liu et al. (ICML ‘21). We also empirically verify that ERM predictions are strongly predictive of the spurious attributes (UMAPs of feature representations - Fig. 2, Fig. 6, Fig. 11; quantifying representation metrics - Fig. 3; GradCAM saliency maps - Fig. 12).
> > > >
> > > > We studied how the ERM model’s predictions affect CNC’s performance in Sec 5.3 (cf. Fig. 7d). As suggested, we performed a systematic ablation reporting worst-group accuracy with the contrastive model while degrading the quality of the inferred group labels. We started with “perfect” ERM predictions, using the actual spurious attribute labels to set up our contrastive batches, and then added noise to these “predictions” by replacing the true spurious attribute label with another uniformly-randomly sampled one, doing so independently with some probability $p$. We reported the same downstream metrics over $p$ = {0, 0.01, 0.1, 0.25} (the true group run being $p = 0$) in Fig. 7d of the original submission.
> > > >
> > > > As shown in Fig. 7d, worst-group accuracy degrades as $p$ increases (i.e., the “pseudo group prediction” quality decreases). This limitation is also shared by methods such as JTT. However, compared to JTT, CNC is more robust to the inferred group labels being worse quality. We think that coming up with new methods to better infer the group labels is an interesting question for future work.
> > > >
> > > > **To decouple contrastive learning from the effect of wrong group prediction, it is important to compare in the setting where group labels are available.**
> > > > Thank you for this helpful suggestion. We will update the paper to include the worst-group and average accuracies obtained by CNC, when we know the group labels and can use these labels to set up our contrastive batches (we are currently running these experiments).
> > > >
> > > > We note that our prior evaluation in Sec. 5.3 does decouple the effects of the ERM model’s wrong-group prediction, as other methods such as JTT also rely on an initial ERM model’s predictions to help infer groups and train another robust model. Controlling for these differences in "stage 2", the contrastive component of CNC seems to be responsible for the improved worst-group performance compared to JTT.

---

> > > > > ### Author Response · Authors · 2021-11-20
> > > > > **Additional explanation for CNC negative sampling; added experiments for alternate approaches (5/6)**
> > > > >
> > > > > **More explanation and comparisons with possible negative sampling strategies**
> > > > > We reiterate that as discussed in Sec. 4 and B.3, our negative sampling approach gives us a strong signal to learn representations that only depend on the class-specific information. As discussed in Sec 3, trained ERM models are susceptible to learning spurious correlations, so their predictions act as proxies for the spurious features. If two samples have the same ERM prediction, it is likely that they have similar spurious features. Because our goal is to avoid learning spurious correlations, and we aim to do so by training a model to learn representations that only depend on class-specific features and not spurious ones, it is advantageous to not only use contrastive learning to encourage a model to “pull together” (and learn similar) representations for samples with the same class but different spurious features, but also to encourage the model to “push apart” (and learn dissimilar) representations for samples with different classes but similar spurious features.
> > > > >
> > > > > To support this reasoning, we have added additional ablations with alternative negative sampling strategies in Table A.3 of the updated draft. We consider (1) negative sampling by only having a different class as the anchor (negative by class), (2) negative sampling only based on having the same ERM model prediction, but not controlling for different ground-truth classes (negative by prediction), and (3) the negative by class sampling ablation where we also only sample positives as points with the same class as the anchor (similar to SupCon). As shown in our results below, our original and proposed sampling approach substantially outperforms these alternate sampling strategies in worst-group and average accuracy across the CMNIST, Waterbirds, and CelebA datasets.
> > > > >
> > > > > |                   |    CMNIST   |            |  Waterbirds  |             |    CelebA   |            |
> > > > > |-------------------|:-----------:|:----------:|:------------:|:-----------:|:-----------:|:----------:|
> > > > > |                   | Worst-group |   Average  |  Worst-group |   Average   | Worst-group |   Average  |
> > > > > | Negative by class |  66.4 (5.1) | 86.0 (1.6) |  82.2 (0.8)  |  88.9 (0.3) |  79.2 (0.3) | 88.0 (0.1) |
> > > > > | Negative by pred  |  70.0 (5.1) | 87.1 (1.1) |  85.7 (1.3)  |  90.3 (0.2) |  81.1 (1.4) | 88.5 (0.3) |
> > > > > | SupCon           |  0.0 (0.0)  | 22.4 (1.2) | 71.0 (1.92)  |  85.9 (0.8) |  62.2 (1.1) | 90.0 (0.1) |
> > > > > | CNC (Default)           | 77.4 (3.0)  | 90.9 (0.6) |  89.7 (0.2)  | 90.8 (0.1)  | 88.8 (0.9)  | 89.9 (0.5) |
> > > > >
> > > > >
> > > > > We hypothesize that the decrease in performance is because without this sampling, we may actually encourage the Stage 2 model to rely on spurious correlations. For example, if we just ensure that the anchor and negative samples have different classes, then because of spurious correlations the contrastive model may just rely on the different spurious features of the anchors and negatives to learn dissimilar representations. However, by ensuring that the anchors and negatives have similar spurious features (via the same trained ERM model prediction), then the contrastive model is forced to rely on non-spurious features to learn dissimilar representations for the samples. Sampling this way is also a natural form of hard negative mining while avoiding class collisions, two key challenges in contrastive learning, which when resolved, lead to better performance [8, 9].
> > > > >
> > > > > **Additional questions & comments**
> > > > > Apologies for the typo, and thank you for catching this. We have corrected this to be 0.8%.

---

> > > > > > ### Author Response · Authors · 2021-11-20
> > > > > > **Citations for Response to Reviewer FMKR (6/6)**
> > > > > >
> > > > > > [1] Yaroslav Ganin, Evgeniya Ustinova, Hana Ajakan, Pascal Germain, Hugo Larochelle, FrançoisLaviolette, Mario Marchand, and Victor Lempitsky. Domain-adversarial training of neural networks. The journal of machine learning research, 17(1):2096–2030, 2016.
> > > > > >
> > > > > > [2] Haoliang Li, Sinno Jialin Pan, Shiqi Wang, and Alex C Kot.  Domain generalization with adversarial feature learning. In Proceedings of the IEEE Conference on Computer Vision and PatternRecognition, pp. 5400–5409, 2018.
> > > > > >
> > > > > > [3] Rui Wang, Zuxuan Wu, Zejia Weng, Jingjing Chen, Guo-Jun Qi, and Yu-Gang Jiang. Cross-domain contrastive learning for unsupervised domain adaptation. arXiv preprint arXiv:2106.05528, 2021.
> > > > > >
> > > > > > [4] Hidetoshi Shimodaira. Improving predictive inference under covariate shift by weighting the log-likelihood function. Journal of statistical planning and inference, 90(2):227–244, 2000
> > > > > >
> > > > > > [5] Kate Saenko, Brian Kulis, Mario Fritz, and Trevor Darrell. Adapting visual category models to new domains. In European conference on computer vision, pp. 213–226. Springer, 2010.
> > > > > >
> > > > > > [6] ​​Li, D., Yang, Y., Song, Y. Z., & Hospedales, T. M. (2017). Deeper, broader and artier domain generalization. In Proceedings of the IEEE international conference on computer vision (pp. 5542-5550).
> > > > > >
> > > > > > [7] Chen Fang, Ye Xu, and Daniel N Rockmore. Unbiased metric learning: On the utilization of multiple datasets and web images for softening bias. In Proceedings of the IEEE International Conference on Computer Vision, pp. 1657–1664, 2013.
> > > > > >
> > > > > > [8] J. Robinson, Ching-Yao Chuang, S. Sra, and S. Jegelka.  Contrastive learning with hard negative samples. In International Conference on Learning Representations, 2021.
> > > > > >
> > > > > > [9] M. Wu, M. Mosse, Chengxu Zhuang, D. Yamins, and Noah D. Goodman.  Conditional negative sampling for contrastive learning of visual representations. In International Conference on Learning Representations, 2021.

---

> > ### Comment · Reviewer_FMKR · 2021-11-24
> > **Thank you for the detailed response, I still have some concerns**
> >
> > Thank you for the hard work in making the detailed response and I appreciate it a lot. Some of my concerns are addressed. I still have some questions/concerns about the paper, as detailed below:
> >
> > ***Comparison to the DA literature***
> >
> > I agree that the DA literature considers a different setting, but my point was that the idea of representation alignment is already extensively studied in DA literature which needs more discussion.
> > I’m glad to see the detailed discussion in Sec. D.4, and I also hope the authors could discuss (or at least mention)  these related ideas in the main body for future versions.
> > I still don’t think the application of the essential idea of representation alignment (not the contrastive approach) in this specific setup is itself as novel as the authors claimed, and the claim of its novelty needs to be softened in the paper.
> >
> > I appreciate that the author did additional experiments to compare with (adjusted) DA methods (though I didn’t mean to ask for them). The results make sense to me because DANN requires unstable adversarial training and it does not use the available label information. As a side note, for a really fair comparison, “domain” needs to be defined by majority vs minority group (and I actually don’t think the definition of “domain” is not clear for it) and the conditional version of DANN [1] which utilizes the labels should be compared. This could be a really good baseline to illustrate the effectiveness of the proposed contrastive approach.
> >
> > [1] Li, Ya, et al. "Deep domain generalization via conditional invariant adversarial networks." Proceedings of the European Conference on Computer Vision (ECCV). 2018.
> >
> >
> > ***Novelty of the contrastive approach***
> >
> > I now see that adapting the contrastive loss to the setup with specific sampling strategies is novel, but I don’t think it’s clearly highlighted in the current version of the paper. To make it more clear, a more detailed discussion needs to be included to motivate and justify the current sampling strategy when it is introduced in Sec 4.1, and the ablation study of different strategies needs to be included in the main body as well in future versions.
> >
> > ***Justification of the contrastive approach***
> >
> > Unfortunately, the result of the contrastive approach when group information is available is still missing in the response, which is the **most important** to justify the effectiveness of the proposed contrastive approach. The whole Sec 3 is developed in the setup where the **ground-truth** group information is available and it motivates the contrastive approach. To verify that, the comparison really needs to be all done in the setup where the group is available rather than inferred. I don’t understand why the proposed approach is evaluated in the current setup and whether/how the contrastive approach benefits it (please correct me if I miss some important points).
> >
> >
> > ***The assumption of Theorem 3.1***
> >
> > Why do cross-entropy loss and the mean squared loss satisfy boundedness constraint?
> >
> >
> > ***The tradeoff between average and worst-group accuracies***
> >
> > In the authors’ response, it is claimed that “there is often a tradeoff between worst-group and average accuracy”. Why is it necessarily true? In the ideal case, if the representation is perfectly aligned within the same class and separated between classes, the worst-group and average accuracies should be equal and both high. Theorem 3.1 also shows that the alignment loss upper bounds the gap but does not indicate any tradeoff. Or is it just an (unjustified) assumption that seems to be true for previous methods?

---

> > > ### Author Response · Authors · 2021-11-28
> > > **Addressing your remaining concerns (1/3)**
> > >
> > > Thank you for this insightful discussion. We are glad to hear that some of your concerns have been addressed with our responses and updates. In light of this, we ask the reviewer to consider increasing their score. We also hope to address your remaining questions and concerns to further improve the paper and reach a rating consensus.
> > >
> > > Please find our responses to your remaining comments below:
> > >
> > > **Comparison to the DA literature**
> > > Thanks for this additional feedback on our comparison to DA methods. We regret not being able to update the draft anymore, but in the next version of the paper we will make note of these existing distance regularization methods for domain adaptation / domain generalization in the introduction (page 2) and exposition of the methods we compare against in our main results (page 7). We will also make clear that our contribution is our sampling strategy + contrastive loss for the spurious correlations setting, which is to increase alignment between groups with the same class, while also preserving separability between samples with different classes.
> > >
> > > **Comparison to DANN**
> > > We agree that your suggested approach is a good baseline. In fact, for fair comparison, in Table A.2 we did exactly this, comparing against a version of DANN trained with domains “defined by majority vs minority group” which “utilizes the labels” for both training domains. For the camera-ready, we are happy to move this result to the main results in Table 1 (along with the prior main body updates regarding DG / DA methods).
> > >
> > > **Novelty of the contrastive approach**
> > > Thanks for your helpful feedback. We will update the next version of our paper following your suggestions. In Sec. 1 (Intro) and Sec. 4 (presenting the method) we will not only clarify the reasoning behind this sampling, but also discuss why alternate strategies are less preferable (e.g. they can still encourage the contrastive model to rely on spurious feature similarities when “pushing” anchor-negative pairs apart). We will also update Sec. 5.3. to include the sampling strategy ablations (Our plan is to make Table A.3 a figure that can replace Fig. 7c).
> > >
> > > **Justification of the contrastive approach**
> > > We would like to clarify some things that may have been misunderstood, and apologize for the confusion. First, it’s not quite right to say that “the whole Sec 3 is developed in the setup where the ground-truth group information is available and it motivates the contrastive approach”. None of our empirical observations use this group information; in Sec. 3.1, we run models trained with ERM or JTT (using inferred groups) on CMNIST. We use these empirical observations which do not rely on knowing the group labels to first motivate our contrastive approach - stated in the last sentence of Sec. 3.1. The theory that follows in Sec. 3.2 (which as correctly stated, involves group labels) seeks to formalize this empirical intuition and answer some of our motivating hypotheses (restated below). In the process this further motivates CNC.
> > >
> > > To recap, Sec. 3.1 shows that as we increase the dataset’s correlation between color and digit, for ERM models, our metric of interest (worst-group accuracy) decreases, and other representation metrics (that we can optimize for) also follow consistent trends (e.g., alignment loss goes up, mutual information with ground-truth class labels drops, Fig. 3 - for discussion purposes, let’s say that when the representation metrics move in these directions, they “worsen”). Liu et al, ICML 2021 [1] showed that JTT improves worst-group accuracy over ERM, and we observe that JTT’s improved worst-group accuracy also goes hand-in-hand with improved representation metrics.
> > >
> > > However, we noted that JTT does not explicitly act on the representations, and that there still seems to be room for improvement via increasing worst-group accuracy. This connection between worst-group accuracy and representation metrics, *and* the lack of a representation-optimizing method for the spurious correlations setting with inferred group labels, thus motivates our contrastive approach, as we hypothesize that:
> > > 1. A method that aims to “move” the representations (instead of the upweighting used by prior work like JTT) and more explicitly act on these representation metrics can improve these representation metrics
> > > 2. This representation metric improvement will continue to correspond with improved worst-group accuracy.
> > >
> > > We discuss this in our Introduction, and again state this motivation for our contrastive approach in the last sentence of Sec. 3.1.
> > >
> > > Then Sec. 3.2. formalizes this observation and connection between worst-group error and the alignment loss with Theorem 3.1, and allows us to theoretically validate hypothesis 2 above. Note that while Theorem 3.1. defines groups $g, g’$ to refer to true group labels, we can very simply swap this definition for *inferred* groups. The proof’s logic still holds (re: the worst *inferred* group error) to motivate CNC.

---

> > > > ### Author Response · Authors · 2021-11-28
> > > > **Addressing your remaining concerns, additional empirical comparisons (2/3)**
> > > >
> > > > ***“I don’t understand why the proposed approach is evaluated in the current setup and whether/how the contrastive approach benefits it”***
> > > > (Continued from the above response). From this logic, our contrastive approach can thus be used in any situation where we are given some way to organize samples into groups, whether we infer the group labels (which is the problem we tackle in this paper), or we assume access to the ground-truth group labels. We tackle the former because of its practical importance (group labels may be unknown, or we may not always be able to label every datapoint’s group), and less progress has been made in this more challenging setting.
> > > >
> > > > To clarify how the contrastive approach benefits this current setup, we emphasize that JTT, GEORGE, LfF, PGI, and EIIL *all* use similar approaches to first infer group labels (an initial properly-regularized ERM model’s predictions - see Footnote 1, bottom of page 5; Related Work [Section 6]; Further Related Work [Appendix D]). Key differences lie in what these methods do with the inferred labels. The superior worst-group accuracy of CNC (Table 1) suggests that our contrastive approach does benefit this setting (e.g. GEORGE and EIIL run GDRO on the inferred group labels - CNC outperforms both for all datasets).
> > > >
> > > > To further isolate how the contrastive component of CNC can benefit the setting where we have to infer groups, we additionally compared GDRO with CNC when both use the exact same *inferred* group labels on the Waterbirds dataset (see below for a comparison using the true group labels). Across 3 independent seeds we observed that CNC still outperforms GDRO by 2.6% in worst-group accuracy:
> > > >
> > > > |                            | Worst Group Acc. (%) | Average Acc. (%) |
> > > > |----------------------------|:--------------------:|:----------------:|
> > > > | GDRO (CNC inferred groups) |      87.1 (0.5)      |    93.2 (0.4)    |
> > > > | CNC                        |      **89.7 (0.2)**      |    90.8 (0.1)    |
> > > >
> > > > ***“The comparison really needs to be all done in the setup where the group is available rather than inferred”***
> > > >
> > > > As requested, we ran additional experiments with CNC using the *ground-truth* group labels, comparing against default GDRO which requires group labels. Now CNC outperforms GDRO by 2.1% for worst-group accuracy and 1.8% for average accuracy on CMNIST*, while underperforming GDRO by a smaller margin on Waterbirds (1% absolute for worst-group accuracy, roughly the same for average accuracy):
> > > >
> > > > |                             |     CMNIST*    |            |   Waterbirds   |            |
> > > > |-----------------------------|:--------------:|:----------:|:--------------:|:----------:|
> > > > | Method / Accuracy (%)       |   Worst-group  |   Average  |   Worst-group  |   Average  |
> > > > | GDRO                        |   78.5 (4.5)   | 90.6 (0.1) | **91.1 (0.2)** | 92.4 (0.2) |
> > > > | CNC (inferred group labels) |   77.4 (3.0)   | 90.9 (0.6) |   89.7 (0.2)   | 90.8 (0.1) |
> > > > | CNC (true group labels)     | **80.6 (2.8)** | 92.4 (0.2) |   90.1 (0.2)   | 92.4 (0.2) |
> > > >
> > > > To interpret these results, we note that CMNIST* carries a stronger spurious correlation than Waterbirds, where only 0.5% of all samples lie in minority groups for CMNIST* as opposed to the 5% in Waterbirds. Thus one possible takeaway--beyond that CNC also works with true group labels--is that CNC may be preferable in settings with stronger spurious correlations. While just a conjecture right now, we think further studies into how different methods scale with the strength of the spurious correlation is an interesting direction for future work. We will update the paper to include these results as well as more comparisons for additional datasets.

---

> > > > > ### Author Response · Authors · 2021-11-28
> > > > > **Addressing your remaining concerns (3/3)**
> > > > >
> > > > > **“The assumption of Theorem 3.1. Why do cross-entropy loss and the mean squared loss satisfy boundedness constraint?”**
> > > > >
> > > > > Thanks for the follow up question and we are happy to provide more explanations.
> > > > >
> > > > > First we show that for our setting, the cross-entropy loss is upper bounded by $log(1 + (K-1) \exp(2B))$, where $K$ is the number of distinct classes we are trying to predict in the output layer. To see this, recall that the feature representation vectors all have unit norm under CNC’s framework. Since the operator norm of the linear classification layer is at most $B$ as stated, the prediction value for every class (before they are transformed to probability value using softmax) is at most $B$, because this prediction value is the product between a particular row of the linear matrix and the feature vector. Thus, the cross-entropy loss is applied to a vector whose entries are all bounded by $B$. In this scenario, one could verify that the cross-entropy loss must be less than $\log(1 + (K-1) \exp(2B))$, which is approximately $2 log(K) B$.
> > > > >
> > > > > Second, while we did not consider mean squared loss in our experiments, this loss is upper bounded by $4 B^2$. To see this, note that for unit length vectors $u$ and $v$, if the final output layer $W$’s (which is now a vector for regression) norm is upper bounded by $B$, then, $|\langle W, u\rangle - \langle W, v\rangle|^2 \le 2|\langle W, u\rangle|^2 + 2|\langle W, v\rangle|^2 \le 4 B^2$.
> > > > >
> > > > >
> > > > > **The tradeoff between average and worst-group accuracies**
> > > > > We agree with the reviewer that in an ideal case, the neural network can learn perfectly aligned representations and achieve high average performance and worst-group performance (e.g. by learning a “true” correlation that applies to all groups (and datapoints) for classification).
> > > > >
> > > > > However, in practice finding such perfectly separating representations on spuriously correlated datasets, such as Waterbirds and CelebA, has been challenging. Our claim that “there is *often* a tradeoff between worst-group and average accuracy” was in reference to this, where empirically it is typically observed that methods which improve worst-group accuracy also decrease average accuracy (GDRO [2], JTT [1], other reported methods in Table 1). Also note that in general, if a problem is not realizable, the minimizers of the worst-group loss and average loss will not necessarily be the same.
> > > > >
> > > > > We believe it is an interesting research question to study whether such tradeoff is necessary or not, as suggested by the reviewer. For now, one possible explanation for this tradeoff empirically could be that by encouraging a network to focus on the “non-spurious” features (e.g. the actual bird pixels for Waterbirds), we give the network a more difficult classification task than just classifying based on the spurious features (e.g. the background pixels). Otherwise, an ERM model would have learned to classify by the non-spurious features, i.e. the non-spurious “true” correlation is harder to learn (c.f. Hermann & Lampinen, NeurIPS 2020 [3]). With fixed model capacity, the harder task to classify by focusing on the non-spurious features could thus lead to more misclassifications on the majority group at test-time, resulting in reduced average accuracy overall (albeit improving performance on the minority group, as the non-spurious learned correlation holds for all groups).
> > > > >
> > > > > [1] Liu, Evan Z., et al. "Just train twice: Improving group robustness without training group information." International Conference on Machine Learning. ICML, 2021.
> > > > >
> > > > > [2] Sagawa, Shiori, et al. "Distributionally robust neural networks for group shifts: On the importance of regularization for worst-case generalization." In International Conference on Learning Representation. ICLR, 2020.
> > > > >
> > > > > [3] Hermann, Katherine L., and Andrew K. Lampinen. "What shapes feature representations? exploring datasets, architectures, and training." Advances in Neural Information Processing Systems, 2020.

---

> > > > ### Comment · Reviewer_FMKR · 2021-11-30
> > > > **I will increase my score, but I still think the paper can be improved a lot before acceptance**
> > > >
> > > > Thanks for the detailed response. As some of my concerns are addressed, I will increase my score to 5. However, I still think the paper can be improved a lot before acceptance (which requires non-trivial work for revision), as discussed below.
> > > >
> > > >
> > > > **Major concern**
> > > >
> > > > While I acknowledge the empirical improvement of the proposed method in terms of the worst-group performance in the practical setup considered, my major concern, i.e., the justification of the proposed contrastive approach and its gap to the empirical study, still remains.
> > > >
> > > > In Sec 3.1, the alignment loss is defined w.r.t. **true** group label. Although the models used for empirical analysis do not rely on group labels, the measurement of both alignment loss and mutual information still uses **true** group label. Furthermore, the theory is developed in the case where **true** group label is available; I don’t agree with the implication of the statement “while Theorem 3.1. defines groups to refer to true group labels, we can very simply swap this definition for inferred groups” because it does not make sense to consider worst-group accuracy w.r.t. *inferred* groups instead of true ones. This is why I believe to justify the proposed approach, most empirical analysis should be done in the setup where true group labels are available. But as seen by the initial results, the proposed contrastive approach does not perform well on a real dataset Waterbirds. Though I totally agree that the setup where the group label is not available is much more practical, more convincing justifications and analysis for why the proposed method benefits in this setup seem to still lack.
> > > >
> > > > Minor about the implemented DANN baseline (I’m not sure whether the authors or I misunderstand something): I was suggesting to use the *conditional* version of DANN which uses the label for not only the classifier *but also the bottleneck*, i.e., the domain classifier could utilize the label information. Compared to the original version, this could have the advantage of avoid removing more information than necessary from the representations. See the cited paper for details.
> > > >
> > > >
> > > > **Suggestions for future versions**
> > > >
> > > > I feel that the paper could be improved a lot from the current versions and would like to provide some suggestions for future revisions. In particular, I would consider these major edits:
> > > > * Compress Sec 3 for representation alignment;
> > > > * Highlight the novelty of the proposed contrastive approach compared to existing work, i.e., the specific sampling strategy. Include a detailed discussion for justifying the strategy and ablation study with comparison to other sampling strategies to the main body;
> > > > * Provide more extensive scientific study in the setup where true group labels are available, for justifying the effectiveness of the proposed approach. This would not only backup the method but also give an extra bonus of improving over GDRO in that setup;
> > > > * Provide more insightful explanation and analysis for why the proposed method may favor the current setup.
> > > >
> > > > It would be a very cool paper, if the proposed method is well motivated and justified in a more scientific setup, and lead to promising improvement of worst-group performance in the practical setup. I hope these suggestions could help improve the paper :)

---

> > > > > ### Author Response · Authors · 2021-11-30
> > > > > **Response to standing concerns**
> > > > >
> > > > > Thanks for your response and detailed feedback. We respond to your concerns regarding our proposed contrastive approach and our empirical study below.
> > > > >
> > > > > **“It does not make sense to consider worst-group accuracy w.r.t. inferred groups instead of true ones.”**
> > > > >
> > > > > We can still justify using CNC with the inferred group labels instead of the true group labels because in our spurious correlations setting, we can use the Stage 1 ERM model to infer group labels that often match the true group labels. If the inferred group labels are close enough to the true labels, then worst-group accuracy w.r.t. the inferred groups will also be close to the worst-group accuracy w.r.t. the true labels.
> > > > >
> > > > > In general, we do not expect to infer the groups perfectly, but we reason that we can estimate these group labels sufficiently because in our spurious correlations setting, an ERM-trained model naturally learns spurious correlations such that its predictions and representations actually align with the spurious attribute labels. This is empirically validated in Sec. 3, described as key motivation in our Introduction, and widely discussed in prior work [1, 2]. Recall that the combination of spurious attribute label and ground-truth class label defines the groups, so if we predict the spurious attribute label with high accuracy, we can infer the groups with high accuracy.
> > > > >
> > > > > In practice, we observe that the stage 1 ERM model estimates the group labels with 88.9% accuracy for CMNIST* and 94.7% accuracy for Waterbirds (measured by how well the Stage 1 ERM model predicts the spurious attributes). Again we note that the ERM model predictions are not perfect, but these accuracies are much higher than chance, and the high *actual* worst-group accuracy achieved by CNC with the *inferred* groups also suggests that we can infer the group labels sufficiently with the ERM-trained model. The alignment loss and mutual information are measured w.r.t. the true group labels, but this just shows that even with the inferred group labels, CNC can improve these metrics.
> > > > >
> > > > > The theorem we introduced (Theorem 3.1) is used to further support the empirical observation we made regarding alignment in Sec 3.1.
> > > > >
> > > > >
> > > > > **“More convincing justifications and analysis for why the proposed method benefits in this setup seem to still lack.”**
> > > > >
> > > > > We would like to add that our proposed approach is meant to work on top of inferred group labels the same way the most related prior methods that we compare against (GEORGE, LfF, JTT, EIIL, PGI) also rely on inferred groups. These prior works are also proposed for this setup, and we outperform them handily. We believe this empirically justifies CNC in this setup.
> > > > >
> > > > > The prior works also do not provide additional justification for why their methods benefit the setup without true group labels. For example, JTT aims to upweight the minority groups, PGI aims to minimizes KL divergences of the network’s softmaxed logits between same-class groups. However, they tackle the setting where the true labels are not available, and merely substitute these true labels with inferred group labels. We are thus a bit confused about the additional requested justification for our contrastive approach, as there is no more justification for why these prior methods benefit the setup when group labels are unknown.
> > > > >
> > > > > **“But as seen by the initial results, the proposed contrastive approach does not perform well on a real Waterbirds dataset.”**
> > > > > We respectfully disagree with this characterization - we note that CNC with true group labels outperforms GDRO on CMNIST* by 2 worst-group accuracy points, and underperforms it on Waterbirds by only 1 point. We reiterate how it is impressive that CNC does almost as well with the inferred group labels (also shown by the ablation experiment in Sec 5.3), in our main intended setting of improving worst-group performance when the group labels are not available.
> > > > >
> > > > > **Minor about the implemented DANN baseline**
> > > > > Thanks for the clarification on this. In the next version we will run experiments accordingly.
> > > > >
> > > > > **Suggestions for a future version.**
> > > > > Thanks for these suggestions. However we are curious what is meant by a “more extensive scientific study for justifying the effectiveness of the proposed approach”. In the final revision we will include group label results for CelebA and CivilComments. However we still believe that our extensive current results, comparing against ablations and other methods that infer group labels, show that our proposed approach is effective in an important problem setting.
> > > > >
> > > > >
> > > > >
> > > > > [1] Marco Tulio Ribeiro, Sameer Singh, and Carlos Guestrin. “why should i trust you?” explaining the predictions of any classifier. In Proceedings of the 22nd ACM SIGKDD international conference on knowledge discovery and data mining, pp. 1135–1144, 2016.
> > > > >
> > > > > [2] Sara Beery, Grant Van Horn, and Pietro Perona. Recognition in terra incognita. In Proceedings of the European Conference on Computer Vision (ECCV), pp. 456–473, 2018.

---

### Official Review · Reviewer_GgTx · 2021-11-01

**Correctness:** 3
**Technical Novelty And Significance:** 2
**Empirical Novelty And Significance:** 2
**Recommendation:** 5
**Confidence:** 4

**Main Review:**

## Strength
- The paper is mostly well written and easy to follow.
- Spurious correlation avoidance without assuming any prior knowledge of spurious features is an important problem with wide impact.

## Weakness
**Standard Deviations** are important when reporting worst group accuracy metric. For example, the smallest group in CelebA (with supervised group labels) has only around 100 examples.
Expect to see std. dev. from at least three runs for all the experiments.
Looks like all your numbers are from a single run, given the typical std. dev. on these datasets and metric, I cannot judge the significance of your results.

**More Comparisons** needed. There are some existing contrastive regularization based methods [1, 2], perhaps several more.
Authors should compare and argue the merits of theirs over others both intuitively or analytically and empirically.
[1] proposed to learn representations that minimize the divergence between predictions on examples of the same label class but different partition (group).
In that regard, I find this work very similar to [1] with some differences as below:
- [1] regularizes the differences in aggregate prediction probabilities while this paper minimize per-sample differences
- [1] looks at prediction probability differences and KL measure while this work looks at representation differences and Eucledian distance.
- [1] uses [3] for partitioning the dataset on spurious attribute while this work only uses the ERM trained base model.
The differences to me look only superficial.

**Ablation study** on importance of first stage ERM training is needed. The authors state that the prediction accuracy of the spurious attribute in the case of CelebA is only around 59%.
I wonder what is the significance of ERM base model at all and what would happen if we instead simply regularize the distances for any pair of points of the same true class (this would be similar to [2]).

**Sensitivity to Stage 1 prediction (Sec 5.3):**
I do not understand why in Fig. 7 (d), we see the average accuracy also decreasing with p.
I expect the average accuracy to remain stable or increase as the worst group accuracy deteriorates.
Both average and worst group accuracy decreasing could indicate optimization problems?
Also, can you intuitively explain why you expect CNC to be more robust than JTT to stage 1 predictions?
Also in Table 1, why is the Avg. accuracy of CNC so much worse than other methods on CivilComments dataset?

## Minor
- Around expression (5), inconsistency around use of hat: $\hat L_{align}$ and $L_{align}$.
- Again around the same expression, stick to using either $\hat L_{align}(f_\theta)$ or $\hat L_{align}(f_{enc})$.
- The theorem looks intuitive but I could not follow the proof due to notation difficulties.

## References
1. Ahmed, F., Bengio, Y., van Seijen, H. and Courville, A., 2020, September. Systematic generalisation with group invariant predictions. In International Conference on Learning Representations.
2. Arpit, D., Xiong, C. and Socher, R., 2019. Predicting with high correlation features. arXiv preprint arXiv:1910.00164.
3. Creager, E., Jacobsen, J.H. and Zemel, R., 2021, July. Environment inference for invariant learning. In International Conference on Machine Learning (pp. 2189-2200). PMLR.

**Summary Of The Paper:**

The paper studies training of models that are robust to spurious correlations without any supervision on spurious correlation revealing example groupings.
They observe that representations from an ERM model trained on such data, have strong correlation with the spurious attribute.
Their algorithm (CNC) is aimed at repressing spurious attribute information from the representations and was shown to be effective on standard sub-population shift datasets.
Also, CNC is shown to be more robust to noise in spurious attribute recovery than the existing methods.

**Summary Of The Review:**

I have novelty, significance and experiment rigor related concerns as detailed in my review.

---

> ### Author Response · Authors · 2021-11-20
> **Updated paper with additional comparisons, ablations, and clarifications (1/3)**
>
> Thank you for your careful and constructive review. We summarize our responses to the comments under “Weakness.”
>
> - First, we clarify that in Table 1, the reported results for CNC are already averaged over 3 random seeds, with the standard deviations previously reported in Table 11 (Sec. E.3). We have now updated Table 1 to include these standard deviations.
>
> - Second, following the reviewer’s suggestions, we have compared CNC against PGI [1] (and additional methods suggested by Reviewer FMKR) in the updated draft. We discuss our comparison to PGI (added to Table 1) further below. We note that despite the apparent similarities, CNC still outperforms PGI by 3.9 to 15.9 worst-group accuracy points on the benchmarks we consider. We are currently implementing comparisons to *Predicting with High Correlation Features* [2]. Our original submission also contained comparisons to EIIL [3] in our main results (Table 1), and we discussed the differences between CnC and EIIL in Sec. 6 and Sec. C.3 (now D.3 in the updated draft). We include additional baseline comparisons in our response to Reviewer FMKR and in Table A.2.
>
> - Third, we have updated the main results to include the performance of CNC without the Stage 1 ERM model’s predictions (denoted SupCon*, Table 1). In this case we only try to push together anchors and positives that share the same class, and push apart anchors and negatives with different classes (we do not account for different inferred spurious attributes). We observe worse worst-group performance, and include additional discussion in Sec. A.3. (along with other sampling procedure ablations).
>
> We believe that these results have addressed the issues that the reviewer pointed out; thus, we ask the reviewer to consider increasing their score. Next, we describe the above responses in more detail. Following this, we also respond to the other remaining questions below.
>
> **Including standard deviations**
> Apologies for the initial omission in the original Table 1. We have updated our main results in Table 1 with error bars for all datasets for CnC and Group DRO, as well as for all methods for CMNIST. For JTT, LfF, and CVaR DRO, we used the results reported in the JTT paper which unfortunately does not provide error bars (similarly, the CIM paper does not provide error bars). We are rerunning these results now and will include them in the final paper.

---

> > ### Author Response · Authors · 2021-11-20
> > **Updated paper with additional comparisons, ablations, and clarifications (2/3)**
> >
> > **Comparison to PGI**
> > Thank you for bringing the PGI method to our attention again. We added comparisons to PGI in Table 1 on CMNIST, Waterbirds, and CelebA, reporting the mean worst-group and average accuracies over 3 seeds. While we agree that PGI’s goal to learn invariant representations is similar to ours, the key difference and contribution of CNC is how we learn these invariances, in our case via a connection to contrastive learning and “hard” positive + negative sampling to train a model to ignore spurious correlations, which has not been done before.
> >
> > We first tried comparing against the standard PGI algorithm. However, even after extensive hyperparameter tuning (described further in Sec. E.2.5), we could not get PGI to work well (e.g., on Waterbirds, we obtained $51.0 \pm 4.9\%$ worst-group accuracy and $79.6 \pm 2.6\%$ average accuracy). We hypothesize this is due to the strong spurious correlations in our datasets: the PGI paper only considers datasets where 20% of the training samples do not exhibit a dominant correlation and fall under minority groups. Our evaluation benchmarks are more difficult due to stronger spurious correlations, e.g., in Waterbirds only 5% of samples do not exhibit the dominant correlation; similarly only ~7% of training samples lie in the smallest group in CelebA. With randomly sampled batches, it may be difficult to properly regularize KL divergence for samples in the same class but different groups. In fact, we spoke with the PGI authors who also suggested that the extreme spurious correlations in Waterbirds would make it difficult for PGI to work well on this dataset.
> >
> > We then tried a modification to PGI involving rebalanced batches. We use the PGI environment inference labels to sample batches such that each batch has the same number of “anchors”, “positives”, and “negatives” as in CNC (where anchors and positives are same-class samples with different inferred environments). We include full details in Sec. E.2.5. We report the worst-group and average accuracies for this below, and include these results and details in our updated draft. Although this modification improves PGI’s performance, CnC still substantially outperforms it.
> >
> > |               |    CMNIST   |            |  Waterbirds |            |    CelebA   |            |
> > |---------------|:-----------:|:----------:|:-----------:|:----------:|:-----------:|:----------:|
> > | Method / Acc. | Worst-group |   Average  | Worst-group |   Average  | Worst-group |   Average  |
> > | PGI           |  73.5 (1.8) | 88.5 (1.4) |  73.8 (0.8) | 84.6 (0.1) |  77.8 (1.8) | 82.0 (0.6) |
> > | CNC           |  77.4 (3.0) | 90.9 (0.6) |  89.7 (0.2) | 90.8 (0.1) |  88.8 (0.9) | 89.9 (0.5) |
> >
> > Our empirical comparisons suggest that CNC is more effective at improving worst-group accuracy on these benchmarks. We also looked at the alignment loss for these models, and found that CNC models obtained lower alignment loss:
> >
> > |     |    CMNIST   |  Waterbirds |    CelebA   |
> > |-----|:-----------:|:-----------:|:-----------:|
> > | PGI | 0.48 (0.10) | 0.76 (0.03) | 0.77 (0.01) |
> > | CNC | 0.46 (0.01) | 0.60 (0.00) | 0.61 (0.01) |
> >
> > While we think further exploration into these empirical differences would be a good direction for future work, one hypothesis for why CNC may be preferable to PGI intuitively is that the cross-entropy and KL-divergence loss components of PGI may encourage learning different or conflicting optima, leading to more difficult optimization. For example, the KL-divergence term can be satisfied if the model only predicts one class for all samples (e.g., the softmax distributions may be “spiky”). The KL-divergence term can also be satisfied if the predicted probabilities are uniform for all samples. Both of these solutions are not good for learning to discern ground-truth classes, so PGI requires careful tuning to balance this objective with the cross-entropy loss component. On the other hand, the contrastive objective of CNC encourages same-class alignment and different-class separability *at the same time* via the positive and negative samples. This naturally agrees with the cross-entropy loss, and we found that CNC seems to attain high worst-group accuracy across a wide range of values of the balancing loss hyperparameter ($\lambda$ in Eq. 7, see additional ablation in Table A.5).

---

> > > ### Author Response · Authors · 2021-11-20
> > > **Updated paper with additional comparisons, ablations, and clarifications (3/3)**
> > >
> > > **Ablation on the importance of Stage 1 ERM training / just regularizing same class representation distance**
> > > We included this ablation originally in Table 9, Section E.2.2, where we compared CNC with and without the Stage 1 ERM model. In our updated draft we have moved this result to the main paper (denoted as SupCon*), and report these numbers below:
> > >
> > > |         |    CMNIST   |            |  Waterbirds |            |    CelebA   |            |
> > > |---------|:-----------:|:----------:|:-----------:|:----------:|:-----------:|:----------:|
> > > |         | Worst-group |   Average  | Worst-group |   Average  | Worst-group |   Average  |
> > > | SupCon* |  0.0 (0.0)  | 22.4 (1.2) | 71.0 (1.92) | 85.9 (0.8) |  62.2 (1.1) | 90.0 (0.1) |
> > > | CNC     |  77.4 (3.0) | 90.9 (0.6) |  89.7 (0.2) | 90.8 (0.1) |  88.8 (0.9) | 89.9 (0.5) |
> > >
> > > To explain the drop in worst-group performance compared to default CNC, recall that we use the Stage 1 ERM model to set up “hard” positive and negative samples during contrastive learning. Without this, we do supervised contrastive learning. But because of spurious correlations, where many but not all points with the same class have similar spurious features, just aiming to regularize the distances for any pair of same-class points can still encourage the model to rely on similarities between spurious features, leading to poorer worst-group performance.
> > >
> > > We note this ablation without our proposed sampling is similar to [2]. We added a similar ablation just regularizing the L2 distance between representations in Sec. A.1. We find that the contrastive component of CNC leads to superior performance. For details please see our response to Reviewer U9kN.
> > >
> > > **Additional clarifications**
> > >
> > > ***Why does average accuracy also decrease with $p$?***
> > > In our Colored MNIST test set, each group is uniformly represented in the test data (i.e., the test set is balanced). Thus, as we add noise to the Stage 1 ERM model’s predictions, any drop in performance on samples that did not exhibit the dominant spurious correlation (the minority groups in the training set) can also lead to a drop in the average test set accuracy (computed as a simple average over all test samples). If we instead compute a reweighted average to mimic the training set distribution, we indeed see that the average accuracy remains stable $\sim 99.5\%$ for both methods (despite increasing noise, both JTT and CNC lead to models that correctly classify $100\%$ of the unseen test samples that follow the spurious correlations).
> > >
> > > ***Can you intuitively explain why you expect CNC to be more robust than JTT to stage 1 predictions?***
> > > Our intuition is that CNC learns better representations that separate the different groups more compared to JTT in the second stage. This is illustrated in Fig. 6. For example, we can see that CNC’s learned representation exhibits higher margins between waterbirds and landbirds compared to JTT. This higher margin of error may enable CNC to be more robust to label noise from stage 1’s ERM predictions compared to JTT. We hypothesize that CNC is able to learn better representations due to our sampling procedure that encourages the second-stage model to ignore spurious correlations **both** via pushing together anchor-positive pairs, and pushing apart anchor-negative pairs.
> > >
> > >
> > > ***Why is the avg. accuracy of CNC so much worse than other methods on the CivilComments dataset?***
> > > Thanks for the important question. We think the gap in average accuracy is an interesting phenomenon that we are still looking into further.
> > >
> > > ***Minor***: Thanks for pointing out the notation issues. We have updated the corresponding notations in the revised paper.
> > >
> > > [1] Faruk Ahmed, Yoshua Bengio, Harm van Seijen, and Aaron C. Courville. Systematic generalisation with group invariant predictions. In ICLR, 2021
> > >
> > > [2] Arpit, Devansh, Caiming Xiong, and Richard Socher. Predicting with high correlation features. arXiv preprint arXiv:1910.00164 (2019).
> > >
> > > [3] Elliot Creager, Jörn-Henrik Jacobsen, and Richard Zemel.  Environment inference for invariant learning. InInternational Conference on Machine Learning, pp. 2189–2200. PMLR, 2021.

---

> > > > ### Comment · Reviewer_GgTx · 2021-11-23
> > > > **Thanks for the response**
> > > >
> > > > Thanks for providing standard deviation and additional comparisons.
> > > >
> > > > I now see the empirical merit of CNC over PGI and Arpit et.al. (SupCon*).
> > > > I have increased my score.
> > > >
> > > > CNC demonstrated that DRO loss over the recovered group labels (JTT) is inferior when compared with the use of recovered group labels in contrastive loss of CNC. I find that interesting.
> > > >
> > > > However, I still think CNC in its current state is not ready because:
> > > > 1. As you have stated, there is not much conceptual difference between CNC and PGI. PGI performs poorly majorly due to optimization difficulties in minimizing KL-loss.
> > > > 2. I have concerns regarding blanket minimization of pair-wise representation distances from the same class and group. That would collapse different modes of the same class and (overly-)constrain the representation capacity, discouraging the network from learning multi-modal representations of the same class. Perhaps, that is why CNC does poorly on average accuracy on the standard tasks.

---

> > > > > ### Author Response · Authors · 2021-11-25
> > > > > **Responses to remaining concerns**
> > > > >
> > > > > We are glad that our previous revisions and responses could improve our manuscript, and appreciate you raising your score. As it is still borderline, we would appreciate it if you could take into account our responses to your standing concerns. Please find our responses below; we hope they can further improve the paper, resolve your concerns, and allow us to reach a more satisfying consensus.
> > > > >
> > > > > **"As you have stated, there is not much conceptual difference between CNC and PGI. PGI performs poorly majorly due to optimization difficulties in minimizing KL-loss."**
> > > > >
> > > > > We stated that PGI and CNC may share some of their goals, but we do not think CNC should be rejected on the basis of this. CNC and our work are still demonstrably novel because of methodological differences and improvements.
> > > > >
> > > > > Prior work before PGI also seeks to align representations via "invariant learning" (e.g. IRM [1], which we compare against via EIIL, Table 1), yet PGI was still accepted at ICLR 2021. PGI was also not the first to tackle group robustness without group information (c.f. GEORGE proposed by Sohoni et al., NeurIPS 2020 [2]). We think that despite the existence of this prior work, PGI's method alone is still a novel and significant contribution, with a significant difference being *how* PGI accomplishes alignment. A large part of PGI’s merit comes from these differences being linked to improved performance, which also contributes new understanding of different approaches and how they perform.
> > > > > * Likewise, we believe CNC also delivers meaningful contributions via (1) a new way to accomplish alignment to improve robustness (to spurious correlations), (2) substantially improved empirical performance, (3) increased understanding and intuitive rationale for why CNC may work better than prior approaches.
> > > > >
> > > > > **"I have concerns regarding blanket minimization of pair-wise representation distances from the same class and group. That would collapse different modes of the same class and (overly-)constrain the representation capacity, discouraging the network from learning multi-modal representations of the same class. Perhaps, that is why CNC does poorly on average accuracy on the standard tasks."**
> > > > >
> > > > > Thanks for these insightful remarks. We agree that in the ideal solution, class representations would not collapse to single points. However, the concerns stemming from your logic here would also apply to SimCLR [3] and SupCon (among other contrastive learning approaches), the latter of which demonstrates superior performance to supervised learning with the cross-entropy loss [4] (c.f. Table 3).
> > > > >
> > > > > Wang & Isola, 2020 [5] also show that the contrastive methods may prevent this class collapse due to "uniformity", which prefers a feature distribution that preserves maximal information. This is a second property (other than alignment) which the contrastive loss optimizes for. We did not discuss this earlier due to alignment's greater connection to our problem of reducing dependence on spurious correlations, where the concern is that features with the same class but different spurious attributes are not aligned or brought close together with standard ERM. In the camera-ready, we will include this discussion in an updated Section A.1, where we explain the benefits of CNC and our contrastive loss compared to training with the alignment loss alone (where the alignment loss alone could be encouraging this intra-class collapse for all points with the same class).
> > > > >
> > > > > That being said, we believe CNC can inspire future work where alternative contrastive objectives are swapped in, which may encourage and allow for greater intra-class representation entropy and multi-modality. However, via CNC we want to first introduce this contrastive step itself as a general method and new way to align representations for the spurious correlations setting, and wish to encourage future work that builds on this direction.
> > > > >
> > > > > [1] Arjovsky, Martin, Léon Bottou, Ishaan Gulrajani, and David Lopez-Paz. Invariant risk minimization. arXiv preprint arXiv:1907.02893 (2019).
> > > > >
> > > > > [2] Sohoni, Nimit S., Jared A. Dunnmon, Geoffrey Angus, Albert Gu, and Christopher Ré. No subclass left behind: Fine-grained robustness in coarse-grained classification problems. Advances in Neural Information Processing Systems (2020).
> > > > >
> > > > > [3] Chen, Ting, Simon Kornblith, Mohammad Norouzi, and Geoffrey Hinton. "A simple framework for contrastive learning of visual representations." In International conference on machine learning, pp. 1597-1607. PMLR, 2020.
> > > > >
> > > > > [4] Khosla, Prannay, Piotr Teterwak, Chen Wang, Aaron Sarna, Yonglong Tian, Phillip Isola, Aaron Maschinot, Ce Liu, and Dilip Krishnan. "Supervised contrastive learning." Advances in Neural Information Processing Systems (2020).
> > > > >
> > > > > [5] Wang, Tongzhou, and Phillip Isola. Understanding contrastive representation learning through alignment and uniformity on the hypersphere. In International Conference on Machine Learning, pp. 9929-9939. PMLR, 2020.

---

> > > > > > ### Comment · Reviewer_GgTx · 2021-11-29
> > > > > > **Thanks for the further clarification.**
> > > > > >
> > > > > > I could not follow your response to my second concern.
> > > > > >
> > > > > > *Regarding the first concern*:
> > > > > > I see the differences between PGI and yours, however I do not perceive that to warrant an accept (from me).
> > > > > > There is a long line of work on representation alignment starting from Ganin et.al. '15, IRM '19, etc. as you have stated. I understand that high-dimensional feature alignment is easier said than done, which may be why we see so many techniques with this objective. Instead of presenting yet another alignment technique, what I would like to see is a discussion on existing alignment methods and how they would/could fail when population sizes are highly disproportionate or their optimization issues.

---

### Official Review · Reviewer_U9kN · 2021-11-01

**Correctness:** 4
**Technical Novelty And Significance:** 2
**Empirical Novelty And Significance:** 3
**Recommendation:** 6
**Confidence:** 4

**Main Review:**


This paper takes a representation focused perspective on avoiding learning spurious corrections. The main motivation (well backed by experiments on toy datasets) is this: there is a positive correlation between within-class alignment loss, and worst-group error. On its own this observation isn’t hugely surprising, and arguably overlaps with observations in prior work. The novelty of this work comes from using this observation as the inspiration for CnC, a supervised contrastive approach to re-training the ERM model to reduce the within-class alignment loss.

Some positives:

- The paper is well written
- The method is easy to implement, intuitive, and seems to work pretty well.
- In section 3 the combination of empirical observation and theoretical bound make the conclusion quite convincing.
- The method specifically samples hard positive/negative samples. This is potentially an even more important point on the novelty of the method than is currently emphasized [see the second main question below, about SupCon as a baseline].
- The logic and ideas in this paper are very linear, making it easy to quickly grasp the main takeaways.
- The worst-group performance seems to decay slightly better than JTT as the level of spurious correlation increases (Fig. 7).

(For weaknesses, see the questions below).


---
## Questions:

I have *two major questions that I am hoping to see answers to:*

First: The motivation for CnC is centered on the class-conditional alignment loss. There is even a bound on the worst-group loss in terms of the average-group and class-conditional alignment loss. So why not replace step 2, and instead fine-tune the model using L_avg_group + L_alignment? Or even just train models from scratch with this loss. It would be good to compare to these. If CnC is simply more empirically successful than this alternative, then it would be good to see this.

Second: In a related vein to the previous question, how much is CnC buying us as compared to the usual supervised contrastive training? It would be good to see SupCon as a baseline in Table 1. This seems an important baseline, since the main idea of CnC is to pull items from the same class together in feature space, which is also done using SupCon.  The main (even only?) difference is the hard positive/negative sampling approach of CnC.


---

## Miscellaneous comments and questions:

These are just a few things I was curious about. I am not per se asking for any response from the authors, but offer them up in the spirit of constructive feedback:

- What if you iterate your method? That is, CnC samples positives and negatives according the *fixed* ERM model from step 1. What if you repeat step 2 again using the new and improved model obtained from the first step 2 run? Maybe it would just immediately saturate in performance, but I am curious.
- Is there a way to incorporate spurious attribute information into the positive and negative sampling methods if it were available?
- Do you have a rationale as to why CnC did worse than JTT for CivilComments? I’m not bothered at all by this result, since I would never ask for across-the-board improved empirical results. But I am curious at to whether any lessons can be learned about the relative strengths and weaknesses of the two methods. I notice that CivilComments has more (8) spurious feature values than the other datasets - could this be related?
- Perhaps consider using more divergent colors in Fig 6. The different shades appear fine on my compute screen, but are hard to distinguish on a printout (maybe my printer is just bad…).


**Summary Of The Paper:**

This paper considers the goal of training classifiers that achieve strong worst-case performance across groups with different spurious features, without assuming access to supervision based on the spurious features. The paper proposes a two-step method, which fist trains a model with ERM, then re-trains the model using a supervised contrastive approach where positive and negative samples are selected on the basis of misclassification characteristics of the ERM model.

 The method addresses an important problem, and is well motivated & evaluated. In all this is a nice piece of work and I am cautiously happy to recommend its acceptance at this point. I do, however, have a couple of questions I would like to see some answers to (see below). Depending on the answers to these questions, and discussion with other reviewers, I am happy to consider raising my score.


**Summary Of The Review:**


 The method addresses an important problem, and is well motivated and evaluated. In all this is a well executed piece of work and I am cautiously happy to recommend its acceptance at this point. I do, however, have a couple of questions I would like to see some answers to (see above). Depending on the answers to these questions, and discussion with other reviewers, I am happy to consider raising my score.

---

> ### Author Response · Authors · 2021-11-20
> **Updated paper to address main questions, additional responses to all questions below (1/2)**
>
> Thank you for your positive comments and support for CNC! We appreciate your generous comments on our easy to implement and intuitive method with convincing empirical and theoretical motivation.
>
> We have updated the paper to answer your two major questions, and respond to these questions below. Following this, we respond to your additional comments and questions.
>
> **Why not train the model with $\mathcal{L}_\text{avg group} + \mathcal{L}_\text{alignment}$?**
>
> Thanks for the great suggestion. We have added a comparison to this objective in Table A.1. We replace the second stage training objective of CNC with the alignment loss. We use the same batch sampling procedure, but instead of training the encoder layers with a contrastive loss, we minimize $\mathcal{L}_\text{alignment}$ for each batch (between the anchor and positive pairs). We also report these results below:
>
> |                 |    CMNIST   |            |  Waterbirds |             |    CelebA   |             |
> |-----------------|:-----------:|:----------:|:-----------:|:-----------:|:-----------:|:-----------:|
> |                 | Worst-group |   Average  | Worst-group |   Average   | Worst-group |   Average   |
> | Loss Align | 67.3 (3.1)      | 83.7 (1.1)      | 83.3 (0.8)  | 89.4 (0.2)  | 84.6 (0.5)  | 88.0 (0.3) |
> | Default Contrastive             | 77.4 (3.0)  | 90.9 (0.6) | 89.7 (0.2)  | 90.8 (0.1)  | 88.8 (0.9)  |  89.9 (0.5) |
>
> We find that CNC with the contrastive loss outperforms CNC with the alignment loss. One advantage of CNC is that with the contrastive loss (and specifically the “hard” positive and negative samples), we both align same-class samples with different spurious features, *and* push apart hard negative samples in different classes but with similar spurious features. This provides additional signal for improving separation between the different classes, so the robust model only learns to rely on ground-truth-specific features for discriminating between datapoints. On the other hand, the $\mathcal{L}_\text{avg group} + \mathcal{L}_\text{alignment}$ objective does not incorporate these hard negatives. Optimizing this objective also requires careful balancing of the loss components: while $\mathcal{L}_\text{alignment}$ only explicitly aligns samples in the same class but different groups, it alone can also be reduced by mapping *all* samples to similar representations. We thus require balancing this with the cross-entropy term to maintain class-separability. CNC may be preferable because the contrastive objective with positives and negatives encourages class-specific alignment and class separability together.
>
> Finally, we also tried applying $\mathcal{L}_\text{avg group} + \lambda \mathcal{L}_\text{alignment}$ over standard random minibatches, which can be viewed as a regularized version of ERM. We added the $\lambda$ hyperparameter following our discussion above. However, despite finetuning $\lambda \in$ {$0.1, 0.5, 1, 10, 100$}, we were not able to obtain meaningfully better worst-group performance over ERM.
>
> **Or even just train models from scratch with this loss?**
>
> Thanks for the excellent question. Based on our experience we suspect this would be an inferior  approach compared to the two-stage approach that we (and several previous works) adopt. Due to time constraints, we will include this comparison in the next version of the paper. Note that to do loss alignment training from scratch, we would need to know the group labels beforehand. The comparison is thus more suitable against GDRO, and CNC with known group labels.

---

> > ### Author Response · Authors · 2021-11-20
> > **Updated paper to address main questions, additional responses to all questions below (2/2)**
> >
> > **How much is CnC buying us compared to usual supervised contrastive training?**
> >
> > We have updated the main table (Table 1) in the paper to include this ablation. In our prior submission, we originally included these results for SupCon in Table 9 in the Appendix (Section E.2.1) on the Waterbirds and CelebA datasets. We also added this comparison with CMNIST, and include the SupCon vs CNC results below:
> >
> > |        |    CMNIST   |            |  Waterbirds  |             |    CelebA   |            |
> > |--------|:-----------:|:----------:|:------------:|:-----------:|:-----------:|:----------:|
> > |        | Worst-group |   Average  |  Worst-group |   Average   | Worst-group |   Average  |
> > | SupCon |  0.0 (0.0)  | 22.4 (1.2) | 71.0 (1.9)  |  85.9 (0.8) |  62.2 (1.1) | 90.0 (0.1) |
> > | CNC    | 77.4 (3.0)  | 90.9 (0.6) |  89.7 (0.2)  | 90.8 (0.1)  | 88.8 (0.9)  | 89.9 (0.5) |
> >
> > Performance drops without the hard negative and positive sampling; worst-group accuracy in particular worsens substantially without using the Stage 1 model to help sample points based on learned spurious correlations and inferred groups. We reason this is because merely encouraging the network to pull together same-class points may not necessarily prevent the model from learning spurious correlations. In our setting, a majority of samples with the same class exhibit similar spurious features. If a majority of the samples that are being pulled together have the same spurious attributes, and likewise a majority of the samples being pushed apart have different spurious attributes, then (similar to ERM training) the network may rely on spurious features to optimize the contrastive objective, i.e. learning similar representations for anchors and positives based on similar spurious features, while learning dissimilar representations for anchors and negatives due to different spurious features.
> >
> > **What if we iterate on our method?**
> > We think this is an interesting extension to our proposed work. While the idea makes sense, for it to work we need the Stage 2 model to also incorrectly predict training samples. With our current number of training epochs, the Stage 2 model obtains close to 100% accuracy on the training samples (~99.9% on Waterbirds).
> >
> > Because of time constraints, we will explore variations of the suggested protocol further in the next version of the paper. It may be the case that running Stage 2 for only a few epochs, before using remaining mistakes to train a new Stage 3 contrastive model, can lead to a more robust model at the end.
> >
> >
> > **Is there a way to incorporate spurious attribute information into the positive and negative sampling methods if it were available?**
> > Yes! If we have spurious attribute information, we can set up our contrastive batches such that given an anchor, we train a network to learn similar representations only for points with the anchor’s class, but different known spurious attribute values. Similarly, we can sample negatives from points with different classes, but the same known spurious attribute value as the anchor. We are currently running these experiments.
> >
> > **Do you have rationale for why CnC did worse than JTT for CivilComments?**
> > Thanks for the important question. We are still exploring possible answers for this. We ran additional seeds with CNC, and note that based on the error bars, the difference for worst-group accuracy is not significant. The gap in average accuracy is still an interesting question that we are looking into further.
> >
> > **Perhaps consider using more divergent colors in Fig 6.**
> > Thanks for this suggestion. We will update the figure with more divergent colors.

---

> > > ### Comment · Reviewer_U9kN · 2021-11-29
> > > **Thanks for the additional details**
> > >
> > > Hi authors,
> > >
> > > I just wanted to quickly note here that I very much appreciate the additional experiments you ran, and the thoughts you shared on some of the other design components.
> > >
> > > In particular I am given more confidence seeing the sanity check comparison to vanilla SupCon - thanks! The other observation, on using $\mathcal{L}_{avg} + \mathcal{L}_\text{align}$, is also valuable. It's good to see that this version does something fairly non-trivial (e.g.,  its worst case performance on CelebA would be 2nd best in Table 1, and on Waterbirds would be 3rd best). To me this gives confidence in the basic motivation by showing its value can (fairly) successfully manifest in more than one possible method. The choice to go with CnC then comes down to simply being empirically superior & algorithmically more convenient (no need to tune this $\lambda$).
> > >
> > > In summary, this additional information cements my confidence in what is going on, and I will maintain my generally positive assessment of your work:)

---

> > > > ### Author Response · Authors · 2021-12-03
> > > > **Thanks for your review!**
> > > >
> > > > Thanks for your positive feedback and suggested experiments!  (also apologies for this delayed comment). We were happy to add both the comparison to SupCon and the alignment loss, which we agree provides further insight into CNC's benefits over alternative approaches.

---

### Author Response · Authors · 2021-11-22
**Thank you for reviews; shared response + summary of paper updates**

We thank the reviewers for their thoughtful reviews and insightful comments. We are glad that they appreciated our well-motivated and intuitive method Correct-N-Contrast (CNC) (U9kN, ktqR), our important and practical setting to improve robustness to spurious correlations without knowledge of training data spurious attributes or group labels (GgTx, FMKR), and our theoretical justification and empirical analysis to motivate and understand CNC’s SoTA performance (U9kN, FMKR, ktqR).

**On comments related to novelty**
We emphasize that our claimed contribution is not just to align representations. Rather, for our scope of improving robustness to *spurious correlations without group labels*, we specifically introduce new and natural connections with contrastive learning to propose a new way to align representations in this setting. We demonstrate that for our spurious correlations problem setting, this alignment approach works better than other directions and methods previously considered.
- We enhance the standard anchor-positive-negative sampling from the contrastive learning literature with an intuitive and novel hard sampling strategy that encourages learning class-specific representations while ignoring spurious features.
- We discuss differences between our method and problem setting with both related methods and related problem settings (including unsupervised domain adaptation and domain generalization) in our updated related works (Section 6, Appendix D).
- We also show that these differences are meaningful and significant, as CNC achieves substantial improvement in worst-group (robust) accuracy compared to prior methods designed for our setting and related problem settings (Table 1, Table A.2).

We would also like to emphasize that as a method, CNC has demonstrated significant progress towards resolving the spurious correlations problem. We obtain state-of-the-art performance on several well-known image datasets, and significantly close the worst-group accuracy gap with existing methods that require knowledge of training data *group* labels or spurious attributes.

**Summary of updates to paper**
We appreciate the reviewers’ thoughtful suggestions on improving the paper. We have updated the paper following reviewers’ suggestions. We added additional ablations on CNC and empirical comparisons to related work, as well as extensions to the original theory that are now consistent with our experimental settings. We summarize these updates below:

- To demonstrate the merit of CNC’s contrastive learning objective, we added experiments comparing CNC to:
  - Minimizing the alignment loss directly (Appendix A.1, Table A.1).
  - *Predictive Group Invariance* (PGI), which provides an alternative approach to aligning representations (Table 1, discussed in Section 6, Section D.3).
  - *Domain adversarial neural networks* (DANN), an unsupervised domain adaptation method that also provides an alternative approach to aligning representations (Table A.2, discussed in Section A.2, Section D.3).

- To demonstrate the merit of CNC’s proposed “hard” positive and negative contrastive sampling strategy and additional justification for the Stage 1 ERM model, we:
  - Moved experiments comparing CNC to standard supervised contrastive learning (SupCon) up to the main table (Table 1). (SupCon just samples positives and negatives based on whether they have the same or different class as their anchor datapoints.)
  - Added ablations comparing default CNC to CNC trained with several additional sampling strategies (Table A.3)

- To show that CNC’s theoretical motivation holds for assumptions that are well-motivated and satisfied under experimental settings, we extended the proof of Theorem 3.1 to cases where the loss function is Lipschitz continuous for any fixed constant and upper bounded by a fixed constant as well (Appendix B). The original theoretical takeaways motivating CNC still hold.

Finally, while we focus on tackling the problem of spurious correlations, we believe our work could potentially inspire future works to use contrastive learning to address related problems, such as those arising in domain generalization settings to improve out-of-distribution robustness.

---

### Decision · Program_Chairs · 2022-01-20

**Decision:**

Reject

**Comment:**

The paper worked on an important problem (robustness concerning spurious correlations) and proposed a useful method (achieving SOTA worst-group performance without the true group labels). However, the motivation part is weak so that it is unclear why to go from the theoretical/empirical observations to the proposed method (more specifically the contrastive learning part). The novelty is also not very strong as argued by reviewers (the real novelty was not highlighted by the authors and thus cannot easily be appreciated by readers). It is indeed a borderline case but seems to be below the bar of acceptance and the two reviewers staying on the positive sides would not like to fight for it. Since there is still room for improvement, we hope the paper would benefit from a cycle of revisions for a re-submission and the improved version would be accepted in the near future.

By the way, what GgTx suggested is not really an out-of-scope study, as far as I understood. The authors certainly think that the paper/method has been clearly motivated. This reviewer was asking for a strong motivation, namely, what is missing or what is wrong in existing methods or the SOTA method so that we need/have to apply the proposed method? Without clarifying this point, the paper/method is partially but not fully motivated and the method may look like another alternative though it should be a better one. Instead of showing the better performance, the reviewer would like to see the conceptual advantage of the proposal by understanding what is missing/wrong in the current SOTA method. Therefore, I think this is a great question for the authors to maximize the impact of their work in the end.